# Vector Linking via Cross-Model Local Isometric Consistency

**Ziying Chen** [1]  **Yang Cao** [1]  **He Sun** [1 2]  **Beining Yang** [1]  **Tianjian Yang** [1]

## Abstract

We study *Vector Linking*: given two embedding clouds produced by different black-box encoders over partially overlapping datasets, recover cross-model object correspondences using only vectors. Empirically and theoretically, we show that independently trained contrastive encoders exhibit local geometric consistency: short-range distances are approximately preserved up to a scale factor, while long-range distances are not due to model-specific distortion. Building on this, we propose an iterative, reference-based geometric embedding hashing that recovers vector links from a tiny seed set of paired anchors. It represents each vector by distances to sampled paired anchors, proposes candidate links via hash-space matching, and aggregates evidence across views in a Beta–Bernoulli posterior to bootstrap high-confidence links as new anchors. Experiments across multiple benchmarks and embedding model pairs demonstrate accurate and robust linking under varying overlap, seed budgets, and out-of-domain anchors, with applications to vector database integration and cross-model clustering. Code is available at https://github.com/DBgroup-Edinburgh/VecLinking.

## 1. Introduction

Information systems increasingly rely on embedding-based retrieval: large collections of objects are mapped to vectors and indexed for similarity search. In practice, however, embedding models evolve quickly, and different systems often adopt different fine-tuned encoders. As a result, practitioners are left with multiple vector indices whose representations are not directly comparable, even when those indices contain many of the same objects. This interoperability gap hinders unified retrieval, cross-index deduplication, joint clustering, and vector database integration.

**Vector Linking.** We study *vector linking*: recovering which vectors in two embedding clouds correspond to the same underlying object when the clouds are produced by different black-box contrastive encoders and overlap only partially. Formally, let $\mathcal{O}_1$ and $\mathcal{O}_2$ be two datasets of objects with *unknown* overlap $\Omega = \mathcal{O}_1 \cap \mathcal{O}_2$. Let $f_1$ and $f_2$ be two encoders, and let $\mathbf{E}_1 = f_1(\mathcal{O}_1)$ and $\mathbf{E}_2 = f_2(\mathcal{O}_2)$ be the resulting embedding sets. We assume access only to $\mathbf{E}_1$, $\mathbf{E}_2$, and a small seed set of paired anchors $S \subseteq M^*$, where $M^* = \{(f_1(x), f_2(x)) : x \in \Omega\} \subseteq E_1 \times E_2$. The goal is to recover as many pairs in $M^*$ as possible without access to the raw objects, model parameters, gradients, or retraining.

This setting differs from standard embedding alignment in two important ways. First, the overlap is *partial and unknown*: there is no global bijection between the two embedding sets, and the non-overlapping regions do not simply behave like outliers. Instead, they can substantially alter the global geometry seen by each encoder. Second, we target a strict *post-hoc black-box* regime. Many compatibility and alignment methods assume access to training data, encoder internals, or training-time intervention; here, only static vectors are available. Together, these two properties make a single global transformation unreliable.

**Local isometric consistency**. Our starting point is a simple but robust finding. When we compare pairwise distances between shared objects across independently trained contrastive encoders, short distances remain strongly correlated while long-range distances decorrelate quickly. Equivalently, small neighborhoods are far more stable across models than the global arrangement of the embedding clouds. Theoretically, we show that this pattern is not merely an empirical coincidence: by analyzing a localized alignment-uniformity surrogate for contrastive learning, under standard assumptions, we show that independently trained contrastive encoders can induce locally isometric metrics up to scale.

**Geometric embedding hashing**. Motivated by this observation, we propose *Geometric Embedding Hashing* (GEH). The basic unit of GEH is a *distance-to-anchor signature*: given a small set of paired anchors, each vector is represented by its distances to those anchors within its own embedding space. If two vectors correspond to the same object, and if the chosen anchors lie in their local neighborhoods,

[1]School of Informatics, University of Edinburgh, Edinburgh, United Kingdom [2]Shenzhen Institutes of Advanced Technology, Chinese Academy of Sciences, Shenzhen, China Correspondence to: Yang Cao <yang.cao@ed.ac.uk>. *Proceedings of the $43^{rd}$ International Conference on Machine Learning*, Seoul, South Korea. PMLR 306, 2026. Copyright 2026 by the author(s).

then these relative distance patterns should remain similar up to scale even when the global shapes of the two embedding clouds differ markedly. GEH therefore compares normalized, scale-free signatures rather than raw distances.

A single anchor set is not locally informative for every point, so GEH does not rely on one global hash. Instead, it repeatedly samples many small anchor subsets, or *views*, matches points independently in each induced hash space, and treats resulting matches as noisy votes. A Beta-Bernoulli posterior aggregates evidence across views, and high-confidence matches are promoted as new anchors for the next round. This multi-view bootstrapping lets GEH grow a tiny seed set into a large correspondence set while gracefully filtering spurious collisions caused by model-specific distortion and partial overlap.

We evaluate GEH on multiple BEIR benchmarks and five encoder pairs spanning both API-based and open-weight models. Across varying overlap ratios, seed budgets, and out-of-domain seed settings, GEH consistently outperforms eight linear, nonlinear, and optimal-transport baselines using only 15 to 30 seed pairs. For instance, with only 15 paired seeds, GEH achieves over 90% recall on FiQA (Maia et al., 2018) for `Mistral` and `OpenAI`. We further show that the recovered links improve downstream tasks including vector database integration and cross-model clustering.

The results suggest that vector linking is a practical primitive for embedding interoperability, and that local geometric consistency across contrastive encoders holds key to tackle it.

**Contributions & organization.** We contribute as follows.

- We propose *vector linking*, the problem of recovering correspondences between two black-box embedding clouds under partial, unknown overlap.

- We establish, both empirically and theoretically, a cross-model *local distance consistency* property, forming the foundation of encoder-invariant hashing (Section 2).

- We develop a multi-view geometric hashing algorithm with posterior-guided bootstrapping that accurately recovers vector links without accessing raw objects or model internals, using only tiny seeds (Sections 3-4).

- We demonstrate accurate and robust linking across multiple benchmarks and embedding model pairs (Section 5).

- We further demonstrate its benefits to vector database integration and cross-model clustering (Section 6).

**Related Work.** Geometric point set registration under partial overlap has been studied via hypothesis testing (e.g., RANSAC (Fischler & Bolles, 1981), TEASER++ (Yang et al., 2020)), iterative refinement (e.g., ICP (Besl & McKay, 1992), Go-ICP (Yang et al., 2016)), and invariant signatures (geometric hashing (Lamdan & Wolfson, 1988)). These tools are primarily designed for 3D rigid space and can-

not handle high-dimensional heteroscedastic model-induced distortion and unknown overlap that vector linking targets.

Embedding alignment methods for *e.g.,* bilingual lexicon induction, learn a global mapping between spaces (linear/Procrustes, OT/GW) (Mikolov et al., 2013; Xing et al., 2015; Smith et al., 2017; Lample et al., 2018; Artetxe et al., 2018; Alvarez-Melis & Jaakkola, 2018; Grave et al., 2019), often relying on an approximate global isomorphism. Such global consistency has also been exploited for domain adaptation (Shen et al., 2021; Hu et al., 2022; Wang & Mahadevan, 2011; Wang et al., 2018; Ganin et al., 2016; Hoffman et al., 2017), which further demand training-time access not available in the black-box vector linking setting.

Unlike embedding alignment that seeks a coupling that makes the *entire* spaces globally comparable, vector linking instead seeks a *partial* one-to-one correspondence relation on the unknown shared support, while leaving vectors outside the overlap unmatched. This creates an objective mismatch for global alignment, as there is no global bijection to recover. Further, non-overlapping regions are structured and potentially large, so they do not behave like removable outliers. Alignment can thus improve global fit on unmatched regions while worsening correspondences on the overlap.

Vector linking bootstraps downstream interoperability tasks such as cross-model vector database integration (Yang et al., 2025) and joint clustering (Enevoldsen et al., 2025), which assumes that that reliable cross-model anchor pairs are already known. Vector linking addresses this assumption by recovering correspondences from black-box vector clouds.

**Conflict of Interest Disclosure**. The authors declare no financial conflicts of interest. All authors are affiliated solely with academic institutions and the embedding models evaluated in this work are independent third-party systems.

## 2. Foundation of Embedding Hashing

This section provides the geometric foundation behind the idea of encoder-invariant geometric hashing. We first establish an empirical short-to-long range transition in cross-model distance consistency (Section 2.1). We then provide a localized geometric explanation for why contrastive encoders tend to preserve local geometry (Section 2.2).

### 2.1. Emergence of Local Distance Consistency

We begin by quantifying how pairwise euclidean distances compare across embedding spaces. Let $\mathbf{E}_1$ and $\mathbf{E}_2$ be embeddings of the same raw dataset $D$ produced by two different encoders (*e.g.,* `Mistral` vs. `OpenAI`). We sample pairs $(u, v)$ of objects from $D$, compute $d_{\mathbf{E}_1}(u, v)$ and $d_{\mathbf{E}_2}(u, v)$, and bin pairs by $d_{\mathbf{E}_1}(u, v)$. We report, per bin, the Pearson correlation between $d_{\mathbf{E}_1}(u, v)$ and $d_{\mathbf{E}_2}(u, v)$ in

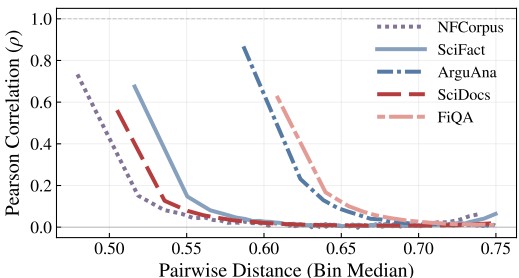

*Figure 1.* **Consistency (linear correlation)** Vs. **vector distances:** The x-axis shows the pairwise distance in the reference space (Mistral), while the y-axis reports the Pearson correlation ($\rho$) of these distances with their counterparts in the target space (OpenAI).

Fig. 1 across multiple BEIR benchmarks (See C.1 for more).

**Local consistency.** For short distances, *e.g.,* $d_{\mathbf{E}_1}(u, v) \lesssim 0.57$ for ArguAna, the correlation is substantially positive with $\rho$ above 0.8. Further, as shown in Appendix A.5, we also find that top-$k$ retrieval exhibits strong consistency with small $k$ (*e.g.,* $k < 10$) across embeddings compared to large $k$, which indicates that nearby pairs under $\mathbf{E}_1$ tend to remain nearby under $\mathbf{E}_2$. This seems to suggest that a linear correlation in the short-distance regime has substantially more statistical significance compared to long distances.

**Global Decorrelation** ($\rho \approx 0$). As distances increase, the correlation decays rapidly, consistently nearing zero. This collapse indicates that the scaling factor $\alpha$ is not globally constant. While the models agree on the "shape" of local neighborhoods, they diverge significantly on the global arrangement and distribution of data objects. This renders long-range distances inconsistent across embedding spaces.

This suggests that short distances are consistent across encoders, and thus a distance-to-anchor vector that consists of short distances can be a viable choice for encoder-invariant geometric hashing. We have also observed that such correlation is significant weaker for non-contrastive encoders (see Appendix A.4 for more). To further confirm this, we still want to check if this is just an empirical coincidence or something fundamental to the embedding models.

### 2.2. A Geometric Justification of Local Isometries

We give a geometric explanation for the short-distance consistency of Fig. 1. We show that the phenomenon is inherent to contrastive encoders rather than an empirical coincidence.

**Geometric modeling.** We model data as a random variable $X$ supported on a smooth $d$-dimensional manifold $\mathcal{M} \subset \mathbb{R}^N$ with density $p(x)$ *w.r.t.* the intrinsic Riemannian volume measure on $\mathcal{M}$. We denote geodesic distance by $d_{\mathcal{M}}(x, y)$, which serves as an intrinsic notion of semantic dissimilarity between the data objects modeled by $x$ and $y$.

An embedding model (*a.k.a.* encoder) is a map $f : \mathcal{M} \to \mathbb{R}^K$ with normalized outputs $f(\mathcal{M}) \subset S^{K-1}$. Let the Jacobian of the encoder $f$ at $x$ be $J_f(x) \in \mathbb{R}^{K \times d}$, where $d$ is the intrinsic dimension of $\mathcal{M}$; it maps tangent vectors from the data manifold to the embedding space. We denote the metric tensor induced by the encoder $f$ as $G_f(x) := J_f(x)^\top J_f(x) \in \mathbb{R}^{d \times d}$, which characterizes how local distances are distorted by the map. (A1) We assume that $f$ is twice differentiable and injective, and that $G_f(x)$ is positive definite for all $x \in \mathcal{M}$, *i.e.,* $J_f(x)$ has full rank $d$.

**Short-range neighborhoods.** For each $x \in \mathcal{M}$, let $\delta_{\mathcal{M}}(x) > 0$ be such that whenever $d_{\mathcal{M}}(x, y) < \delta_{\mathcal{M}}(x)$ the shortest geodesic from $x$ to $y$ on $\mathcal{M}$ is unique. For such $y$, define the geodesic displacement $v(x, y) \in T_x \mathcal{M}$ as the tangent vector at $x$ pointing toward $y$ along this unique shortest geodesic, normalized so that $\|v(x, y)\| = d_{\mathcal{M}}(x, y)$.

**Contrastive learning**. We consider encoders trained via contrastive learning with InfoNCE-type contrastive loss objectives. The training signal comes from: (i) positive pairs, which are two semantic-preserving "views" of the same data object, and (ii) negative pairs, which pair unrelated data objects. Given $x \in \mathcal{M}$, a "positive view" $x^+$ is sampled by a stochastic augmentation. We assume the following. (A2: local positives) $d_{\mathcal{M}}(x, x^+) < \delta_{\mathcal{M}}(x)$ almost surely. (A3: local isotropy) Following (Dao et al., 2019; Wang & Isola, 2020), we assume that the distribution of $x^+$ is centered and isotropic on the local tangent space. Specifically, fix $x \in \mathcal{M}$ and let the geodesic displacement vector $v = v(x, x^+) \in T_x \mathcal{M}$. We assume $\mathbb{E}[v \mid x] = 0$ and $\mathbb{E}[vv^\top \mid x] = c \, I_d$. (We discuss relaxations to $c = c(x)$ in Appendix A.3.)

**Global contrastive surrogate**. We adopt the standard alignment-uniformity perspective on contrastive learning (Wang & Isola, 2020; Zimmermann et al., 2021): positives should be mapped close (alignment), while the overall representation distribution should be spread out to avoid collapse (uniformity). Let $X$ be the random data point on $\mathcal{M}$ and let $X^+$ denote its positive view. Write $Z := f(X)$ for the induced random representation.

Following (van den Oord et al., 2018; Poole et al., 2019), for InfoNCE-like contrastive losses, alignment loss minimizes $\mathbb{E}\big[\|f(x) - f(x^+)\|^2\big]$, and uniformity can be modeled by maximizing the entropy of the representation, which we interpret intrinsically on $f(\mathcal{M})$ as the standard differential entropy (Cover & Thomas, 2006). Specifically, let $q$ denote the density of $Z$ on $f(\mathcal{M})$ *w.r.t.* the induced $d$-dimensional surface volume on $f(\mathcal{M})$, and define entropy $H(Z) := -\mathbb{E}\big[\log q(Z)\big]$. Then we have the implementation-independent surrogate of contrastive loss:

$$\mathcal{L}_\lambda(f) := \mathbb{E}\big[\|f(X) - f(X^+)\|^2\big] - \lambda H(Z),$$

where $\lambda > 0$ is a model-dependent coefficient that balances alignment and uniformity. This surrogate is not identical to

InfoNCE, but captures its geometric pressure toward (i) local alignment of positives and (ii) spread of representations.

**A localized geometric view**. As we focus on local geometric properties, we develop a localized view of the global contrastive loss surrogate. Note that $\mathcal{L}_\lambda(f)$ is an expectation over $X$, it can be written as an average of per-point contributions. Hence, we can write $\mathcal{L}_\lambda(f)$ equivalently as

$$\mathbb{E}\Big[ \underbrace{\mathbb{E}\big[\|f(X)-f(X^+)\|^2 \mid X\big]}_{\varphi_{\text{align}}:\text{local alignment at } X} + \underbrace{\lambda \log q(f(X))}_{\varphi_{\text{uni}}:\text{local uniformity at } X} \Big].$$

Motivated by this, we define a localized loss at $x \in \mathcal{M}$, denoted by $\mathcal{L}_\lambda(x; f)$, such that $\mathcal{L}_\lambda(f) = \mathbb{E}[\mathcal{L}_\lambda(X; f)]$; hence

$$\mathcal{L}_\lambda(x; f) := \varphi_{\text{align}}(X = x) + \varphi_{\text{uni}}(X = x).$$

Under the encoder assumption (A1), by area formula and change-of-variables (Lee, 2003) we have $q(f(x)) = p(x)/\sqrt{\det(G_f(x))}$. Therefore, up to an $x$-dependent constant not involving $f$, we have $\mathcal{L}_\lambda(x; f) = \varphi_{\text{align}}(X = x) - \frac{\lambda}{2} \log \det(G_f(x)) + \text{const}(x)$. Further by the short-range data augmentation (A2, A3), $\varphi_{\text{align}}(X = x) \approx c \cdot \text{tr}(G_f(x))$ (ignore higher-order terms; see Corollary B in Appendix A). Thus the leading-order localized geometric objective is

$$\widetilde{\mathcal{L}}_\lambda(x; f) := c \cdot \text{tr}(G_f(x)) - \frac{\lambda}{2} \log \det(G_f(x)).$$

**Locally optimal encoders**. We say that an encoder $f$ is *locally optimal at* $x \in \mathcal{M}$ (*w.r.t.* $\lambda$) if its $G_f(x)$ minimizes the leading-order localized geometric objective, *i.e.,* $G_f(x) \in \arg\min_{G \succ 0} \{c \cdot \text{tr}(G) - \frac{\lambda}{2} \log \det(G)\}$.

Consider a manifold $\mathcal{M}$. Then we show the following (see Appendix A for a full proof).

**Theorem 1:** *Let $f_1$ and $f_2$ be two encoders locally optimal at point $x \in \mathcal{M}$ with parameters $\lambda_1$ and $\lambda_2$, respectively. For any $y \in \mathcal{M}$ with $d_\mathcal{M}(x, y) < \delta_\mathcal{M}(x)$:*

$$\|f_1(x) - f_1(y)\| = \kappa \cdot \|f_2(x) - f_2(y)\| + \mathcal{O}(d_\mathcal{M}(x, y)^2),$$

*where $\kappa = \sqrt{\lambda_1/\lambda_2}$.* $\square$

Theorem 1 is inherently local: it holds only for $y$ within neighborhood radius $\delta_\mathcal{M}(x)$, consistent with the empirical decorrelation at long distances in Fig. 1. Moreover, the same analysis extends to a relaxation of (A3) where the local augmentation scale is point-dependent, *i.e.,* $\mathbb{E}[vv^\top \mid x] = c(x)I_d$, yielding a region-dependent $\kappa$ (see Appendix A.3).

## 3. Encoder-Invariant Geometric Hashing

Theorem 1 establishes a local cross-model geometric consistency: under local optimality, two contrastive encoders preserve short-range distances up to a scale factor. We now translate this local property into a concrete *hashing framework* for vector linking. The goal is to construct, for each vector, a signature that is (approximately) encoder-invariant over the overlap $\Omega$, so that matching objects collide (or become nearest neighbors) in a shared hash space.

**Distance-to-anchor hash**. A *geometric view* of $\Omega$ across $\mathbf{E}_1$ and $\mathbf{E}_2$ is a set $\mathcal{A}$ of paired vectors $\{(a_1, a_1'), \ldots, (a_k, a_k')\} \subseteq \mathbf{E}_1 \times \mathbf{E}_2$, where each pair $(a_j, a_j')$ encodes the same overlap object in $\Omega$ and is referred to as a paired anchor. Fix a distance function $\text{dist}(\cdot, \cdot)$. Given a view $\mathcal{A}$, we define the *distance-to-anchor hash* of $u \in \mathbf{E}_1$ and $v \in \mathbf{E}_2$ *w.r.t.* $\mathcal{A}$ as: $\mathbf{r}_\mathcal{A}(u) := \big(\text{dist}(u, a_1), \ldots, \text{dist}(u, a_k)\big) \in \mathbb{R}^k$ and $\mathbf{r}_\mathcal{A}'(v) := \big(\text{dist}(v, a_1'), \ldots, \text{dist}(v, a_k')\big) \in \mathbb{R}^k$, respectively.

As Theorem 1 predicts an unknown scale factor between encoders, we compare hashes $\mathbf{r}_\mathcal{A}(u)$ and $\mathbf{r}_\mathcal{A}'(v)$ using a *scale-free* similarity. A simple choice is cosine similarity over normalization: $\text{sim}_\mathcal{A}(u, v) := \langle \widehat{\mathbf{r}}_\mathcal{A}(u), \widehat{\mathbf{r}}_\mathcal{A}'(v)\rangle$, where $\widehat{\mathbf{r}}_\mathcal{A}(u) := \frac{\mathbf{r}_\mathcal{A}(u)}{\|\mathbf{r}_\mathcal{A}(u)\|_2}$ and $\widehat{\mathbf{r}}_\mathcal{A}'(v) := \frac{\mathbf{r}_\mathcal{A}'(v)}{\|\mathbf{r}_\mathcal{A}'(v)\|_2}$.

**Locality $\Rightarrow$ encoder-invariant hashing**. Let $x \in \Omega$ have representations $u = f_1(x) \in \mathbf{E}_1$ and $v = f_2(x) \in \mathbf{E}_2$. Consider a view $\mathcal{A} = \{(a_1, a_1'), \ldots, (a_k, a_k')\}$ whose anchors correspond to overlap objects $x_1, \ldots, x_k \in \Omega$ (*i.e.,* $a_j = f_1(x_j)$ and $a_j' = f_2(x_j)$). If all anchors are short-range for $x$, *i.e.,* $d_\mathcal{M}(x, x_j) < \delta_\mathcal{M}(x)$ for all $j$, then applying Theorem 1 with $y = x_j$ yields the componentwise relation $\text{dist}(u, a_j) = \kappa \text{dist}(v, a_j') + O(d_\mathcal{M}(x, x_j)^2)$. Hence

$$\mathbf{r}_\mathcal{A}(u) = \kappa \mathbf{r}_\mathcal{A}'(v) + \boldsymbol{\epsilon}_\mathcal{A}(x), \|\boldsymbol{\epsilon}_\mathcal{A}(x)\| = O\big(\max_j d_\mathcal{M}(x, x_j)^2\big),$$

In the ideal case $\boldsymbol{\epsilon}_\mathcal{A}(x) = 0$, the two hashes are exactly related by a positive scalar, so after $\ell_2$ normalization they coincide and our scale-free similarity satisfies $\text{sim}_\mathcal{A}(u, v) = 1$. When anchors are sufficiently close, the second-order remainder is small, hence the normalized hashes remain close and $\text{sim}_\mathcal{A}(u, v)$ stays near 1. Therefore, distance-to-anchor hashing is approximately encoder-invariant in the short-range regime, providing a geometric basis for vector linking via hash-space nearest neighbors.

The locality requirement above is essential. As shown empirically in Section 2.1, long-range distances decorrelate across encoders. Therefore, if a view contains anchors that are far from the candidate vectors, those hash coordinates become dominated by model-specific distortion and can overwhelm the signal from locally consistent distances. This implies that global hashing with one fixed anchor set is unreliable for points that do not lie near that anchor set, which is precisely the typical case under partial, unknown overlap.

**Localizing hashes via multi-view voting**. While distance-to-anchor hashes are encoder-invariant when they are constructed from anchors in the short-range neighborhood, it is, however, nontrivial to decide what counts as short-range dis-

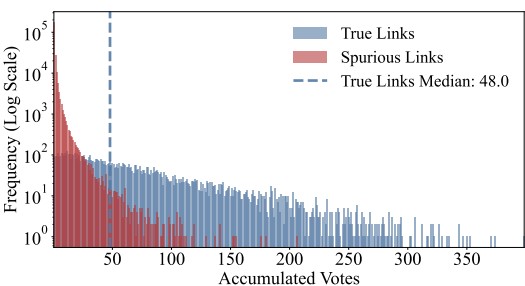

*Figure 2.* **Signal (true links)** Vs. **Noise (spurious links):** x-axis is accumulated votes for candidate links on ArguAna(`GTE` vs. `OpenAI`); y-axis reports pair frequency (logscaled).

tances, as the threshold $\delta_{\mathcal{M}}$ is unknown and data-dependent.

Rather than explicitly estimating the unknown locality threshold $\delta_{\mathcal{M}}$, we treat locality *statistically* by sampling many small, diverse geometric views. $\mathcal{A}^{(1)}, \ldots, \mathcal{A}^{(T)}$ from the current pool of paired anchors (via bootstrapping, as we will see in Section 4). Each view induces its own hash space and proposes candidate links by nearest-neighbor matching under $\text{sim}_{\mathcal{A}^{(t)}}(\cdot, \cdot)$; the proposed pairs are treated as votes. True links are supported across many views that happen to include locally relevant anchors, whereas spurious collisions caused by long-range, model-specific distortion are view-dependent and rarely accumulate. Since matching within each view uses a scale-free similarity, this aggregation remains effective even when the proportionality factor varies over the manifold, *e.g.,* under the relaxed isotropy model of (A3), *i.e.,* $\mathbb{E}[vv^{\top} \mid x] = c(x)I_d$ (Appendix A.3).

We tested this voting mechanism in a one-shot diagnostic on ArguAna encoded by `GTE` (Li et al., 2023) and `OpenAI` (OpenAI). We sampled a fixed pool of 500 ground-truth paired anchors from the overlap and drew 500 random views, each containing 30 anchors. In each view we computed hashes and collected voted links via hash collisions, then tallied for each candidate pair the number of views in which it is proposed. Fig. 2 shows the resulting vote histogram on a log scale. We observe a sharp separation: spurious links (red) follow a steep exponential decay, with the vast majority receiving negligible support. In contrast, true links (blue) exhibit a robust distribution with a median of 48 votes, demonstrating that stable local geometry consistency allows true links to survive across diverse views.

## 4. Bootstrapping Multi-View Hashing

Section 3 shows that distance-to-anchor hashing is reliable only when a view contains anchors that are locally relevant, and we need many such views to statistically form the localized hash via voting. We alleviate the high demand of anchors by an iterative bootstrapping framework that starts from a tiny seed set of paired anchors and grows it using multi-view hash collisions with posterior-guided promotion.

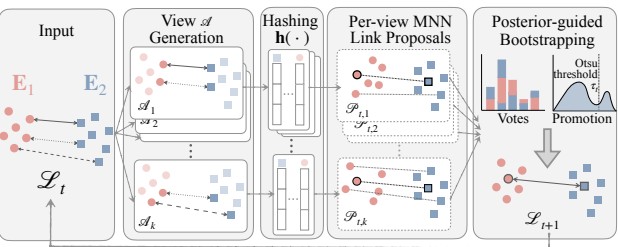

*Figure 3.* **The geometric embedding hashing** (GEH) **framework**

**Framework**. The framework, denoted by GEH (Geometric Embedding Hashing) and shown in Fig. 3, takes as input embedding clouds $\mathbf{E}_1$ and $\mathbf{E}_2$, and a tiny seed set of paired anchors $\mathcal{S} \subseteq \mathbf{E}_1 \times \mathbf{E}_2$ known via *e.g.,* domain knowledge, and outputs a set of inferred links $\mathcal{L}_T \subseteq \mathbf{E}_1 \times \mathbf{E}_2$.

Starting with $\mathcal{L}_0 := \mathcal{S}$, GEH generates $\mathcal{L}_T$ through iterations. At iteration $t$, it derives $\mathcal{L}_t$ by using inferred links identified at iteration $t-1$ in $\mathcal{L}_{t-1}$ as hash anchors, in three steps:

- *(View generation)* It samples $m_t$ geometric views $\mathcal{A}_{t,1}$, $\ldots, \mathcal{A}_{t,m_t}$ from $\mathcal{L}_{t-1}$, such that each view $\mathcal{A}_{t,k} \subseteq \mathcal{L}_{t-1}$ is a subset of paired anchors of fixed size $s_t$.

- *(Per-view link proposals)* For each view $\mathcal{A}_{t,k}$, it computes view-specific distance-to-anchor hashes and proposes a set of candidate links $\mathcal{P}_{t,k} \subseteq \mathbf{E}_1 \times \mathbf{E}_2$ by nearest-neighbor matching in the hash space.

- *(Posterior-guided bootstrapping)* It aggregates all proposals into a confidence score for each candidate link and promotes high-confidence links as new anchors in $\mathcal{L}_t$.

We next instantiate these three steps in full.

**View generation**. At iteration $t$, we draw $m_t$ views, $\mathcal{A}_{t,1}$, $\ldots, \mathcal{A}_{t,m_t} \subseteq \mathcal{L}_{t-1}$, each of size $s_t$. To stabilize early iterations, we include the seed set $\mathcal{S}$ in every view. This ensures views share a reliable core signal even when $\mathcal{L}_{t-1}$ is small.

*View sampling.* The quality of a view depends on anchor diversity: clustered anchors yield redundant hash coordinates and poor stability. Hence, we choose each view by greedy Furthest Point Sampling (FPS) (Gonzalez, 1985) on one side of the paired anchors. Let $\mathcal{B}_{t-1} := \mathcal{L}_{t-1} \setminus \mathcal{S}$, *i.e.,* anchors bootstrapped at previous iteration $t-1$. To form view $\mathcal{A}_{t,k}$, FPS starts from a random element of $\mathcal{B}_{t-1}$ and then iteratively adds the paired anchor whose (chosen-side) vector maximizes its minimum distance to the current subset. This encourages views with widely separated anchors and improves stability (see Appendix B.1 for detailed analysis).

*View scheduling.* As bootstrapping (iteration) progresses, the anchor pool $\mathcal{L}_{t-1}$ grows. We increase view diversity by sampling more views from $\mathcal{L}_{t-1}$ while reducing anchors per view. Let $g_t := \frac{|\mathcal{L}_{t-1}|}{|\mathcal{S}|}$. We define a scaling factor $\text{sf}_t := 1 + c \log g_t$ for some $c \geq 0$; we set the number of views at iteration $r$ to $m_t := \lceil m_0 \, \text{sf}_t \rceil$ and the size of each view to

$s_t := \lceil \rho_0 |\mathcal{L}_{t-1}|/\text{sf}_t \rceil$, with $\rho_0 \in (0, 1]$.

Intuitively, this increases the number of views as anchors grow, while making each view smaller so that it preferentially reflects local geometry. Further, this stabilizes the per-anchor coverage, *i.e.,* each anchor appears in roughly a constant number of views per iteration (see Appendix B.2).

**Per-view link proposal**. Given a view $\mathcal{A}_{t,k} = \{(a_1, a'_1), \ldots, (a_{s_t}, a'_{s_t})\}$, we construct a view-specific hash for each point and propose links by similarity search in hash space.

*Kernelized hashes*. Raw distance-to-anchor vectors can be dominated by far anchors where cross-model distances are least consistent. We therefore apply a monotone kernel that downweights large distances. For $u \in \mathbf{E}_1$ (analogously for $v \in \mathbf{E}_2$), we use kernelized hash $\big(\mathbf{h}_{t,k}(u)\big)_j := \exp\Big(-\frac{\text{dist}(u,a_j)}{\sigma_{t,k}}\Big)$, for $j = 1, \ldots, s_t$, where $\text{dist}(\cdot, \cdot)$ is the cosine distance between $\ell_2$-normalized embeddings and $\sigma_{t,k} > 0$ is a per-view bandwidth by median heuristic: the median of the nonzero pairwise $\ell_2$ distances between hashes within the view. This preserves the rank ordering induced by distances while emphasizing the short-range regime where Theorem 1 applies.

*Mutual nearest neighbor search in hash space*. We compare hash vectors by cosine similarity in $\mathbb{R}^{s_t}$. To mitigate hubness in nearest-neighbor search, we use CSLS (Lample et al., 2018) with parameter $k_{\text{CSLS}}$. Let $\text{csls}_{t,k}(u, v)$ denote the resulting view-specific similarity score. We then propose only mutual nearest neighbors (MNN): a pair $(u, v)$ is proposed if $v$ maximizes $\text{csls}_{t,k}(u, \cdot)$ and $u$ maximizes $\text{csls}_{t,k}(\cdot, v)$. The set of all proposed pairs is denoted $\mathcal{P}_{t,k}$. At multi-million scale, computing hashes and scoring $\text{csls}_{\mathcal{A}}$ for all points can be costly, so we restrict hash construction and MNN to a local $k_{\text{NN}}$ neighborhood around each view's anchors (Appendix B.3). This restriction is consistent with Section 2.1: we save cost on candidate pairs that are outside a view's local neighborhood which are less reliable and contribute little useful voting signal.

**Posterior-guided bootstrapping**. This step expands $\mathcal{L}_{t-1}$ from tiny seeds ($\mathcal{L}_0 = \mathcal{S}$) by promoting only links that are consistently supported across many views. This is exactly the signal/noise separation observed in Section 3.

*Vote counts*. Storing statistics for all $|\mathbf{E}_1| \cdot |\mathbf{E}_2|$ pairs is infeasible. We track only pairs that are proposed at least once. Define the candidate universe up to iteration $t$ by $\mathcal{C}_t := \bigcup_{r=1}^{t} \bigcup_{k=1}^{m_r} \mathcal{P}_{r,k}$. Each view contributes one binary vote for each $(u, v) \in \mathcal{C}_t$, denoted by $Y_{t,k}(u, v) := \mathbb{I}\big[(u, v) \in \mathcal{P}_{t,k}\big]$. Define the cumulative number of positive votes as $\nu_{(u,v),t} := \sum_{r=1}^{t} \sum_{k=1}^{m_r} Y_{r,k}(u, v)$, and the total number of views as $N_{\leq t} := \sum_{r=1}^{t} m_r$.

*Beta–Bernoulli posterior as link confidence score*. We model success probability of $(u, v) \in \mathcal{C}$ as $\theta_{(u,v)}$ with an un-

informative $\text{Beta}(1, 1)$ prior. By Beta–Bernoulli conjugacy,

$$\theta_{(u,v)} \mid \mathcal{Y}_{(u,v),0:t} \sim \text{Beta}\big(1 + \nu_{(u,v),t}, \ 1 + N_{\leq t} - \nu_{(u,v),t}\big),$$

where $\mathcal{Y}_{(u,v),0:t} := \{Y_{r,k}(u, v)\}_{r=0,\ldots,t;\, k=1,\ldots,m_r}$. We use the posterior mean $\hat{\theta}_{(u,v),t} = (1 + \nu_{(u,v),t})/(2 + N_{\leq t})$ as the link confidence score.

*Adaptive link promotion*. To identify links without tuning a fixed threshold, we compute an iteration-specific threshold $\tau_t$ using Otsu's rule (Otsu, 1979) applied to the histogram of $\{\hat{\theta}_{(u,v),t}\}_{(u,v) \in \mathcal{C}_t}$. We promote all pairs with $\hat{\theta}_{(u,v),t} \geq \tau_t$. Since each vector should match at most one target vector, we enforce one-to-one matching by greedily selecting non-conflicting promoted pairs in decreasing $\hat{\theta}_{(u,v),t}$.

As Otsu maximizes between-class variance, $\tau_t$ adapts to the typically bimodal separation between consistently supported links (high posterior) and transient collisions (low posterior), concentrating $\mathcal{L}_t$ on stable, consensus-supported links, while filtering out distortion-driven spurious links.

**Termination & complexity**. GEH stops when $\mathcal{L}_t$ stabilizes, *e.g.,* when no new or very few pairs are promoted for consecutive iterations. Per view, hash construction costs $O((|\mathbf{E}_1| + |\mathbf{E}_2|) s_t)$, and nearest-neighbor search is performed in $s_t$ dimensions; total cost scales with the number $\sum_{t=1}^{T} m_t$ of evaluated views. In practice $m_t$ is small (tens) and per-view cost is moderate. (See Appendix C.2 for details.)

For extremely large-scale cases, the local-neighborhood restriction bounds the hash construction to $O(s_t^2 k_{\text{NN}})$. The cost of identifying each view's local set is amortized by a one-time $k$-nearest-neighbor index over $E_1 \cup E_2$ of build cost $O\big(|E_1| + |E_2|\big)$, reused across all $T$ iterations and $\sum_t m_t$ views. (See Appendix B.3 for more details.)

## 5. Effectiveness

We evaluate the effectiveness of GEH for vector linking.

### 5.1. Experimental Setup

**Benchmarks**. We used 6 BEIR (Thakur et al., 2021) text retrieval benchmarks: NFCorpus, SciFact, ArguAna, SciDocs, FiQA, and FEVER(see Table 5 in Appendix C.1). Each benchmark provides a corpus $\mathcal{D}$ of documents and built-in query-answer pairs for retrieval performance evaluation.

**Vector linking setup**. Given a corpus $\mathcal{D}$, we constructed two partially overlapping corpora $\mathcal{D}_1, \mathcal{D}_2$ as follows. We sampled an overlap set $\Omega \subset \mathcal{D}$ and split the residual $\mathcal{D} \setminus \Omega$ uniformly at random into two disjoint sets $\mathcal{U}_1, \mathcal{U}_2$ and set $\mathcal{D}_1 := \Omega \cup \mathcal{U}_1$ and $\mathcal{D}_2 := \Omega \cup \mathcal{U}_2$. We controlled the overlap level via $\alpha := \frac{|\mathcal{D}_1 \cap \mathcal{D}_2|}{|\mathcal{D}_1 \cup \mathcal{D}_2|} = \frac{|\Omega|}{|\Omega| + |\mathcal{U}_1| + |\mathcal{U}_2|}$. We embedded $\mathcal{D}_1$ and $\mathcal{D}_2$ with encoders $f_1$ and $f_2$, respectively, to obtain the two embedding clouds: $\mathbf{E}_1 := f_1(\mathcal{D}_1)$ and $\mathbf{E}_2 := f_2(\mathcal{D}_2)$. We set the ground-truth correspondence set

*Table 1.* **Vector linking at overlap $\alpha$=0.3, seeds $|\mathcal{S}|$=15:** each cell shows **precision/recall/F1**(%); **bold** indicates best per column.

| Method | Qwen-OpenAI NFCorpus | GTE-OpenAI SciFact | GTE-Mistral ArguAna | Mistral-OpenAI SciDocs | Qwen-Kalm FiQA |
|---|---|---|---|---|---|
| Linear | 2.8/7.2/4.0 | 2.2/5.0/3.1 | 0.4/0.3/0.3 | 2.2/1.7/1.9 | 0.7/0.8/0.8 |
| CCA | 46.7/10.6/17.3 | 29.3/5.2/8.8 | 25.2/3.9/6.8 | 11.0/1.4/2.4 | 12.5/0.7/1.4 |
| MLP | 36.1/3.3/6.0 | 14.2/1.4/2.5 | 11.0/0.7/1.2 | 10.4/0.3/0.5 | 9.2/0.1/0.3 |
| RCSLS | 38.9/3.0/5.6 | 28.6/2.1/3.8 | 26.2/0.2/0.4 | 32.6/0.3/0.6 | 15.8/0.2/0.4 |
| Proc | 52.5/11.8/19.3 | 37.9/5.7/9.9 | 30.8/4.8/8.4 | 15.6/1.6/2.9 | 20.6/1.3/2.4 |
| UGW | 14.8/2.4/4.2 | 15.9/2.5/4.4 | 5.9/0.1/0.2 | 3.8/0.1/0.1 | 2.4/0.0/0.0 |
| AO | 22.5/22.5/22.5 | 5.6/5.6/5.6 | 0.6/0.6/0.6 | 0.2/0.2/0.2 | 0.1/0.1/0.1 |
| GEH | **82.1/95.6/88.3** | **83.2/89.1/86.0** | **77.1/84.5/80.7** | **82.8/81.6/82.2** | **79.8/79.9/79.8** |

$M^* := \{(f_1(x), f_2(x)) : x \in \Omega\}$. All methods were given only $\mathbf{E}_1, \mathbf{E}_2$ and were not told $\Omega$, nor had access to $f_i$ or $\mathcal{D}_i$.

By default, we drew a seed set $\mathcal{S} \subset M^*$ by uniformly sampling overlap items. We report results for three seed sizes $|\mathcal{S}| \in \{15, 20, 30\}$. We also evaluated out-of-domain (OOD) seeds which are drawn from a different dataset (Section 5.4). In each case, all methods received the same $\mathcal{S}$.

**Models**. We used 5 pairs of major embedding models: (a) `Mistral` (Mistral-embed (Jiang et al., 2023)) vs. `OpenAI` (Text-embedding-3-small (OpenAI)), (b) `GTE` (GTE-Qwen2-7B-instruct (Li et al., 2023)) vs. `Mistral`, (c) `GTE` vs. `OpenAI`, (d) `Qwen` (Qwen3-Embedding-8B (Zhang et al., 2025)) vs. `KaLM` (KaLM-Embedding-Gemma3-12B (Zhao et al., 2025)), and (e) `Qwen` vs. `OpenAI`.

**Baselines**. As there is no prior method that directly tackles vector linking with partial overlap, we adapt embedding alignment methods by incorporating ideas from GEH: they first align $\mathbf{E}_1$ to $\mathbf{E}_2$ by supervising on $\mathcal{S}$, yielding a shared embedding space with aligned $\mathbf{E}_1$ and $\mathbf{E}_2$; they then use CSLS based MNN search to identify links as GEH does for link proposal. Specifically, we trained 5 alignment methods:

- Linear: regression with MSE (Mikolov et al., 2013).
- CCA: canonical correlation analysis (Lu et al., 2015).
- MLP: two-layer MLP trained with cosine loss on seeds.
- RCSLS: RCSLS (Joulin et al., 2018), a retrieval-based linear mapping optimized via SGD.
- Proc: orthogonal Procrustes alignment on seeds (closed-form SVD) (Smith et al., 2017).
- UGW: unbalanced Gromov-Wasserstein (Séjourné et al., 2021), quadratic OT on intra-space distance matrices, warm-started from a seed-biased coupling.

We also tested AO (Anchor Optimization (Cannistraci et al., 2023)), which was given overlap size to adapt to the task.

**Metrics**. We measured the output of each method, *i.e.,* set of predicted links $M$, by (a) Precision $:= \frac{|M \cap M^*|}{|M|}$, (b) Recall $:= \frac{|M \cap M^*|}{|M^*|}$, and (c) F1 $:= \frac{2\,\text{Precision}\cdot\text{Recall}}{\text{Precision}+\text{Recall}}$. In tables, we report all the metrics, and bold indicates the best.

Further details can be found in Appendix C.1.

*Table 2.* **Scalability on FEVER** on `Mistral` $\leftrightarrow$ `OpenAI`, single A100 80 GB GPU, overlap $\alpha = 0.3$, $|\mathcal{S}| = 30$. **Bold** marks the best per column; runtime is end-to-end wall-clock seconds. (UGW cannot complete on FEVER so it is not reported.)

| Method | Precision (%) | Recall (%) | Runtime (s) |
|---|---|---|---|
| Linear | 4.13 | 0.00 | 4420 |
| CCA | 5.22 | 0.61 | **1613** |
| MLP | 1.93 | 0.01 | 4414 |
| RCSLS | 6.40 | 0.11 | 3348 |
| Proc | 9.03 | 0.73 | 4494 |
| GEH | **93.8** | **68.9** | 3328 |

### 5.2. Performance on Vector Linking

**Overall**. We first evaluated the performance of all methods for linking all 5 pairs of embedding models across all datasets except FEVER (reserved for scalability test). Table 1 summarizes their recall, precision, and F1 with only 15 seed anchors in $\mathcal{S}$ for an overlap ratio $\alpha = 30\%$ (see Tables 18-26 in Appendix C.3.1 for a complete report).

The results are very encouraging: our method GEH consistently outperforms all other methods across all cases by a substantial margin. For example, on FiQA, we need only 15 seeds to recover the entire overlap between `Qwen` and `KaLM` models, achieving 79.9% in recall, 79.8% in precision, and 79.8% in F1-score, respectively, while the second-best method achieves only 1.3%, 20.6%, and 2.4%. The results for linking across other model pairs and datasets are similar.

**Varying overlap and seeds**. We further evaluated the impact of overlap ratio $\alpha$ and number of seed anchors $|\mathcal{S}|$. The results over SciFact dataset for linking `Mistral` and `OpenAI` models are shown in Table 3 (see Appendix C.3.1 for more).

Our method, GEH, is particularly robust and stable to the seed anchors; for instance, with 15 seeds, it already performed as well as it did with 30 seeds, while all other methods required more seeds to improve performance. All methods performed better with larger overlap, but the large gap between GEH and others remained stable and significant.

**Scalability**. We also evaluated the scalability of GEH on FEVER (5.4 million corpus) using `Mistral` $\leftrightarrow$ `OpenAI` embeddings, fixing the overlap ratio to $\alpha = 0.3$, seed budget $|\mathcal{S}| = 30$. All methods are run on a single NVIDIA A100 (80 GB). For GEH, the per-view local-set restriction described in Section 4 is activated due to the size of FEVER.

Table 2 reports the precision, recall, and end-to-end wall-clock runtime on a single A100. GEH attains 93.8% precision and 68.9% recall in 3328 s. GEH remains within the same order of magnitude as the fastest baseline (CCA, $\sim 2.1\times$ slower) and is faster than all other baselines despite its iterative design. The gap is large: relative to CCA, GEH improves precision from 5.22% to 93.8% and recall from 0.61% to 68.9%, at only $\sim 2.1\times$ the wall-clock cost.

Note that CSLS+MNN link extraction is shared by all align-

*Table 3.* **Vector linking on SciFact (`Mistral`↔`OpenAI`):** each cell reports **precision/recall/F1** (%). Best values per metric are **bolded**.

| Method | Overlap .15 | | | Overlap .20 | | | Overlap .30 | | |
|--------|----------|----------|----------|----------|----------|----------|----------|----------|----------|
| | Seeds 15 | Seeds 20 | Seeds 30 | Seeds 15 | Seeds 20 | Seeds 30 | Seeds 15 | Seeds 20 | Seeds 30 |
| Linear | 4.2/4.4/4.3 | 6.8/7.8/7.3 | 11.0/13.3/12.1 | 4.2/4.5/4.3 | 7.2/8.1/7.6 | 10.6/16.5/12.9 | 2.9/5.2/3.7 | 6.1/11.5/8.0 | 10.9/23.6/14.9 |
| CCA | 16.7/5.5/8.3 | 21.3/11.5/14.9 | 34.4/29.2/31.6 | 16.8/4.9/7.6 | 28.9/10.0/14.9 | 42.0/26.2/32.3 | 26.3/4.9/8.2 | 33.4/10.0/15.4 | 53.7/26.4/35.4 |
| MLP | 15.4/1.6/2.8 | 19.4/4.2/6.9 | 26.0/11.2/15.7 | 15.1/1.9/3.3 | 22.7/5.2/8.5 | 35.8/13.2/19.2 | 21.7/1.3/2.5 | 29.0/3.8/6.7 | 40.3/11.8/18.3 |
| RCSLS | 46.2/2.3/4.3 | 39.9/4.5/8.1 | 39.7/12.0/18.4 | 47.9/1.8/3.5 | 39.4/3.2/5.9 | 44.3/9.7/15.9 | 43.3/1.4/2.7 | 39.6/2.8/5.2 | 53.3/9.9/16.6 |
| Proc | 17.2/4.7/7.4 | 26.3/12.0/16.5 | 39.8/29.2/33.7 | 23.4/6.8/10.5 | 35.8/11.5/17.4 | 48.3/27.0/34.6 | 29.3/5.3/8.9 | 41.6/10.5/16.8 | 62.3/29.6/40.1 |
| UGW | 3.8/0.4/0.7 | 3.7/0.4/0.7 | 3.6/0.4/0.7 | 1.1/0.1/0.2 | 1.1/0.1/0.2 | 1.1/0.1/0.2 | 4.7/0.3/0.5 | 4.7/0.3/0.5 | 3.5/0.2/0.4 |
| AO | 4.2/4.2/4.2 | 9.4/9.4/9.4 | 25.4/25.4/25.4 | 2.7/2.7/2.7 | 6.5/6.5/6.5 | 32.0/32.0/32.0 | 6.5/6.5/6.5 | 27.2/27.2/27.2 | 43.9/43.9/43.9 |
| GEH | **63.3/84.0/72.2** | **63.0/81.2/70.9** | **62.3/82.3/70.9** | **73.6/87.0/79.7** | **73.2/85.3/78.8** | **72.2/85.8/78.4** | **84.0/87.7/85.8** | **83.8/86.5/85.1** | **83.6/87.0/85.3** |

*Table 4.* **Ablation of** GEH **components** on SciDocs with `Mistral` VS. `OpenAI`, at $\alpha$=0.15 and seeds $|\mathcal{S}|$=15: GEH denotes the complete pipeline: FPS view sampling, Kernelized signature, adaptive view scheduling, multi-view posterior aggregation, and bootstrapping. Each subsequent row removes one component relative to GEH; the row − FPS & Kernel removes both per-view components jointly. Each cell reports the seed-level mean ± standard deviation (%) of the corresponding metric. A † marks cells where the seed-level coefficient of variation ($\sigma/\mu$) exceeds 10%.

| Variant | Precision (%) | Recall (%) | F1 (%) |
|---------|---------------|------------|--------|
| GEH | 62.1 ± 1.1 | 81.7 ± 0.7 | 70.5 ± 0.6 |
| − Kernel | 61.0 ± 8.3† | 52.9 ± 35.0† | 51.0 ± 33.3† |
| − FPS | 32.8 ± 26.7† | 16.3 ± 19.2† | 20.9 ± 23.3† |
| − FPS & Kernel | 28.3 ± 23.3† | 0.8 ± 0.5† | 1.4 ± 1.1† |
| − Adaptive schedule | 33.4 ± 0.6 | 61.8 ± 1.6 | 43.4 ± 0.9 |
| − Multi-view voting | 24.0 ± 2.6† | 39.4 ± 5.0† | 29.8 ± 3.4† |
| − Bootstrapping | 1.9 ± 0.6† | 2.1 ± 1.2† | 1.9 ± 0.8† |

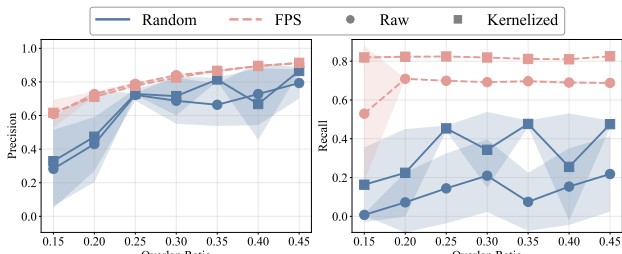

*Figure 4.* **Impact of view construction and distance encoding:** on SciDocs with `Mistral` VS. `OpenAI`, we compared the precision (left) and recall (right) of view strategies (FPS, Random), each with Kernelized or Raw distances. Shaded areas show variance (±1 std).

$$m_t=1 \text{ and } \mathcal{A}_{t,1}=\mathcal{L}_{t-1} \text{ (Section 3).}$$

- − **Bootstrapping**: run a single iteration on the $\mathcal{S}$; no anchor-pool growth (Section 4).

Table 4 reports the results. Bootstrapping and multi-view voting account for most of the absolute performance. Without bootstrapping, only the 15 seeds were available as anchors and GEH collapsed to 1.9/2.1/1.9 for Precision(%)/ Recall(%)/ F1 (%); without multi-view voting, single-view link proposals could not separate true links from distortion-driven collisions and performance fell to 24.0/39.4/29.8.

Removing FPS or the kernel signature makes performance highly variable (*e.g.*, recall $\sigma$ rose from 0.7 to 19–35): these two per-view components act as variance reducers, FPS by spreading anchors so each view stays well-conditioned and the kernel by suppressing the long-range distance regime that decorrelates across encoders (Section 2.1).

**View generation and hash encoding**. We further examined the view generation strategy and hash encoding in GEH. With 15 fixed seeds, we vary the overlap ratio from 0.15 to 0.45 and compare two view generation strategies, namely, *Random* (uniformly sampled) and *FPS* (furthest-point-sampled), each combined with two distance encodings: *Raw* (unprocessed distance dist), *Kernelized* ($\exp(-\text{dist}/\sigma)$).

Fig. 4 reports the precision and recall of vector linking on SciDocs with `Mistral` and `OpenAI`. Kernelized encoding consistently dominates the Raw variant for each view strategy, especially in recall, confirming that emphasizing short-range distances improves robustness. For view con-

ment baselines, so the incremental cost of GEH comes only from evaluating multiple views. Crucially, each view operates in the low-dimensional distance-to-anchor hash space induced by a small anchor set, rather than re-matching in the original embedding space. The iterative procedure thus performs many moderate-cost hash-space retrievals rather than repeated dense searches over full embedding clouds.

## 5.3. Ablation Study

**Staged ablation**. We ablate the five components of GEH on SciDocs (`Mistral` VS. `OpenAI`) with $\alpha$=0.15 and $|\mathcal{S}|$=15. For each variant we run 10 independent trials, each with a fresh random seed controlling both the draw of $\mathcal{S}$ and the internal randomness of view generation, and report the across-trial mean and standard deviation ($\mu \pm \sigma$).

Each variant is GEH but with the named component swapped for a simpler default. The replacement defaults are:

- − **Kernel**: use the raw distance vector $r_{\mathcal{A}}$ instead of the kernelized hash $h_{\mathcal{A}}$ (Section 4).

- − **FPS**: draw view anchors uniformly at random from $\mathcal{L}_{t-1}$ instead of by FPS (Section 4).

- − **FPS & Kernel**: both per-view defaults applied jointly (raw hashes plus random view draws).

- − **View scheduling**: freeze the schedule at iteration zero, $\rho_t \equiv \rho_0$ and $m_t \equiv m_0$ (Section 4).

- − **Multi-view voting**: use single-view MNN proposal,

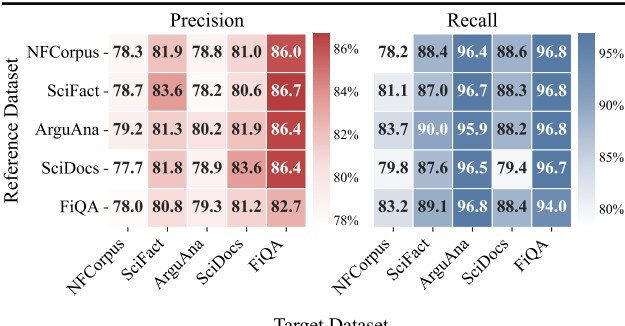

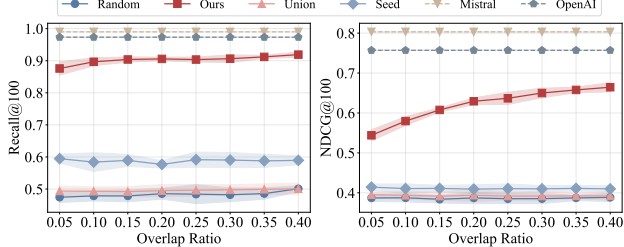

Figure 6. **Integrated vector database retrieval performance:** over SciFact, Recall@100 (left) and NDCG@100 (right) vs. overlap ratio $\alpha$, where the overlap contains *no benchmark answers*. `Mistral` and `OpenAI` are the theoretical upper limit of retrieval quality where we embed all objects with one single model.

*Figure 5.* **Out-of-domain anchor transfer:** Precision (left) and recall (right) of our method on five target datasets (columns) when supervised seeds are drawn from an out-of-domain reference dataset (rows). All runs use 30% overlap in the target and 30 OOD seeds.

structions, FPS always gives the better performance, particularly in recall, indicating that dispersed geometric views are preferable in the presence of cross-model distance distortion due to their better stability (see analysis in Appendix B.1).

### 5.4. Robustness to Domain Shifts (OOD Analysis)

In practice, exact vectors stored in a private index may be inaccessible; users can encode a small public corpus with both models and use those embeddings as references. To simulate this, we replace in-domain anchors with *out-of-domain (OOD)* anchors drawn from a different dataset.

For each target dataset, we construct `Mistral` and `OpenAI` embeddings, enforce a 30% overlap, and draw 30 OOD seeds from a separate reference dataset. Fig. 5 reports the precision and recall across all reference-target dataset pairs: precision remains in 77-87% range and recall mostly between 80-97%. Additional results across seed budgets and overlap ratios are given in Appendix C.3.2. This indicates that small OOD corpora are sufficient to serve as anchors for geometric hashing to link private embedding clouds.

## 6. Applications

Finally, we demonstrate applications of vector linking.

**Vector database integration**. We demonstrate its benefit for *vector database integration*, enabling unified retrieval across vector databases embedded by distinct encoders.

*Setup*. We used the integration protocol of (Yang et al., 2025), which learns clustered Procrustes over known paired anchors to transform one vector database and merge it with the target database for unified querying. Instead of assuming paired anchors are given, we used GEH to infer links across databases and then apply the integration protocol. Following (Yang et al., 2025), we evaluated the integrated database via the recall@100 and NDCG@100 of benchmark queries.

*Baselines*. We compared against (i) *Random*, random anchor pairing; (ii) *Seed*, mapping learned from seed anchor

pairs only; and (iii) *Union* retrieval without cross-space mapping (directly taking the union of two databases). As an optimal reference, we also used a single model to re-embed the full unioned corpus encoded by the two databases, where the retrieval performance is the theoretical upper limit.

*Results*. Using the split in Section 5 with answer-free overlap (Appendix D.1.1), we evaluate query performance of integrating `Mistral` and `OpenAI` databases, with overlap ratio $\alpha$ varying from 5% to 40% and 30 seeds. Figure 6 reports results over SciFact. Our method substantially outperforms all baselines on both Recall@100 and NDCG@100, with performance improving as overlap increases, approaching to the theoretical limit of using `Mistral` or `OpenAI` alone.

**Cross-model clustering**. We also demonstrate *cross-model clustering*. Using vector linking, we stitched the two embedding sets and run clustering detection to recover clusters spanning across datasets. Our method recovers consistent cross-embedding cluster assignments for 75–98% of overlapping objects, and achieves cluster quality within $\approx 1\%$ of the clusters obtained when all objects are embedded by a single encoder (see Appendix D for details).

## 7. Conclusion

We introduced vector linking, recovering cross-model correspondences from two black-box embedding clouds under partial, unknown overlap. Our core observation is that independently trained contrastive encoders exhibit *local* cross-model geometric consistency. This motivates encoder-invariant geometric hashing based on distance-to-anchor signatures, and we instantiate it with a multi-view iterative algorithm that bootstraps a large anchor pool from a tiny seed set that promotes short-range distances. Experiments across multiple benchmarks and model pairs show robust, high-accuracy linking and enable downstream tasks such as vector database integration and cross-model clustering.

Future work includes reducing seed assumptions, extending to multi-model linking, and studying when local consistency holds beyond the current contrastive surrogate.

## Impact Statement

This paper presents work whose goal is to advance the field of Machine Learning. There are many potential societal consequences of our work, none which we feel must be specifically highlighted here.

## Acknowledgements

We thank the anonymous ICML reviewers and area chair for their constructive feedback. This work is supported by RAEng RF\201920\19\319 and the Huawei-Edinburgh Joint Lab.

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

# A. Proofs and Additional Details of Section 2

## A.1. Localizing the alignment term $\varphi_{\text{align}}$ in $\mathcal{L}_\lambda(f)$

Fix an anchor point $x \in \mathcal{M}$. Let $\exp_x : T_x\mathcal{M} \to \mathcal{M}$ denote the Riemannian exponential map. For any positive view $x^+$ satisfying (A2), let $v := v(x, x^+) \in T_x\mathcal{M}$ be the geodesic displacement, so $x^+ = \exp_x(v)$ and $\|v\| = d_{\mathcal{M}}(x, x^+)$. We write $O(\|v\|^k)$ to denote a scalar remainder term $\epsilon_k(v)$ such that there exist constants $r_0 > 0$ and $C$ with $|\epsilon_k(v)| \le C\|v\|^k$ for all $\|v\| \le r_0$. We use $\|\cdot\|$ for the Euclidean norm on vectors and the corresponding induced operator norms on linear/bilinear maps, and $\langle\cdot, \cdot\rangle$ for the standard inner product on $\mathbb{R}^K$. For a differentiable map $h$ between finite-dimensional vector spaces, $Dh(u)$ denotes its differential (Jacobian as a linear map) at $u$.

**Lemma A:** *Under assumptions (A1)–(A2),*

$$\|f(x) - f(x^+)\|^2 = v^\top G_f(x)\, v + O(\|v\|^3). \qquad \qquad \square$$

**Proof:** Fix $x \in \mathcal{M}$. Define $g(u) := f(\exp_x(u))$ for $u$ in a neighborhood of $0 \in T_x\mathcal{M}$. By (A1) and the smoothness of $\exp_x$, the map $g$ is $C^2$ in a neighborhood of $0$. Hence, by multivariate Taylor's theorem, there exist constants $r_0 > 0$ and $C$ such that for all $u \in T_x\mathcal{M}$ with $\|u\| \le r_0$,

$$g(u) = g(0) + Dg(0)\, u + \rho(u), \qquad \|\rho(u)\| \le C\|u\|^2.$$

By the chain rule, $Dg(0) = Df(x) \circ D(\exp_x)(0)$. Since $D(\exp_x)(0) = \text{Id}_{T_x\mathcal{M}}$ (*e.g.*, (Lee, 2003)), we have $Dg(0) = Df(x)$. In an orthonormal basis of $T_x\mathcal{M}$, $Df(x)$ is represented by $J_f(x)$ and $G_f(x) = J_f(x)^\top J_f(x)$. Let $v = v(x, x^+)$ and assume $\|v\| \le r_0$. Since $x^+ = \exp_x(v)$, we have $f(x^+) - f(x) = g(v) - g(0) = J_f(x)\, v + \rho(v)$ with $\|\rho(v)\| \le C\|v\|^2$. Therefore, $\|f(x^+) - f(x)\|^2 = \|J_f(x)v + \rho(v)\|^2 = \|J_f(x)v\|^2 + 2\langle J_f(x)v, \rho(v)\rangle + \|\rho(v)\|^2$. The cross term satisfies

$$\left|2\langle J_f(x)v, \rho(v)\rangle\right| \le 2\|J_f(x)v\|\, \|\rho(v)\| \le 2\|J_f(x)\|\, \|v\| \cdot C\|v\|^2 = O(\|v\|^3),$$

and the last term satisfies $\|\rho(v)\|^2 \le C^2\|v\|^4 = O(\|v\|^3)$ as $\|v\| \to 0$. Hence $\|f(x^+) - f(x)\|^2 = \|J_f(x)v\|^2 + O(\|v\|^3) = v^\top J_f(x)^\top J_f(x)v + O(\|v\|^3) = v^\top G_f(x)\, v + O(\|v\|^3)$, which proves the lemma. $\qquad \square$

This immediately gives us a rewriting of the alignment term in the localized surrogate used in Section 2.2.

**Corollary B:** *Under (A1)–(A3),*

$$\mathbb{E}[\|f(x) - f(x^+)\|^2 \mid x] = c \cdot \text{tr}(G_f(x)) + O\big(\mathbb{E}[\|v\|^3 \mid x]\big).$$

$$\square$$

**Proof:** By Lemma A, there exist $r_0 > 0$, $C < \infty$, and a remainder $\epsilon(v)$ such that for all $\|v\| \le r_0$,

$$\|f(x) - f(x^+)\|^2 = v^\top G_f(x)v + \epsilon(v), \qquad |\epsilon(v)| \le C\|v\|^3.$$

Taking conditional expectation given $x$ and using linearity,

$$\mathbb{E}[\|f(x) - f(x^+)\|^2 \mid x] = \mathbb{E}[v^\top G_f(x)v \mid x] + \mathbb{E}[\epsilon(v) \mid x].$$

Moreover,

$$|\mathbb{E}[\epsilon(v) \mid x]| \le \mathbb{E}[|\epsilon(v)| \mid x] \le C\, \mathbb{E}[\|v\|^3 \mid x],$$

so $\mathbb{E}[\epsilon(v) \mid x] = O(\mathbb{E}[\|v\|^3 \mid x])$.

For the quadratic form, note $v^\top G_f(x)v = \text{tr}(G_f(x)vv^\top)$, hence

$$\mathbb{E}[v^\top G_f(x)v \mid x] = \text{tr}\big(G_f(x)\, \mathbb{E}[vv^\top \mid x]\big).$$

Under (A3), $\mathbb{E}[vv^\top \mid x] = cI_d$, so $\mathbb{E}[v^\top G_f(x)v \mid x] = c \cdot \text{tr}(G_f(x))$. Combining the above yields the claim. $\qquad \square$

We will also use a variant of Lemma A stated as follows:

**Corollary C:** *Under (A1), for $y$ in a sufficiently small normal neighborhood of $x$ and $v = \exp_x^{-1}(y)$, we have*

$$\|f(y) - f(x)\| = \sqrt{v^\top G_f(x)v} + O(\|v\|^2). \qquad \square$$

**Proof:** The proof of Lemma A is purely local and uses only that the second point lies in a normal neighborhood of $x$. Hence the same argument applies with $x^+$ replaced by $y$:

$$\|f(y) - f(x)\|^2 = v^\top G_f(x)v + O(\|v\|^3).$$

Write $a(v) := v^\top G_f(x)v$ and let $r(v) = O(\|v\|^3)$ denote the scalar remainder so that $\|f(y) - f(x)\|^2 = a(v) + r(v)$. Since $G_f(x) \succ 0$ by (A1), there exists $m > 0$ such that $a(v) \geq m\|v\|^2$ for all $v$. For sufficiently small $\|v\|$, $a(v) + r(v) \geq \frac{m}{2}\|v\|^2$. Hence, for some constants $C, c' > 0$,

$$\left|\sqrt{a(v) + r(v)} - \sqrt{a(v)}\right| = \frac{|r(v)|}{\sqrt{a(v) + r(v)} + \sqrt{a(v)}} \leq \frac{C\|v\|^3}{c'\|v\|} = O(\|v\|^2).$$

This completes the proof of Corollary C. $\qquad \square$

### A.2. Proof of Theorem 1

We give a full proof of Theorem 1. The intuition of the proof is that, for a nearby point $y$ around $x$, the encoder admits a first-order Taylor approximation along the unique short geodesic from $x$ to $y$. The leading term is governed by the Jacobian $J_f(x)$, hence by the induced metric $G_f(x) = J_f(x)^\top J_f(x)$. Local encoder optimality forces $G_f(x)$ to be a scalar multiple of the identity, which makes local distances proportional across encoders up to a scalar.

Consider any $x \in \mathcal{M}$ and let $y \in \mathcal{M}$ satisfy $d_{\mathcal{M}}(x, y) < \delta_{\mathcal{M}}(x)$. Let $\exp_x : T_x\mathcal{M} \to \mathcal{M}$ denote the Riemannian exponential map. Since we are within the normal neighborhood of $x$, there exists a unique $v \in T_x\mathcal{M}$ such that $y = \exp_x(v)$ and $\|v\| = d_{\mathcal{M}}(x, y)$ ((Lee, 2003)). Throughout, all $\mathcal{O}(\cdot)$ terms are as $y \to x$ (equivalently $\|v\| \to 0$).

*(1) local metric minimization.* For $i \in \{1, 2\}$, local optimality at $x$ means that $G_{f_i}(x)$ minimizes

$$\Phi_i(G) := c\,\mathrm{tr}(G) - \frac{\lambda_i}{2}\log\det(G) \qquad \text{over } G \succ 0.$$

Note that $-\log\det(G)$ is strictly convex, so $\Phi_i$ is strictly convex and thus has a unique minimizer. For any symmetric direction $H$, since $\frac{d}{dt}\mathrm{tr}(G + tH)\big|_{t=0} = \mathrm{tr}(H)$ and $\frac{d}{dt}\log\det(G + tH)\big|_{t=0} = \mathrm{tr}(G^{-1}H)$ (Jacobi's formula; *e.g.,* (Petersen et al., 2008)), we have

$$\frac{d}{dt}\Phi_i(G + tH)\Big|_{t=0} = \mathrm{tr}\left(\left[cI_d - \frac{\lambda_i}{2}G^{-1}\right]H\right).$$

At the minimizer this derivative is $0$ for all symmetric $H$. Since $\mathrm{tr}(AH) = 0$ for all symmetric $H$ implies $A = 0$ (take $H = A$), we obtain $cI_d - \frac{\lambda_i}{2}G^{-1} = 0$. Therefore,

$$G_{f_i}(x) = \frac{\lambda_i}{2c}I_d, \qquad i \in \{1, 2\}.$$

*(2) Local distance expansion via the induced metric.* Note that from Corollary C, for $i \in \{1, 2\}$

$$\|f_i(y) - f_i(x)\| = \|J_{f_i}(x)v\| + \mathcal{O}(\|v\|^2) = \sqrt{v^\top G_{f_i}(x)v} + \mathcal{O}(\|v\|^2).$$

Substituting $G_{f_i}(x) = \frac{\lambda_i}{2c}I_d$ (from step (1)) and $\|v\| = d_{\mathcal{M}}(x, y)$, we have

$$\|f_i(y) - f_i(x)\| = \sqrt{\frac{\lambda_i}{2c}}\,d_{\mathcal{M}}(x, y) + \mathcal{O}(d_{\mathcal{M}}(x, y)^2).$$

*Step 3: Compare encoders.* Let $\kappa := \sqrt{\lambda_1/\lambda_2}$. Then

$$\kappa \, \|f_2(y) - f_2(x)\| = \sqrt{\frac{\lambda_1}{2c}} \, d_{\mathcal{M}}(x, y) + \mathcal{O}(d_{\mathcal{M}}(x, y)^2).$$

Thus, by comparing with the $i = 1$ expansion, we have

$$\|f_1(y) - f_1(x)\| = \kappa \, \|f_2(y) - f_2(x)\| + \mathcal{O}(d_{\mathcal{M}}(x, y)^2),$$

which is equivalent to the stated form $\|f_1(x) - f_1(y)\| = \kappa \, \|f_2(x) - f_2(y)\| + \mathcal{O}(d_{\mathcal{M}}(x, y)^2)$.

### A.3. Relaxing (A3 → A3′): Point-Dependent Local Augmentation Assumption

Assumption (A3) in Section 2 simplifies the positive-pair distribution by $\mathbb{E}[vv^\top \mid x] = cI_d$ with a constant for all $x \in \mathcal{M}$. In practice, the magnitude of a semantic-preserving augmentation may also depend on the anchor point (*e.g.,* some examples admit larger perturbations than others). A natural relaxation is to allow the isotropic scale to vary with $x$.

**A relaxed local-isotropy model (A3′).** We replace (A3) of Section 2.2 by the following point-dependent variant, denoted by (A3′): Fix $x \in \mathcal{M}$ and let $v = v(x, x^+) \in T_x\mathcal{M}$ denote the geodesic displacement to a positive view. Assume $\mathbb{E}[v \mid x] = 0$ and $\mathbb{E}[vv^\top \mid x] = c(x) \, I_d$ for some function $c(\cdot) : \mathcal{M} \to (0, \infty)$.

*Effect on localized alignment term* $\varphi_{\text{align}}$. By Lemma A and taking conditional expectation, we have

$$\mathbb{E}[\|f(x) - f(x^+)\|^2 \mid x] = \mathbb{E}[v^\top G_f(x)v \mid x] + O\big(\mathbb{E}[\|v\|^3 \mid x]\big).$$

Using $v^\top G_f(x)v = \text{tr}(G_f(x)vv^\top)$ yields $\mathbb{E}[\|f(x) - f(x^+)\|^2 \mid x] = \text{tr}\big(G_f(x) \, \mathbb{E}[vv^\top \mid x]\big) + O\big(\mathbb{E}[\|v\|^3 \mid x]\big)$.

Under (A3′), $\mathbb{E}[vv^\top \mid x] = c(x)I_d$, hence

$$\mathbb{E}[\|f(x) - f(x^+)\|^2 \mid x] = c(x) \, \text{tr}(G_f(x)) + O\big(\mathbb{E}[\|v\|^3 \mid x]\big).$$

Therefore the leading-order localized objective becomes

$$\widetilde{\mathcal{L}}_\lambda(x; f) = c(x) \, \text{tr}(G_f(x)) - \frac{\lambda}{2} \log \det(G_f(x)).$$

*Local optimum*. Since the leading-order local objective depends on $f$ only through $G_f(x)$, we minimize over $G \succ 0$ to characterize the optimal local metric; we then call $f$ locally optimal at $x$ if its induced metric matches this minimizer. Minimizing over $G$ yields the first-order condition

$$c(x) \, I_d - \frac{\lambda}{2} G^{-1} = 0,$$

so the unique minimizer is $G_f^\star(x) = \frac{\lambda}{2c(x)} \, I_d$.

Thus the encoder remains locally a scaled isometry on $T_x\mathcal{M}$, but the scale factor depends on $x$ through $c(x)$.

**Implication on Theorem 1**. If two encoders $f_1$, $f_2$ satisfy the same analysis but potentially with different augmentation scales $c_1(x), c_2(x)$ and parameters $\lambda_1, \lambda_2$, then the local distance expansions become (for $i \in \{1, 2\}$):

$$\|f_i(x) - f_i(y)\| = \sqrt{\frac{\lambda_i}{2c_i(x)}} \, d_{\mathcal{M}}(x, y) + O\big(d_{\mathcal{M}}(x, y)^2\big)$$

for $y$ in the normal neighborhood of $x$. Eliminating $d_{\mathcal{M}}(x, y)$ gives a point-dependent scaling relation

$$\|f_1(x) - f_1(y)\| = \kappa(x) \, \|f_2(x) - f_2(y)\| + O\big(d_{\mathcal{M}}(x, y)^2\big)$$

with $\kappa(x) = \sqrt{\dfrac{\lambda_1 \, c_2(x)}{\lambda_2 \, c_1(x)}}$. In particular, if the two encoders share the same local positive-pair distribution in the sense that

$c_1(x) = c_2(x)$ for all $x$, then $\kappa(x)$ reduces to the constant $\sqrt{\lambda_1/\lambda_2}$ as stated in Theorem 1.

_Remark._ Allowing $c = c(x)$ provides a simple mechanism for why a *single global* scale does not fit all points: even when encoders are locally conformal, they may "expand" or "contract" neighborhoods by different amounts at different anchors. This complements the empirical observation that short-range distances are substantially more consistent than long-range distances, while also explaining residual variability within the short-range regime.

### A.4. Correlation Analysis

We report the Pearson correlation coefficient $\rho$ between pairwise Euclidean distances measured in the reference embedding space and the corresponding distances between the same item pairs in the target space. Besides the contrastive encoder pairs, we also include classic word embedding models `GloVe` (Pennington et al., 2014) and `fastText` (Bojanowski et al., 2016) trained without a contrastive objective as non-contrastive baselines. The same two models are reused as the non-contrastive baseline in the top-$k$ Jaccard analysis of §A.5.

We probe the local-consistency / global-decorrelation pattern of Fig. 1 along four axes: *(i)* encoder pair: six contrastive encoder pairs (Fig. 7, panels a–f); *(ii)* target dimensionality: with `Mistral` as the reference, varying the target embedding dimensionality of `OpenAI` on SciFact (Fig. 7, panel g); *(iii)* task domain: `Mistral`→`OpenAI` on two clustering benchmarks (Fig. 7, panel h); *(iv)* encoder family: contrastive vs. non-contrastive on four retrieval datasets (Fig. 7, panels i–l), where each panel overlays `GloVe`↔`fastText` with `Mistral`↔`OpenAI`. For (i)–(iii), $\rho$ is high at short range and decays as the reference distance grows, indicating that the geometric signal underlying GEH is a property of the contrastive encoder family. In contrast, the non-contrastive pair in (iv) already starts at markedly lower $\rho$ at short range, and its long-range tail does not always decay. This supports our restriction of GEH's analysis to contrastive encoders (cf. §2.1).

### A.5. Retrieval Result analysis

To empirically validate the local geometric consistency of embedding spaces, we analyze the consistency of top-$k$ retrieval results across different embedding models. Given two embedding sets $\mathbf{E}_1$ and $\mathbf{E}_2$ encoding the same corpus, we perform top-$k$ nearest neighbor retrieval for each query point in both spaces and measure their agreement using the Jaccard index: $J_k = |\mathcal{N}_k^{(1)} \cap \mathcal{N}_k^{(2)}|/|\mathcal{N}_k^{(1)} \cup \mathcal{N}_k^{(2)}|$, where $\mathcal{N}_k^{(i)}$ denotes the set of $k$ nearest neighbors in embedding space $i$. We evaluate this metric across multiple embedding model pairs.

As shown in Figure 8, the behavior splits sharply by encoder family. For pairs of contrastive encoders (panels a–c), the Jaccard index starts at $J \approx 0.7$–$0.8$ at $k=1$ and decays monotonically to a plateau around $0.37$–$0.45$ at $k=50$, empirically confirming Theorem 1. In contrast, when at least one side is a non-contrastive encoder (panels d–f), the Jaccard index never enters this short-range high-consistency regime: `GloVe` ↔ `fastText` (panel d) stays near $J \approx 0.1$–$0.17$ over the entire range of $k$, and a `Mistral`/ `OpenAI` query side that yields $J \approx 0.8$ against a contrastive target collapses to $J \approx 0.2$ against a non-contrastive target (panels e–f).

## B. Details of Section 4

### B.1. Why FPS for View Sampling (Stability Analysis)

We justify FPS view sampling for the kernelized distance-to-anchor hash used in the main text. In a nutshell, we show that a hash is stable if the anchor-induced distance coordinates provide diverse directional information, and clustered anchors yield redundant coordinates and amplify cross-model distance distortions.

**Local stability via the tangent jacobian** Since we employ cosine distance, we operate on the unit hypersphere $\mathbb{S}^{D-1} \subset \mathbb{R}^D$. The hashing function $\mathbf{h}_{\mathcal{A}} : \mathbb{S}^{D-1} \to \mathbb{R}^{m_t}$ can be defined by the kernel with cosine distance:

$$h_{\mathcal{A}}(w)_j = \exp\left(-\frac{1 - \langle w, a'_j \rangle}{\sigma}\right), \quad j = 1, \dots, m_t.$$

We differentiate $h_{\mathcal{A}}(w)_j$ with respect to $w$ and restrict the domain to the tangent space of the sphere, $T_v\mathbb{S}^{D-1} = \{z \in \mathbb{R}^D : \langle z, v \rangle = 0\}$. The $j$-th row of the Jacobian $J_{\mathcal{A}}(v) \in \mathbb{R}^{m_t \times D}$ is given by the projection of the gradient onto $T_v$:

$$(J_{\mathcal{A}}(v))_{j,:} = \frac{1}{\sigma} h_j(v) \cdot \underbrace{(a'_j - \langle a'_j, v \rangle v)^{\top}}_{\mathbf{p}_j^{\top}}.$$

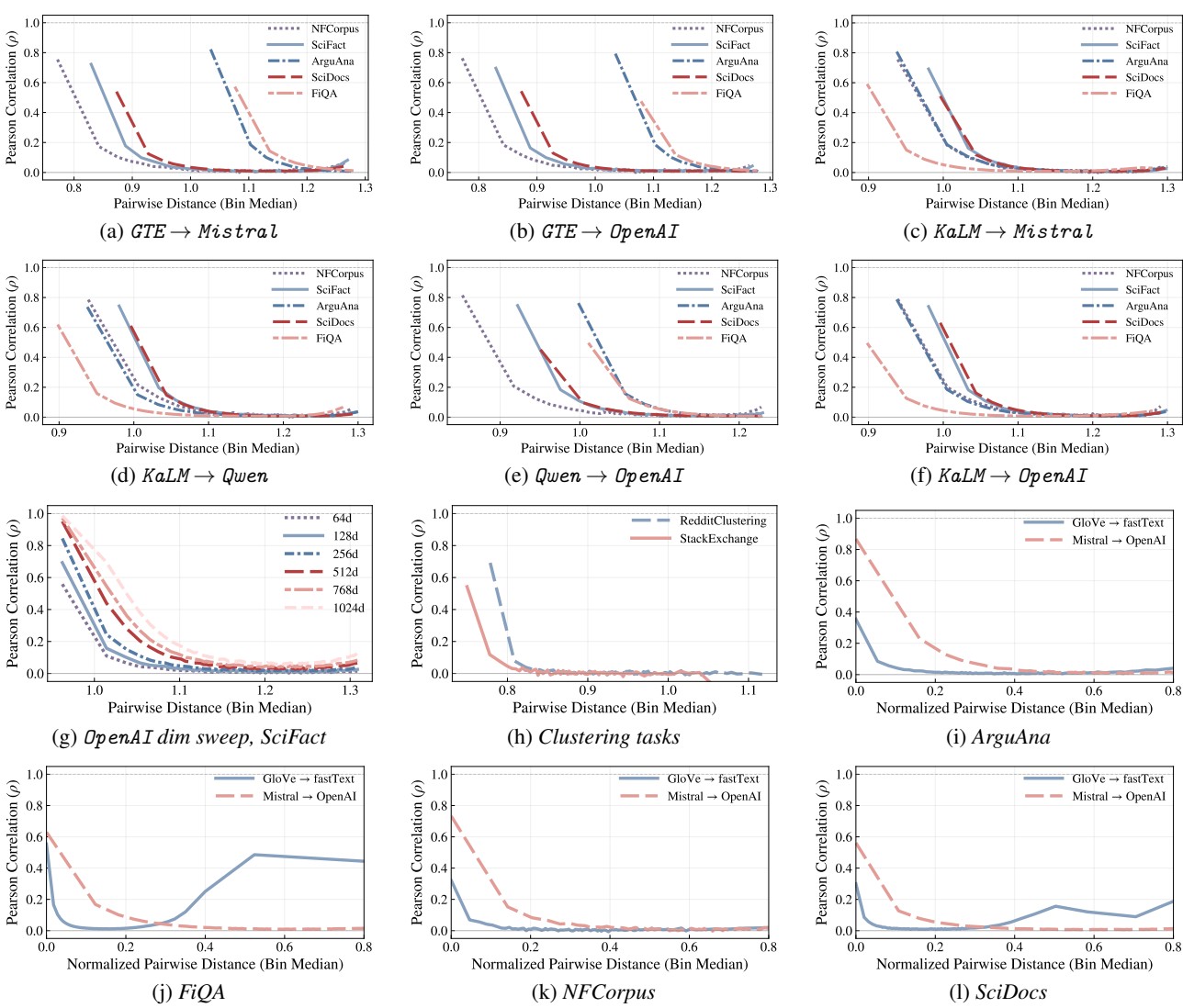

*Figure 7.* **Distance consistency across embedding spaces.** Each subplot shows Pearson correlation $\rho$ between pairwise distances in the reference space and their counterparts in the target space, binned by the reference distance. (a–f) Six contrastive encoder pairs. (g) `Mistral`→`OpenAI` on SciFact for a sweep of `OpenAI` dimensionalities. (h) `Mistral`→`OpenAI` on two clustering benchmarks. (i–l) Non-contrastive comparison: each panel overlays `GloVe`↔`fastText` with `Mistral`↔`OpenAI` on the same dataset.

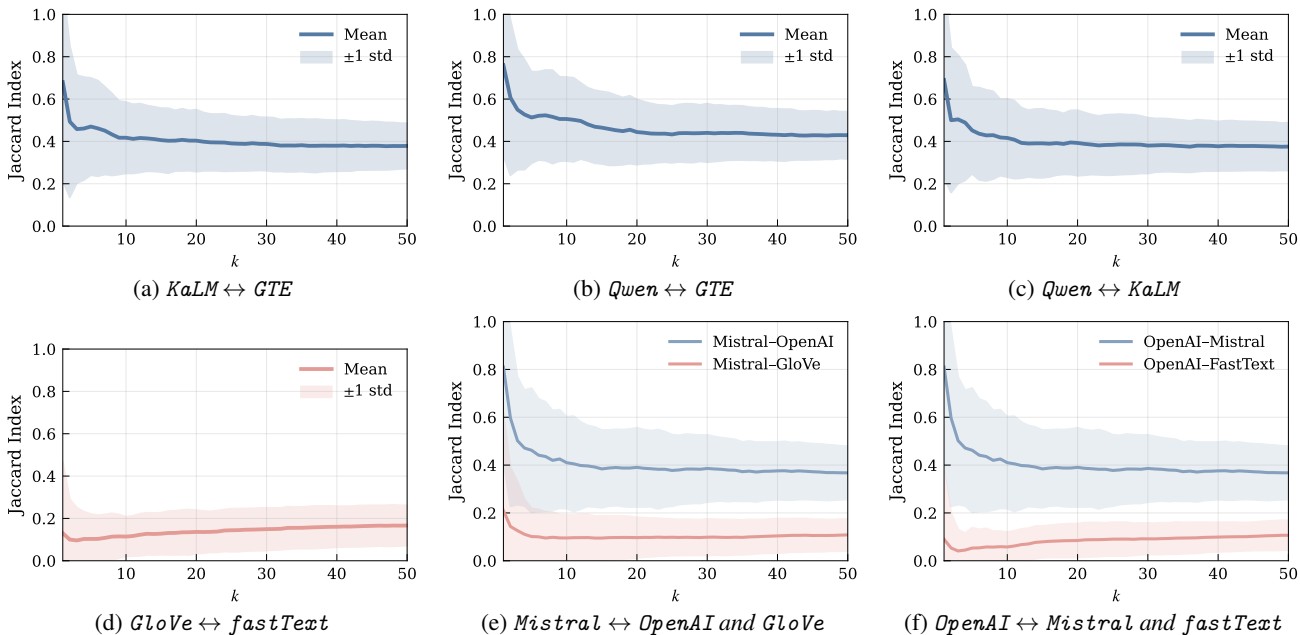

*Figure 8.* **Cross-embedding retrieval consistency analysis:**SciFact , Each panel reports the mean $\pm$ 1 std of the per-query Jaccard index between top-$k$ retrieval results from two embedding spaces over 100 random queries.

Here, $\mathbf{p}_j$ represents the component of the anchor $a'_j$ orthogonal to the query $v$.

We assume the data lies on a submanifold $\mathcal{M} \subset \mathbb{S}^{D-1}$ of intrinsic dimension $d \ll D$. For the hash to be stable (locally injective) on $\mathcal{M}$, the mapping must distinguish perturbations in any tangent direction.Let $\mathbf{\Pi}_v \in \mathbb{R}^{D \times d}$ be an orthonormal basis for the tangent space $T_v\mathcal{M}$. The Restricted Jacobian $\mathbf{J}_{\mathcal{M}}(v) \in \mathbb{R}^{m_t \times d}$ is defined as:

$$\mathbf{J}_{\mathcal{M}}(v) = J_{\mathcal{A}}(v) \cdot \mathbf{\Pi}_v.$$

A necessary condition for stability is that $\mathbf{J}_{\mathcal{M}}(v)$ has full column rank, which requires $m_t \geq d$.The robustness of this stability is quantified by the condition number $\kappa = \sigma_{\max}/\sigma_{\min}$ of $\mathbf{J}_{\mathcal{M}}(v)$. If $\kappa$ is large ($\sigma_{\min} \approx 0$), the hash is insensitive to changes along the corresponding singular vector, leading to ambiguity.

**Geometric Optimality of FPS** The singular values of $\mathbf{J}_{\mathcal{M}}(v)$ are determined by the geometric arrangement of the anchor projections $\{\mathbf{p}_j\}$. Random sampling can select multiple anchors that are locally redundant, making the manifold-tangent components $\{\Pi_v^\top p_j\}$ nearly collinear and leaving some tangent directions weakly sensed. This drives $\sigma_{\min} \to 0$. FPS greedily maximizes minimum pairwise distance, discouraging near-duplicates in each view and promotes larger $\sigma_{\min}$. This reduce angular redundancy among $\{\Pi_v^\top p_j\}$ and thus improve the conditioning of $J_{\mathcal{M}}(v)$ in practice.

Additionally, FPS acts as a covering strategy (a greedy $k$-center heuristic): selected anchors are spread across the current anchor pool, increasing the chance that a sampled view contains anchors that are *locally relevant* for many different query points.

FPS effectively reduce view-specific collisions and make per-view hash-space matching more stable, which in turn improves the quality of votes aggregated by the bootstrapping procedure in Section 4. Consistent with this analysis, FPS significantly outperforms random-based anchor selection in our ablations (Section 5.3).

### B.2. Properties of View Scheduling

This appendix expands on the scheduling rule used in Section 4 for choosing the number of views $m_t$ and the anchors per view $s_t$ as the paired-anchor pool $\mathcal{L}_{t-1}$ grows. The schedule provides two practical properties/guarantees: (i) increasing view diversity over iterations, and (ii) stable per-anchor coverage and computation.

**Objectives.** At iteration $t$, let $n_t := |\mathcal{L}_{t-1}|$ denote the size of the current paired-anchor pool. A view contains $s_t$ paired anchors, and we sample $m_t$ views. Define the *per-view anchor fraction* $\rho_t := s_t/n_t$.

The view schedule decides $m_t$ and $\rho_t$ (equivalently, $s_t$), while aiming to satisfy two practical objectives:

*(O1) Increasing diversity & locality.* As $n_t$ grows, we would like to sample more views (so more chances to hit short-range neighborhoods) while making each view a smaller fraction of the pool $\mathcal{L}_{t-1}$ (so views are more "local" and less dominated by far anchors).

*(O2) Stable per-anchor coverage and computation.* If views were sampled uniformly, a given anchor would appear in an expected $m_t \rho_t$ views per iteration. We would like this quantity to remain roughly stable over iterations (so anchors are neither over-used nor ignored), and we would like the total anchor usage $m_t s_t$ per iteration to scale reasonably.

**Anchor pool growth ratio $g_t$.** We measure progress by growth ratio $g_t := \frac{|\mathcal{L}_{t-1}|}{\max\{|\mathcal{S}|,1\}} \geq 1$, where $\mathcal{S}$ is the (tiny) initial seed set. This ratio ensures that the schedule depends on relative growth of the anchor pool rather than its absolute size, *e.g.,* increasing from 10 to 100 anchors and from 100 to 1000 anchors both correspond to $g_t = 10$.

**Coverage-preserving parameterization.** To satisfy (O2), we parameterize the schedule via a single scaling function $\mathrm{sf}(g) \geq 1$ that is set to satisfy the following:

$$m_t \approx m_0 \, \mathrm{sf}(g_t), \qquad \rho_t \approx \rho_0/\mathrm{sf}(g_t),$$

where $m_0 \in \mathbb{N}$ and $\rho_0 \in (0,1]$ are base parameters. Using this view scheduling would assure approximate invariance (ignoring rounding) $m_t \rho_t \approx m_0 \rho_0$, *i.e., a constant expected per-anchor participation rate* under uniform sampling.

This would also imply linear scaling of total anchor usage since $m_t s_t = m_t \rho_t n_t \approx m_0 \rho_0 |\mathcal{L}_{t-1}|$. Thus it increases the number of views while controlling per-iteration work.

**Determining** $\mathrm{sf}(g)$. For convenience, we consider $\mathrm{sf}(g) = 1 + u(g)$, where we use $u(g)$ to represent the increment induced by the growth of anchor pool. Thus we require $u(1) = 0$ as $g = 1$ corresponds to the initial state $|\mathcal{L}_{t-1}| = |\mathcal{S}|$ with no growth occurred.

To decide $u(g)$, we consider a simple principle: increment in schedule should depend only on the expansion rate of the anchors not the current scale $g$. Let $g$ be the current growth ratio *w.r.t.* $\mathcal{L}_{r-1}$ and let $g'$ denote the growth ratio after expansion to $\mathcal{L}_r$. Define the expansion ratio $\lambda := g'/g$. Then this principle says, if the anchor pool grows by a factor $\lambda \geq 1$, we want the increase in schedule (size and number of views) to depend only on $\lambda$ and not on the current growth scale $g$. In terms of $u(g) = \mathrm{sf}(g) - 1$, this boils down to $u(\lambda g) - u(g) = u(\lambda)$ for $\forall g, \lambda \geq 1$; equivalently,

$$u(\lambda g) = u(\lambda) + u(g) \qquad \forall g, \lambda \geq 1.$$

By the classical Cauchy additive functional equation (after a log change of variables), assuming $u$ is continuous, the unique solutions to this are of form $u(g) = c \log g$ for some constant $c \geq 0$.

Thus we adopt the logarithmic scale factor

$$\mathrm{sf}_t := \mathrm{sf}(g_t) = 1 + c \log g_t.$$

This grows sublinearly and avoids exploding $T_t$ as the anchor pool expands, while still increasing view diversity. We empirically verify the robustness of view-scheduling hyperparameters $m_0$ and $\rho_0$ (see Appendix C.2.4).

## B.3. Per-view Link Proposal at Scale

When $\max(|\mathbf{E}_1|, |\mathbf{E}_2|) \leq \tau$ ($\tau = 5 \times 10^5$), GEH runs the global path of § 4. Otherwise, at iteration $t$, we replace FPS with a $k$-means partition of $\mathcal{L}_{t-1}$ in $E_1$ to ensure diversity of views. The number of partitions is $m_t = \lceil |\mathcal{L}_{t-1}|/d_{\max} \rceil$, where $d_{\max} = \max(d_{E_1}, d_{E_2})$, which ensures each view has sufficient anchors for a dimensionally well-posed local neighborhood. Each anchor in $\mathcal{L}_{t-1}$ is included in its $\rho$ nearest clusters ($\rho = 2$ in our experiments) which matches with the constant per-anchor participation rate in § B.2.

Within each view $\mathcal{A}_{t,k}$, each paired anchor $(a, a') \in \mathcal{A}_{t,k}$ contributes its own local neighborhood: let $\mathrm{NN}_{k_{\mathrm{NN}}}(a, E_1)$ denote the $k_{\mathrm{NN}}$ nearest ambient neighbors of $a$ in $E_1$, and analogously $\mathrm{NN}_{k_{\mathrm{NN}}}(a', E_2)$. The view's local sets are the unions $S_{\mathcal{A}_{t,k}}^{(1)} := \bigcup_{(a,a') \in \mathcal{A}_{t,k}} \mathrm{NN}_{k_{\mathrm{NN}}}(a, E_1)$ and analogously $S_{\mathcal{A}_{t,k}}^{(2)}$. We then build the distance-to-anchor signatures of §3 only for points in $S_{\mathcal{A}_{t,k}}^{(1)} \cup S_{\mathcal{A}_{t,k}}^{(2)}$, compute $\mathrm{csls}_{\mathcal{A}_{t,k}}$ within $S_{\mathcal{A}_{t,k}}^{(1)} \times S_{\mathcal{A}_{t,k}}^{(2)}$, and emit MNN proposals $\mathcal{P}_{t,k}$ within that local bipartite set.

The local sets are identified via two $k$-NN indices over $E_1$ and $E_2$ separately, built once at the start of GEH (cost $O(|E_1| + |E_2|)$) and reused across all $T$ iterations and $\sum_t m_t$ views. Each per-view local-set lookup is then a single $k_{\mathrm{NN}}$-NN query against this static index.

*Table 5.* **BEIR dataset statistics:** #test queries ($Q$), test corpus size ($|C|$), and avg. relevant docs/query ($R$).

| Dataset | $Q$ | $|C|$ | $R$ |
|---|---|---|---|
| SciFact | 300 | 5,183 | 1.1 |
| NFCorpus | 323 | 3,633 | 38.2 |
| ArguAna | 1,406 | 8,674 | 1.0 |
| SciDocs | 1,000 | 25,657 | 4.9 |
| FiQA | 648 | 57,638 | 2.6 |
| FEVER | 6,666 | 5,416,568 | 1.2 |

# C. Details of Section 5

## C.1. Experimental Setup

### C.1.1. DATASETS

We conduct main experiments on five benchmark datasets from BEIR (Thakur et al., 2021), covering biomedical retrieval, financial analysis, citation prediction, argument mining, and fact verification. Below we briefly describe each dataset.

SciFact (Wadden et al., 2020) is a scientific fact-checking benchmark. Queries consist of short scientific claims, while documents are abstracts of scientific papers. The goal is to retrieve supporting or refuting evidence for each claim.

NFCorpus (Boteva et al., 2016) focuses on health-related information retrieval. Queries come from user-generated content such as blog posts, Q&A threads, and video transcripts, and the corpus is built from medical articles in PubMed.

ArguAna (Wachsmuth et al., 2018) addresses argument retrieval. Given an argument as a query, the task is to identify the most relevant counterarguments from a collection of argument pairs mined from online debate portals.

SciDocs (Cohan et al., 2020) is a citation prediction dataset derived from scientific publications. Queries are scientific papers, and the task is to retrieve related works among a large held-out collection.

FiQA (Maia et al., 2018) comes from the financial domain. The queries are investment-related questions posted on StackExchange, while the corpus contains financial articles and answers from the same platform.

FEVER (Thorne et al., 2018) is a fact-verification dataset. The queries are short factual claims, and the corpus introductory sections of Wikipedia pages.

Table 5 provides dataset statistics, including query counts, corpus sizes, and the average number of relevant documents per query.

### C.1.2. EMBEDDING MODELS

We generate embeddings using a mix of proprietary API services and open-weight embedding models.

`Mistral` We use Mistral's commercial Embeddings API with the mistral-embed model(Jiang et al., 2023). Mistral's model family is designed for efficient inference; we use the hosted embeddings endpoint as provided by Mistral.

`OpenAI` We use OpenAI's Embeddings API with text-embedding-3-small(OpenAI). We embed queries and documents using the same embedding model and endpoint (i.e., no separate query/document encoders or prompt format is required by the API).

`GTE` We use gte-Qwen2-7B-instruct(Li et al., 2023), a 7B-parameter text embedding model in the General Text Embeddings (GTE) family, trained on top of the `Qwen2-7B` backbone. We follow the recommended usage: queries are encoded with a query-specific prompt, whereas documents are encoded without instructions.

`Qwen` We use Qwen3-Embedding-8B (Zhang et al., 2025), an instruction-aware embedding model built on the Qwen3 foundation models. We follow the recommended asymmetric encoding: queries are encoded with a query-specific prompt, while documents are encoded unchanged.

`KaLM` We use KaLM-Embedding-Gemma3-12B (Zhao et al., 2025), a 12B-parameter embedding model from Tencent built on the Gemma3 foundation. The model uses symmetric encoding for both queries and documents, with L2-normalized output embeddings.

*Table 6.* **Embedding output dimensionality for the models used in our main experiments.** Abbr. denotes the model shorthand used throughout the paper.

| Model | Abbr. | Dim |
|---|---|---|
| KaLM-Gemma3-12B-2511 | KaLM | 3840 |
| Qwen3-Embedding-8B | Qwen | 4096 |
| text-embedding-3-small | OpenAI | 1536 |
| mistral-embed | Mistral | 1024 |
| gte-Qwen2-7B-instruct | GTE | 3584 |

Table 6 summarizes the embedding dimensionality of the models used in our main experiments.

### C.1.3. BASELINE METHODS

For all alignment baselines (Linear, CCA, MLP, Proc, RCSLS), we align $\mathbf{E}_1$ with $\mathbf{E}_2$ space and infer links using CSLS cosine similarity-based mutual nearest neighbors.

**Linear transformation** (Linear) We performs alignment by learning a linear map from $\mathbf{E}_1$ to $\mathbf{E}_2$. Given seed links $\{(a_i, b_i)\}_{i \in \mathcal{S}}$, where $a_i \in \mathbb{R}^{d_s}$ and $b_i \in \mathbb{R}^{d_t}$ are embeddings of the same item in the source and target spaces, respectively, we learn a bias-free linear transformation $W \in \mathbb{R}^{d_t \times d_s}$ by minimizing mean squared error $\mathcal{L}(W) = \frac{1}{|\mathcal{S}|} \sum_{i \in \mathcal{S}} \|Wa_i - b_i\|_2^2$. We optimize $W$ with Adam (learning rate $10^{-3}$) for 100 epochs.

**Canonical Correlation Analysis** (CCA). We standardize each space independently, fit CCA on the seed pairs to learn one linear projection per space that maximizes correlation between projected seed embeddings.

**Multi-Layer Perceptron** (MLP). We train a single-hidden-layer MLP mapping from the source embedding dimension to the target embedding dimension, with hidden width 512 and ReLU activations. We train on seed pairs using a cosine loss, optimization uses Adam with learning rate $10^{-2}$ and weight decay $10^{-5}$ for 100 training epochs.

**Procrustes** (Proc). We align the two embedding spaces by solving the orthogonal Procrustes problem on the seed links. Let $\mathbf{X}, \mathbf{Y} \in \mathbb{R}^{n \times d}$ denote the corresponding seed embeddings (rows are paired items). We estimate an orthogonal map $\mathbf{R}^\star = \arg\min_{\mathbf{R} \in \mathbb{R}^{d \times d}} \|\mathbf{X}\mathbf{R} - \mathbf{Y}\|_F^2$ s.t. $\mathbf{R}^\top \mathbf{R} = \mathbf{I}$. Let $\mathbf{A} = \mathbf{X}^\top \mathbf{Y}$ and compute its SVD $\mathbf{A} = \mathbf{U}\mathbf{\Sigma}\mathbf{V}^\top$. A closed-form optimum is given by $\mathbf{R}^\star = \mathbf{U}\mathbf{V}^\top$. At inference time, we align embedding $\mathbf{x}$ via $\mathbf{x}\mathbf{R}^\star$.

**Relaxed Cross-domain Similarity Local Scaling** (RCSLS) We implement RCSLS (Joulin et al., 2018), which directly optimizes a linear map to improve CSLS-based retrieval and mitigate hubness. We initialize with the Procrustes solution and optimize the RCSLS objective on the seed pairs using gradient-based optimization, projecting back to the orthogonal group after each update.

**Unbalanced Gromov-Wasserstein** (UGW). We implement Unbalanced Gromov-Wasserstein (Séjourné et al., 2021) with POT's log-domain Sinkhorn solver on intra-view cosine distance matrices (normalized by their mean), warm-started by a seed-biased coupling on the supervised pairs. Links are read from the resulting transport plan via mutual argmax.

**Bootstrapping parallel anchors** (AO). We implement AO (Cannistraci et al., 2023) to discover links via relative representations and Sinkhorn OT. We $\ell_2$-normalize embeddings and follow the original optimization schedule (250 steps; one Sinkhorn iteration per step). We set the anchor budget to the true overlap size, $K = \alpha|\mathcal{D}|$ (i.e., AO is given the overlap cardinality). All other baselines and GEH does not assume knowledge of $\alpha$.

### C.1.4. COMPUTATIONAL RESOURCES

All experiments were run on a Kubernetes cluster. Each run was allocated a single compute node with an AMD EPYC 7713P (64 cores), 896 GB RAM, and one NVIDIA A100 GPU (80 GB), running Ubuntu 22.04.5 LTS.

### C.2. Implementation Details and Extended Analysis of GEH

#### C.2.1. STOPPING CRITERION

We monitor the *mutual-NN ratio*, defined as the fraction of points that participate in at least one mutual nearest-neighbor (MNN) pair in an iteration. Let $\mathcal{P}_t$ be the set of MNN pairs returned at iteration $t$, and let

$$U_t := \{u \in \mathbf{E}_1 : \exists v, (u, v) \in \mathcal{P}_t\} \cup \{v \in \mathbf{E}_2 : \exists u, (u, v) \in \mathcal{P}_t\}$$

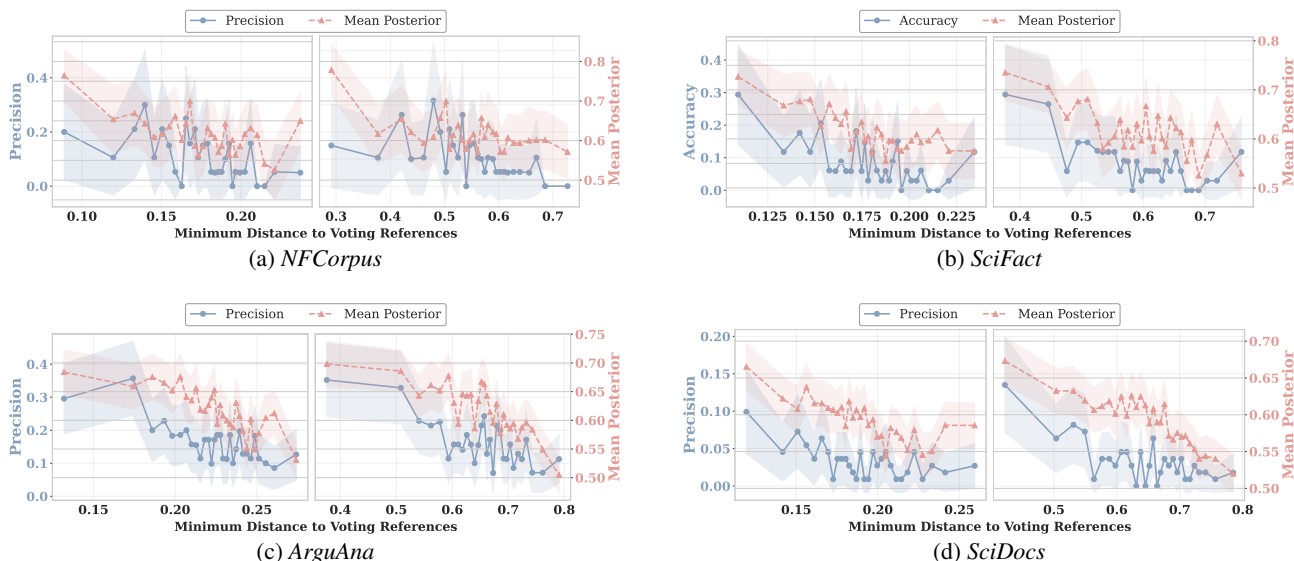

(a) *NFCorpus*  (b) *SciFact*

(c) *ArguAna*  (d) *SciDocs*

*Figure 9.* **Posterior/Precision vs. anchor proximity.** For `Mistral`↔`OpenAI` linking at $\alpha = 0.2$ overlap with $|\mathcal{S}| = 15$ seeds, we bin predicted links by their minimum distance to the anchors that voted for them (30 quantile bins) and plot per-bin empirical precision and mean posterior confidence.

where $M_t := |U_t|, N := \max\{|\mathbf{E}_1|, |\mathbf{E}_2|\}$. We define $\mathrm{MNN\_ratio}_t := M_t/N$ and terminate bootstrapping if any of the following holds: (i) $M_t = 0$ (no mutual pairs); (ii) after burn-in $T_{\min}$ (default 10), the ratio stabilizes, i.e.,

$$\max_{i \in \{t-T_{\min}+1,\ldots,t\}} |\mathrm{MNN\_ratio}_i - \mathrm{MNN\_ratio}_{i-1}| < 0.01;$$

or (iii) a maximum of 100 iterations is reached.

### C.2.2. ADDITIONAL POSTERIOR/PRECISION–PROXIMITY ANALYSES ACROSS DATASETS

We report $\mathcal{L}_1$(link set from the first iteration) under 20% ground-truth overlap with 15 seed pairs on `Mistral` and `OpenAI` embeddings. Predicted links are grouped into 30 quantile bins by their minimum cosine distance (in the corresponding embedding space) to the anchors that voted for them; we plot per-bin precision and mean posterior. As shown in Figure 9, links supported by anchors at smaller distances are more accurate, consistent with our Theorem 1, and the mean posterior closely tracks empirical precision across bins, indicating that the confidence score is well-calibrated.

### C.2.3. ROBUSTNESS TO $\mathcal{S}$ INITIALIZATION

We analyze the sensitivity of GEH to the structural properties of the initial supervision set $\mathcal{S}$. Since GEH relies on a small set of anchors sampled from the overlap region between two embedding sets, we test whether different strategies for selecting these seed pairs materially affect final performance.

**Setup.** We evaluate between `Qwen` and `OpenAI` embeddings across five datasets. We hold the overlap ratio $\alpha$ and number of seed anchors $|\mathcal{S}|$ constant while varying the sampling strategy used to select $|\mathcal{S}|$ from the overlap.

We compare 4 different strategies:

- **Nearest**: Randomly choose one anchor and take its $k-1$ nearest neighbors (in `Qwen` space). This produces a localized supervision pattern.

- **Random**: Seeds are sampled uniformly without replacement. This is the default strategy employed in our main experiments, requiring no prior knowledge of the overlap manifold.

- **FPS**: Greedily build a seed set by repeatedly selecting the candidate that maximizes its minimum cosine distance to previously selected seeds. This yields a highly diverse seed set and has a "spread-out" supervision.

- **Centroids**: We cluster the overlapping vectors (in `Qwen` space) using $k$-means (where $k = |\mathcal{S}|$) and select the vectors nearest to the centroids. This ensures seed anchors are representative of the distribution.

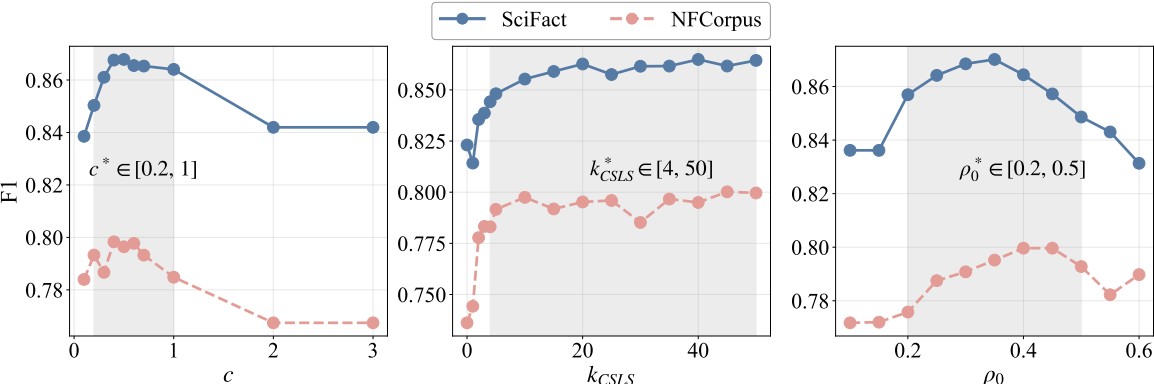

*Figure 10.* **Sensitivity to view scheduling and CSLS hyperparameters.** F1 on SciFact and NFCorpus for `Mistral` ↔ `OpenAI` linking with overlap ratio $\alpha = 0.3$ and $|\mathcal{S}| = 15$ seeds. We vary (left) the logarithmic growth constant $c$ in $\mathrm{sf}(g) = 1 + c \log g$, (middle) the CSLS neighborhood size $k_{\mathrm{CSLS}}$, and (right) the base per-view anchor fraction $\rho_0$. The shaded gray region denotes the near-optimal range achieving at least $97\%$ of the peak F1 for each sweep.

Table 7 reports F1 across all 45 (dataset, $\alpha$, $|\mathcal{S}|$) configurations. Performance is stable across strategies (the largest max−min spread is 3.0 pp), confirming that GEH is robust to the choice of seed anchors.

### C.2.4. SENSITIVITY TO $c$, $\rho_0$, AND $k_{\mathrm{CSLS}}$

We evaluate robustness on SciFact and NFCorpus using `Mistral` ↔ `OpenAI` embeddings, fixing the overlap ratio to $\alpha = 0.3$ and the seed budget to $|\mathcal{S}| = 15$. We sweep three hyperparameters: the view-growth constant $c$ in $\mathrm{sf}(g) = 1 + c \log g$, the base per-view anchor fraction $\rho_0$, and the CSLS neighborhood size $k_{\mathrm{CSLS}}$ used for MNN retrieval. When varying $\rho_0$, we set $m_0 = \lceil 2/\rho_0 \rceil$ to keep the expected per-anchor coverage approximately constant ($m_0 \rho_0 \approx 2$, up to rounding). For each sweep, we report the mean F1 across the two datasets and define the stable range as configurations achieving at least $0.97 \times$ the best mean F1 in that sweep. As shown in Figure 10, performance is stable over broad intervals: $c \in [0.2, 1]$, $k_{\mathrm{CSLS}} \in [4, 50]$, and $\rho_0 \in [0.2, 0.5]$. We use $c = 0.3$, $\rho_0 = 0.4$, and $k_{\mathrm{CSLS}} = 50$ in all experiments.

## C.3. Additional Results of GEH

### C.3.1. ADDITIONAL RESULTS ON VECTOR LINKING

We report the complete experimental grid over 5 model pairs and 5 datasets, across 3 overlap ratios and 3 seed budgets. Across this grid, GEH achieves the best performance among all methods in the vast majority of settings.

### C.3.2. ADDITIONAL RESULTS ON OUT-OF-DOMAIN ANCHORS

Figure 11 extends the main-text OOD analysis (Fig. 5) by sweeping the seed budget $n \in \{15, 20, 30\}$ and the target overlap ratio $\alpha \in \{0.15, 0.2, 0.3\}$. Overall, most reference–target pairs preserve strong precision and recall under OOD seeding; the few degraded cases align with our Theorem 1, which predicts that links supported primarily by long-range anchors are less reliable.

## D. Details of Section 6

### D.1. Implementation Details of Applications

#### D.1.1. VECTOR DATABASE INTEGRATION

We follow the evaluation protocol of Yang et al. (2025): for each benchmark corpus $O$, we place all benchmark answer documents only in $O_1 \cup O_2$ (i.e., $O_\cap$ contains no answer documents). We then build two vector databases $D_1 = \mathrm{emb}_1(O_1 \cup O_\cap)$ and $D_2 = \mathrm{emb}_2(O_2 \cup O_\cap)$, and evaluate retrieval using queries encoded by $\mathrm{emb}_2$, i.e., we query the integrated database with $\mathrm{emb}_2(q)$.

Given vector links $(X, Y)$ induced by $O_\cap$ (vectors in $D_1$ and $D_2$ that encode the same items), we compute an integration mapping $T$ from the vector space of $D_1$ to that of $D_2$ using the local-isometry-based framework of Yang et al. (2025), and

*Table 7.* **F1 (%) by seed-initialization strategy** on `Qwen`↔`OpenAI` linking across five datasets, three overlap ratios $\alpha \in \{0.15, 0.20, 0.30\}$, and three seed budgets $|\mathcal{S}| \in \{15, 20, 30\}$. Bold marks the best strategy within each row. The *Max Gap* column reports the spread (max−min) across strategies in percentage points (pp); the bolded value is the largest spread observed across all 45 setups.

| Dataset | $\alpha$ | $|\mathcal{S}|$ | Nearest | Random | FPS | Centroids | *Max Gap* |
|---|---|---|---|---|---|---|---|
| FiQA | 0.15 | 15 | **62.5** | 61.4 | 60.9 | 61.5 | 1.6 |
| | | 20 | **62.6** | 61.3 | 61.3 | 61.2 | 1.4 |
| | | 30 | 63.0 | 62.2 | 62.1 | **63.0** | 0.9 |
| | 0.20 | 15 | **68.4** | 67.5 | 67.3 | 67.0 | 1.4 |
| | | 20 | **69.0** | 68.1 | 67.2 | 67.8 | 1.8 |
| | | 30 | **68.8** | 67.8 | 68.4 | 67.6 | 1.2 |
| | 0.30 | 15 | **75.5** | 74.2 | 74.2 | 74.3 | 1.3 |
| | | 20 | **75.3** | 75.1 | 74.4 | 74.9 | 0.9 |
| | | 30 | **75.6** | 74.0 | 74.6 | 75.4 | 1.6 |
| SciDocs | 0.15 | 15 | **75.2** | 74.6 | 73.6 | 74.3 | 1.6 |
| | | 20 | 73.8 | **75.4** | 74.6 | 75.2 | 1.6 |
| | | 30 | 74.0 | 74.7 | 74.5 | **74.9** | 0.9 |
| | 0.20 | 15 | 80.1 | 80.3 | 79.5 | **80.5** | 1.0 |
| | | 20 | 79.7 | **80.3** | 80.2 | 80.2 | 0.6 |
| | | 30 | 80.1 | **80.6** | 80.4 | 79.9 | 0.7 |
| | 0.30 | 15 | 86.9 | **87.2** | 86.7 | 86.6 | 0.6 |
| | | 20 | 86.7 | **87.0** | 86.9 | 86.7 | 0.3 |
| | | 30 | 87.0 | **87.2** | 86.9 | 86.6 | 0.6 |
| ArguAna | 0.15 | 15 | 67.1 | 67.6 | 67.1 | **68.7** | 1.6 |
| | | 20 | 67.9 | 68.0 | 67.6 | **68.3** | 0.7 |
| | | 30 | **68.0** | 67.3 | 67.1 | 67.7 | 0.9 |
| | 0.20 | 15 | 74.2 | 74.9 | 74.5 | **74.9** | 0.7 |
| | | 20 | 74.1 | **75.3** | 74.3 | 74.8 | 1.2 |
| | | 30 | 74.1 | 74.6 | 73.9 | **75.0** | 1.1 |
| | 0.30 | 15 | 82.3 | 82.1 | 82.8 | **83.1** | 1.0 |
| | | 20 | 82.4 | 82.0 | 82.4 | **82.9** | 0.9 |
| | | 30 | **82.4** | 82.0 | 82.2 | 82.3 | 0.4 |
| SciFact | 0.15 | 15 | 74.7 | 76.5 | **77.7** | 77.5 | **3.0** |
| | | 20 | 76.3 | **77.4** | 76.7 | 76.3 | 1.1 |
| | | 30 | 75.3 | **75.4** | 75.1 | 74.4 | 1.0 |
| | 0.20 | 15 | 81.8 | 83.1 | 82.4 | **83.2** | 1.4 |
| | | 20 | 82.1 | 83.1 | **83.5** | 82.5 | 1.4 |
| | | 30 | 82.3 | 81.8 | **82.4** | 81.4 | 1.0 |
| | 0.30 | 15 | 89.8 | 89.9 | 89.6 | **90.0** | 0.4 |
| | | 20 | **90.0** | 90.0 | 89.4 | 90.0 | 0.6 |
| | | 30 | 89.4 | **89.7** | 89.2 | 89.3 | 0.5 |
| NFCorpus | 0.15 | 15 | 73.1 | **74.4** | 74.0 | 73.9 | 1.3 |
| | | 20 | **73.3** | 73.0 | 73.3 | 72.8 | 0.5 |
| | | 30 | 71.4 | **72.4** | 71.8 | 71.0 | 1.4 |
| | 0.20 | 15 | **80.2** | 79.4 | 79.7 | 79.7 | 0.8 |
| | | 20 | 79.3 | 79.6 | **80.0** | 79.4 | 0.7 |
| | | 30 | 77.8 | **79.5** | 78.9 | 78.1 | 1.7 |
| | 0.30 | 15 | 87.7 | **88.3** | 87.9 | 87.0 | 1.3 |
| | | 20 | 87.7 | **88.3** | 87.5 | 87.7 | 0.8 |
| | | 30 | **88.1** | 87.4 | 87.4 | 87.2 | 0.9 |

*Table 8.* **Vector linking on NFCorpus (GTE↔Mistral):** each cell reports **precision/recall/F1** (%). Best values per metric are **bolded**.

| Method | Overlap .15 | | | Overlap .20 | | | Overlap .30 | | |
|---|---|---|---|---|---|---|---|---|---|
| | Seeds 15 | Seeds 20 | Seeds 30 | Seeds 15 | Seeds 20 | Seeds 30 | Seeds 15 | Seeds 20 | Seeds 30 |
| Linear | 1.1/1.2/1.1 | 1.8/2.0/1.9 | 3.6/3.9/3.8 | 1.1/1.2/1.2 | 1.9/2.3/2.1 | 4.3/4.7/4.5 | 0.9/1.3/1.1 | 1.5/2.5/1.9 | 4.7/6.6/5.5 |
| CCA | 33.3/12.8/18.5 | 38.4/20.8/26.9 | 47.9/42.5/45.1 | 31.7/8.0/12.8 | 44.3/17.5/25.1 | 55.7/37.2/44.6 | 46.0/7.5/13.0 | 54.1/18.5/27.6 | 68.1/36.6/47.6 |
| MLP | 10.8/2.1/3.5 | 23.0/6.1/9.6 | 35.7/18.4/24.3 | 15.6/2.1/3.7 | 25.1/5.9/9.6 | 41.8/17.9/25.1 | 27.0/2.8/5.1 | 43.0/6.9/11.9 | 54.0/17.7/26.6 |
| RCSLS | **62.8**/2.5/4.8 | 40.4/3.6/6.5 | 33.4/7.4/12.1 | 60.9/2.0/3.9 | 51.3/3.1/5.9 | 44.7/9.3/15.4 | 63.8/1.3/2.6 | 59.2/2.0/3.9 | 50.9/4.9/9.0 |
| Proc | 34.8/13.0/19.0 | 40.1/22.3/28.6 | 48.6/41.0/44.5 | 37.6/9.0/14.6 | 46.0/18.5/26.4 | 56.6/38.2/45.6 | 53.0/9.9/16.6 | 61.8/22.1/32.5 | 72.5/37.7/49.6 |
| UGW | 2.3/1.7/2.0 | 2.1/1.5/1.8 | 2.3/1.7/2.0 | 17.8/13.1/15.1 | 17.7/13.0/15.0 | 18.0/13.3/15.3 | 20.7/10.9/14.3 | 20.5/10.8/14.1 | 19.8/10.4/13.6 |
| AO | 3.3/3.3/3.3 | 22.8/22.8/22.8 | 26.1/26.1/26.1 | 5.6/5.6/5.6 | 11.4/11.4/11.4 | 25.3/25.3/25.3 | 1.7/1.7/1.7 | 2.5/2.5/2.5 | 39.5/39.5/39.5 |
| GEH | 58.8/**78.3**/67.2 | **58.1**/**78.3**/66.7 | 57.7/77.3/66.1 | 67.5/75.4/71.2 | 65.5/74.5/69.7 | 66.0/75.6/70.5 | 79.6/78.8/79.2 | 79.7/78.7/79.2 | 77.5/78.2/77.9 |

*Table 9.* **Vector linking on SciDocs (GTE↔Mistral):** each cell reports **precision/recall/F1** (%). Best values per metric are **bolded**.

| Method | Overlap .15 | | | Overlap .20 | | | Overlap .30 | | |
|---|---|---|---|---|---|---|---|---|---|
| | Seeds 15 | Seeds 20 | Seeds 30 | Seeds 15 | Seeds 20 | Seeds 30 | Seeds 15 | Seeds 20 | Seeds 30 |
| Linear | 0.3/0.1/0.2 | 0.5/0.3/0.4 | 0.9/0.9/0.9 | 0.2/0.2/0.2 | 0.4/0.3/0.3 | 1.0/0.9/1.0 | 0.3/0.3/0.3 | 0.5/0.5/0.5 | 1.2/1.5/1.3 |
| CCA | 13.0/2.1/3.6 | 23.0/5.4/8.8 | 37.2/21.6/27.3 | 17.2/2.3/4.0 | 29.8/7.0/11.4 | 43.6/19.2/26.6 | 18.2/2.0/3.6 | 37.0/6.7/11.3 | 59.1/21.1/31.1 |
| MLP | 7.0/0.4/0.7 | 10.8/1.0/1.8 | 26.0/6.0/9.8 | 11.0/0.5/1.0 | 18.9/1.2/2.2 | 26.8/4.0/7.0 | 11.4/0.4/0.8 | 24.6/1.4/2.7 | 33.8/3.9/7.0 |
| RCSLS | 60.9/0.3/0.7 | 45.3/0.5/1.0 | 19.5/1.0/1.9 | 55.2/0.2/0.5 | 38.0/0.3/0.7 | 20.0/0.7/1.4 | 44.9/0.2/0.3 | 27.9/0.3/0.5 | 20.4/0.6/1.2 |
| Proc | 17.8/2.1/3.7 | 29.5/6.2/10.3 | 47.4/22.8/30.8 | 23.7/2.7/4.9 | 36.2/7.4/12.3 | 52.5/21.8/30.8 | 26.7/2.3/4.2 | 48.1/7.5/12.9 | 67.6/21.7/32.8 |
| UGW | 5.9/0.6/1.1 | 5.9/0.6/1.1 | 5.9/0.6/1.1 | 4.4/0.3/0.6 | 4.4/0.3/0.6 | 4.4/0.3/0.6 | 7.0/0.4/0.7 | 7.1/0.4/0.7 | 7.1/0.4/0.7 |
| AO | 0.4/0.4/0.4 | 0.5/0.5/0.5 | 0.8/0.8/0.8 | 0.3/0.3/0.3 | 0.4/0.4/0.4 | 1.1/1.1/1.1 | 0.2/0.2/0.2 | 0.3/0.3/0.3 | 0.4/0.4/0.4 |
| GEH | **62.8**/55.2/58.7 | 62.5/51.7/56.6 | 62.0/62.3/62.2 | 70.0/60.0/64.6 | 71.7/60.8/65.8 | 71.7/63.8/67.5 | 80.8/64.3/71.6 | 82.2/67.2/74.0 | 81.3/68.0/74.1 |

*Table 10.* **Vector linking on ArguAna (GTE↔Mistral):** each cell reports **precision/recall/F1** (%). Best values per metric are **bolded**.

| Method | Overlap .15 | | | Overlap .20 | | | Overlap .30 | | |
|---|---|---|---|---|---|---|---|---|---|
| | Seeds 15 | Seeds 20 | Seeds 30 | Seeds 15 | Seeds 20 | Seeds 30 | Seeds 15 | Seeds 20 | Seeds 30 |
| Linear | 0.4/0.4/0.4 | 0.6/0.5/0.5 | 1.0/0.8/0.9 | 0.4/0.3/0.3 | 0.6/0.5/0.5 | 1.1/1.2/1.2 | 0.4/0.3/0.3 | 0.6/0.7/0.6 | 1.5/1.5/1.5 |
| CCA | 10.8/3.1/4.8 | 17.9/7.4/10.5 | 33.0/19.6/24.6 | 14.1/3.4/5.4 | 24.0/8.2/12.3 | 37.6/18.2/24.6 | 25.2/3.9/6.8 | 35.5/8.5/13.7 | 50.3/21.3/30.0 |
| MLP | 9.7/1.0/1.8 | 12.0/1.8/3.1 | 19.9/5.3/8.4 | 13.6/1.2/2.2 | 17.3/2.3/4.1 | 27.7/6.2/10.2 | 11.0/0.7/1.2 | 18.0/1.6/3.0 | 33.5/5.1/8.8 |
| RCSLS | 38.3/0.6/1.2 | 41.8/1.1/2.1 | 28.0/2.3/4.2 | 39.3/0.5/1.0 | 30.6/0.8/1.6 | 26.5/1.9/3.5 | 26.2/0.2/0.4 | 26.3/0.3/0.6 | 30.7/1.2/2.3 |
| Proc | 15.9/4.8/7.4 | 24.2/10.1/14.3 | 36.9/21.9/27.5 | 20.5/4.9/7.9 | 28.9/9.5/14.3 | 42.7/21.4/28.5 | 30.8/4.8/8.4 | 39.4/9.9/15.8 | 60.6/26.6/36.9 |
| UGW | 0.0/0.0/0.0 | 0.0/0.0/0.0 | 0.0/0.0/0.0 | 0.0/0.0/0.0 | 0.0/0.0/0.0 | 0.0/0.0/0.0 | 5.9/0.1/0.2 | 5.9/0.1/0.2 | 6.1/0.1/0.2 |
| AO | 1.6/1.6/1.6 | 1.6/1.6/1.6 | 2.9/2.9/2.9 | 0.9/0.9/0.9 | 1.5/1.5/1.5 | 30.6/30.6/30.6 | 0.6/0.6/0.6 | 1.6/1.6/1.6 | 1.5/1.5/1.5 |
| GEH | 53.2/83.4/65.0 | 53.7/81.7/64.9 | 55.5/81.8/66.1 | 64.0/84.2/72.7 | 64.3/84.0/72.8 | 64.1/83.5/72.5 | 77.1/84.5/80.7 | 77.5/83.5/80.4 | 77.7/83.3/80.4 |

*Table 11.* **Vector linking on SciFact (GTE↔Mistral):** each cell reports **precision/recall/F1** (%). Best values per metric are **bolded**.

| Method | Overlap .15 | | | Overlap .20 | | | Overlap .30 | | |
|---|---|---|---|---|---|---|---|---|---|
| | Seeds 15 | Seeds 20 | Seeds 30 | Seeds 15 | Seeds 20 | Seeds 30 | Seeds 15 | Seeds 20 | Seeds 30 |
| Linear | 0.8/0.6/0.7 | 1.3/1.7/1.5 | 3.0/3.4/3.2 | 0.8/1.0/0.9 | 1.4/1.7/1.6 | 3.1/3.9/3.4 | 0.7/1.0/0.8 | 1.8/2.4/2.0 | 3.7/5.3/4.4 |
| CCA | 22.2/5.4/8.7 | 34.6/14.3/20.2 | 43.8/34.4/38.5 | 29.6/7.9/12.5 | 43.6/16.5/24.0 | 57.9/38.9/46.6 | 36.1/5.9/10.2 | 49.6/14.0/21.9 | 70.4/35.8/47.5 |
| MLP | 8.2/1.1/1.9 | 15.1/3.3/5.4 | 32.3/15.1/20.6 | 20.3/2.5/4.4 | 28.3/5.7/9.5 | 41.4/13.9/20.8 | 26.0/1.7/3.2 | 38.0/4.2/7.6 | 57.8/16.6/25.8 |
| RCSLS | **69.9**/1.8/3.5 | 54.3/3.0/5.6 | 40.4/6.3/11.0 | **82.4**/1.5/3.0 | 71.7/2.1/4.1 | 44.0/4.7/8.4 | 66.9/1.0/2.0 | 47.7/1.3/2.6 | 42.9/3.5/6.4 |
| Proc | 25.3/5.8/9.4 | 42.5/15.8/23.1 | 48.0/37.4/42.0 | 31.3/8.0/12.8 | 49.1/16.3/24.5 | 61.1/38.1/46.9 | 45.6/7.7/13.2 | 56.9/15.6/24.5 | 72.9/35.9/48.1 |
| UGW | 6.2/1.8/2.8 | 6.2/1.8/2.9 | 5.8/1.7/2.7 | 9.4/2.3/3.6 | 8.8/2.1/3.3 | 7.9/1.9/3.0 | 13.1/2.3/3.9 | 13.0/2.3/3.9 | 12.9/2.2/3.8 |
| AO | 4.2/4.2/4.2 | 4.9/4.9/4.9 | 25.7/25.7/25.7 | 1.7/1.7/1.7 | 5.8/5.8/5.8 | 6.8/6.8/6.8 | 1.1/1.1/1.1 | 1.7/1.7/1.7 | 1.9/1.9/1.9 |
| GEH | 62.0/**82.0**/70.6 | 62.3/81.6/70.7 | 60.1/81.8/69.3 | 72.3/**84.8**/78.1 | 72.6/83.3/77.6 | 70.4/85.2/77.1 | 80.4/83.6/82.0 | 81.5/83.9/82.7 | 80.9/84.3/82.6 |

*Table 12.* **Vector linking on FiQA (GTE↔Mistral):** each cell reports **precision/recall/F1** (%). Best values per metric are **bolded**.

| Method | Overlap .15 | | | Overlap .20 | | | Overlap .30 | | |
|---|---|---|---|---|---|---|---|---|---|
| | Seeds 15 | Seeds 20 | Seeds 30 | Seeds 15 | Seeds 20 | Seeds 30 | Seeds 15 | Seeds 20 | Seeds 30 |
| Linear | 0.1/0.1/0.1 | 0.1/0.1/0.1 | 0.3/0.2/0.2 | 0.1/0.1/0.1 | 0.1/0.1/0.1 | 0.3/0.3/0.3 | 0.1/0.1/0.1 | 0.2/0.1/0.2 | 0.3/0.3/0.3 |
| CCA | 5.9/1.1/1.9 | 11.4/3.0/4.7 | 23.6/10.9/14.9 | 9.7/1.3/2.3 | 16.4/3.6/5.9 | 32.9/13.3/18.9 | 8.7/0.5/1.0 | 18.2/2.3/4.1 | 34.2/9.2/14.4 |
| MLP | 3.2/0.2/0.3 | 5.5/0.5/0.9 | 10.5/1.5/2.6 | 4.1/0.2/0.4 | 9.0/0.7/1.3 | 12.4/1.6/2.8 | 4.9/0.1/0.2 | 11.2/0.4/0.8 | 19.1/1.4/2.5 |
| RCSLS | 25.6/0.1/0.2 | 17.2/0.2/0.3 | 15.8/0.4/0.9 | 29.0/0.1/0.1 | 23.4/0.1/0.2 | 16.9/0.4/0.7 | 22.1/0.0/0.1 | 21.4/0.1/0.2 | 14.3/0.2/0.4 |
| Proc | 9.2/1.4/2.5 | 17.6/4.0/6.5 | 31.8/12.5/17.9 | 15.5/1.9/3.4 | 25.0/5.0/8.4 | 43.9/16.0/23.4 | 14.0/0.8/1.6 | 25.8/2.7/4.9 | 45.7/10.0/16.4 |
| UGW | 0.0/0.0/0.0 | 0.0/0.0/0.0 | 0.0/0.0/0.0 | 1.1/0.0/0.0 | 1.1/0.0/0.0 | 1.1/0.0/0.0 | 6.4/0.0/0.1 | 6.4/0.0/0.1 | 6.4/0.0/0.1 |
| AO | 0.2/0.2/0.2 | 0.2/0.2/0.2 | 0.3/0.3/0.3 | 0.1/0.1/0.1 | 0.2/0.2/0.2 | 0.3/0.3/0.3 | 0.1/0.1/0.1 | 0.1/0.1/0.1 | 0.2/0.2/0.2 |
| GEH | 54.8/75.1/63.3 | 54.8/74.6/63.2 | 55.4/73.3/63.1 | 64.9/75.0/69.6 | 64.5/74.6/69.2 | 65.0/73.6/69.0 | 78.6/77.9/78.2 | 78.4/77.3/77.8 | 78.5/75.9/77.2 |

*Table 13.* **Vector linking on NFCorpus (`GTE↔OpenAI`):** each cell reports **precision/recall/F1** (%). Best values per metric are **bolded**.

| Method | Overlap .15 | | | Overlap .20 | | | Overlap .30 | | |
|--------|----------|----------|----------|----------|----------|----------|----------|----------|----------|
| | Seeds 15 | Seeds 20 | Seeds 30 | Seeds 15 | Seeds 20 | Seeds 30 | Seeds 15 | Seeds 20 | Seeds 30 |
| Linear | 2.2/5.9/3.2 | 4.5/9.1/6.0 | 9.2/16.0/11.7 | 2.2/5.3/3.1 | 4.1/10.2/5.9 | 8.7/19.2/11.9 | 3.5/7.5/4.8 | 5.7/14.3/8.1 | 11.2/24.7/15.4 |
| CCA | 23.0/8.7/12.6 | 35.8/20.2/25.8 | 42.4/41.0/41.7 | 32.4/9.1/14.2 | 37.6/16.3/22.8 | 50.6/33.6/40.4 | 46.4/9.5/15.8 | 50.9/16.2/24.6 | 65.5/33.2/44.1 |
| MLP | 10.9/2.4/3.9 | 18.3/5.7/8.7 | 24.6/11.0/15.2 | 13.1/2.0/3.4 | 24.4/5.8/9.4 | 28.1/10.0/14.8 | 23.7/2.1/3.8 | 29.1/4.8/8.2 | 44.1/10.9/17.4 |
| RCSLS | 24.6/4.2/7.2 | 26.1/8.0/12.2 | 37.3/23.4/28.8 | 28.3/3.7/6.6 | 36.4/7.9/12.9 | 47.1/23.1/31.0 | 27.2/2.3/4.2 | 35.6/6.3/10.7 | 53.0/20.6/29.6 |
| Proc | 32.3/12.1/17.6 | 40.7/22.1/28.6 | 47.9/42.7/45.2 | 33.8/9.8/15.2 | 45.6/19.8/27.6 | 54.9/38.5/45.2 | 48.2/9.9/16.4 | 56.5/18.2/27.6 | 69.5/36.4/47.8 |
| UGW | 10.9/7.0/8.5 | 10.7/6.9/8.3 | 11.0/7.2/8.7 | 11.7/6.2/8.1 | 11.8/6.2/8.1 | 11.7/6.2/8.1 | 17.2/6.1/8.9 | 17.1/6.0/8.9 | 16.6/5.9/8.7 |
| AO | 4.4/4.4/4.4 | 20.9/20.9/20.9 | 25.5/25.5/25.5 | 3.6/3.6/3.6 | 14.2/14.2/14.2 | 32.3/32.3/32.3 | 3.2/3.2/3.2 | 34.7/34.7/34.7 | 40.4/40.4/40.4 |
| GEH | **58.4/72.6/64.8** | **57.7/78.9/66.6** | **54.7/73.4/62.7** | **68.2/79.4/73.3** | **68.5/81.0/74.3** | **66.6/77.9/71.8** | **80.1/83.3/81.7** | **79.9/81.6/80.7** | **78.9/78.7/78.8** |

*Table 14.* **Vector linking on SciDocs (`GTE↔OpenAI`):** each cell reports **precision/recall/F1** (%). Best values per metric are **bolded**.

| Method | Overlap .15 | | | Overlap .20 | | | Overlap .30 | | |
|--------|----------|----------|----------|----------|----------|----------|----------|----------|----------|
| | Seeds 15 | Seeds 20 | Seeds 30 | Seeds 15 | Seeds 20 | Seeds 30 | Seeds 15 | Seeds 20 | Seeds 30 |
| Linear | 1.8/1.2/1.4 | 2.6/2.7/2.7 | 5.2/6.5/5.7 | 1.9/1.3/1.6 | 2.8/2.6/2.7 | 5.9/8.3/6.8 | 1.5/1.8/1.6 | 2.8/4.4/3.4 | 6.3/11.9/8.2 |
| CCA | 8.5/1.6/2.7 | 15.8/5.2/7.8 | 29.1/16.2/20.8 | 11.7/1.7/2.9 | 23.0/5.8/9.2 | 33.9/14.2/20.1 | 12.3/1.6/2.8 | 28.1/5.2/8.8 | 48.9/16.8/25.0 |
| MLP | 4.5/0.4/0.7 | 6.7/0.8/1.5 | 13.0/2.3/3.9 | 7.1/0.4/0.7 | 10.4/0.9/1.7 | 16.7/2.6/4.6 | 10.4/0.4/0.7 | 17.4/1.1/2.0 | 28.7/3.3/6.0 |
| RCSLS | 14.0/0.6/1.1 | 13.4/1.5/2.7 | 22.8/5.9/9.4 | 13.6/0.4/0.8 | 14.3/1.0/1.9 | 26.2/4.9/8.2 | 14.7/0.4/0.8 | 18.8/1.1/2.2 | 36.0/5.5/9.6 |
| Proc | 11.6/1.5/2.7 | 20.7/5.0/8.0 | 35.9/16.2/22.3 | 15.9/2.0/3.6 | 27.6/5.7/9.5 | 44.9/17.2/24.8 | 20.1/2.1/3.8 | 36.6/6.0/10.4 | 59.6/18.5/28.2 |
| UGW | 3.4/0.1/0.2 | 3.4/0.1/0.2 | 3.3/0.1/0.2 | 3.2/0.1/0.2 | 3.2/0.1/0.2 | 3.2/0.1/0.2 | 4.1/0.1/0.1 | 4.1/0.1/0.1 | 4.2/0.1/0.1 |
| AO | 0.4/0.4/0.4 | 0.6/0.6/0.6 | 20.3/20.3/20.3 | 1.1/1.1/1.1 | 0.5/0.5/0.5 | 2.4/2.4/2.4 | 0.2/0.2/0.2 | 0.3/0.3/0.3 | 0.5/0.5/0.5 |
| GEH | **60.4/84.3/70.4** | **61.5/82.1/70.3** | **63.9/79.1/70.6** | **69.9/84.4/76.5** | **69.7/83.2/75.9** | **71.1/82.3/76.3** | **82.2/84.5/83.3** | **82.2/83.7/82.9** | **82.4/83.2/82.8** |

*Table 15.* **Vector linking on ArguAna (`GTE↔OpenAI`):** each cell reports **precision/recall/F1** (%). Best values per metric are **bolded**.

| Method | Overlap .15 | | | Overlap .20 | | | Overlap .30 | | |
|--------|----------|----------|----------|----------|----------|----------|----------|----------|----------|
| | Seeds 15 | Seeds 20 | Seeds 30 | Seeds 15 | Seeds 20 | Seeds 30 | Seeds 15 | Seeds 20 | Seeds 30 |
| Linear | 2.2/1.2/1.5 | 2.9/2.5/2.7 | 5.6/6.1/5.9 | 2.2/1.6/1.8 | 3.2/3.2/3.2 | 6.4/8.4/7.2 | 2.5/2.3/2.4 | 3.4/5.0/4.1 | 6.4/11.6/8.2 |
| CCA | 13.0/3.8/5.8 | 19.7/8.4/11.8 | 30.7/18.9/23.4 | 16.1/4.1/6.5 | 28.6/10.2/15.0 | 38.7/19.3/25.8 | 20.5/3.6/6.2 | 29.6/7.3/11.7 | 50.5/21.3/29.9 |
| MLP | 5.1/0.7/1.2 | 10.2/1.7/2.8 | 18.0/4.2/6.8 | 6.9/0.9/1.5 | 13.8/2.2/3.7 | 21.0/4.1/6.9 | 9.7/0.8/1.4 | 17.6/1.8/3.2 | 29.0/3.7/6.5 |
| RCSLS | 15.3/1.7/3.1 | 16.8/3.8/6.2 | 25.9/12.4/16.8 | 17.0/1.4/2.6 | 18.0/3.6/6.0 | 33.3/12.7/18.4 | 17.4/1.1/2.2 | 21.1/2.7/4.8 | 40.1/10.6/16.7 |
| Proc | 13.9/4.0/6.3 | 23.4/9.8/13.8 | 34.8/21.7/26.7 | 21.0/5.5/8.8 | 31.2/10.5/15.7 | 43.0/21.3/28.5 | 28.8/5.0/8.5 | 36.8/9.6/15.3 | 57.0/25.2/34.9 |
| UGW | 8.9/0.3/0.6 | 8.9/0.3/0.6 | 8.9/0.3/0.6 | 2.1/0.1/0.1 | 2.1/0.1/0.1 | 2.1/0.1/0.1 | 6.4/0.1/0.2 | 6.4/0.1/0.2 | 6.4/0.1/0.2 |
| AO | 1.3/1.3/1.3 | 11.8/11.8/11.8 | 26.5/26.5/26.5 | 1.7/1.7/1.7 | 31.5/31.5/31.5 | 31.4/31.4/31.4 | 1.4/1.4/1.4 | 43.0/43.0/43.0 | 42.8/42.8/42.8 |
| GEH | **52.2/81.0/63.4** | **53.6/81.0/64.5** | **53.1/80.0/63.8** | **62.6/82.1/71.0** | **64.2/81.2/71.7** | **62.8/80.1/70.4** | **76.7/81.7/79.1** | **77.1/80.3/78.7** | **76.5/79.5/78.0** |

*Table 16.* **Vector linking on SciFact (`GTE↔OpenAI`):** each cell reports **precision/recall/F1** (%). Best values per metric are **bolded**.

| Method | Overlap .15 | | | Overlap .20 | | | Overlap .30 | | |
|--------|----------|----------|----------|----------|----------|----------|----------|----------|----------|
| | Seeds 15 | Seeds 20 | Seeds 30 | Seeds 15 | Seeds 20 | Seeds 30 | Seeds 15 | Seeds 20 | Seeds 30 |
| Linear | 3.1/4.2/3.5 | 5.4/7.7/6.4 | 10.1/14.3/11.8 | 3.2/4.9/3.9 | 4.9/8.9/6.3 | 9.2/16.8/11.9 | 2.2/5.0/3.1 | 5.3/11.8/7.3 | 11.0/23.9/15.1 |
| CCA | 21.8/5.8/9.1 | 32.3/14.8/20.3 | 42.7/34.5/38.2 | 24.8/6.9/10.7 | 39.2/14.2/20.8 | 51.9/34.7/41.6 | 29.3/5.2/8.8 | 43.1/11.9/18.7 | 65.5/35.4/46.0 |
| MLP | 13.1/1.8/3.1 | 14.8/3.6/5.8 | 25.1/10.7/15.0 | 20.0/3.7/6.3 | 23.8/5.8/9.3 | 32.9/12.4/18.0 | 14.2/1.4/2.5 | 22.6/3.3/5.7 | 39.9/10.5/16.6 |
| RCSLS | 27.3/3.5/6.1 | 28.1/7.3/11.5 | 39.4/21.9/28.2 | 25.2/2.4/4.3 | 28.3/5.3/8.9 | 42.0/18.1/25.3 | 28.6/2.1/3.8 | 34.3/5.0/8.8 | 54.5/17.6/26.6 |
| Proc | 27.4/7.1/11.3 | 35.6/14.5/20.6 | 45.8/34.9/39.6 | 27.7/7.7/12.1 | 45.2/15.1/22.7 | 58.6/35.9/44.5 | 37.9/5.7/9.9 | 52.5/13.7/21.7 | 68.6/34.8/46.1 |
| UGW | 10.7/3.1/4.9 | 10.3/3.0/4.7 | 10.4/3.1/4.8 | 11.6/2.6/4.3 | 11.7/2.7/4.3 | 11.8/2.7/4.4 | 15.9/2.5/4.4 | 15.9/2.5/4.4 | 15.5/2.5/4.3 |
| AO | 19.6/19.6/19.6 | 15.3/15.3/15.3 | 26.6/26.6/26.6 | 3.2/3.2/3.2 | 15.5/15.5/15.5 | 31.4/31.4/31.4 | 5.6/5.6/5.6 | 36.2/36.2/36.2 | 44.6/44.6/44.6 |
| GEH | **61.5/85.0/71.4** | **61.5/87.2/72.1** | **60.6/84.7/70.6** | **70.7/86.0/77.6** | **70.2/86.9/77.7** | **69.8/84.1/76.3** | **83.2/89.1/86.0** | **83.0/88.7/85.8** | **82.2/89.1/85.5** |

*Table 17.* **Vector linking on FiQA (`GTE↔OpenAI`):** each cell reports **precision/recall/F1** (%). Best values per metric are **bolded**.

| Method | Overlap .15 | | | Overlap .20 | | | Overlap .30 | | |
|--------|----------|----------|----------|----------|----------|----------|----------|----------|----------|
| | Seeds 15 | Seeds 20 | Seeds 30 | Seeds 15 | Seeds 20 | Seeds 30 | Seeds 15 | Seeds 20 | Seeds 30 |
| Linear | 2.3/0.3/0.5 | 2.1/0.8/1.2 | 3.9/2.9/3.3 | 3.1/0.4/0.7 | 3.2/0.9/1.4 | 4.2/3.0/3.5 | 2.6/0.6/1.0 | 3.2/1.5/2.1 | 4.8/4.8/4.8 |
| CCA | 4.3/0.8/1.4 | 8.4/2.4/3.7 | 17.3/8.3/11.3 | 6.4/0.9/1.6 | 13.0/3.0/4.9 | 26.3/10.7/15.2 | 7.2/0.5/0.9 | 12.6/1.6/2.8 | 26.3/6.8/10.8 |
| MLP | 1.8/0.2/0.3 | 2.7/0.3/0.6 | 7.8/0.8/1.5 | 2.3/0.2/0.4 | 7.6/0.8/1.5 | 14.4/2.0/3.6 | 2.6/0.1/0.2 | 5.8/0.3/0.6 | 13.5/0.8/1.5 |
| RCSLS | 3.1/0.2/0.4 | 4.8/0.8/1.3 | 11.8/3.7/5.6 | 3.0/0.2/0.4 | 5.1/0.6/1.1 | 14.4/3.3/5.4 | 3.8/0.2/0.3 | 6.7/0.6/1.2 | 20.6/3.8/6.4 |
| Proc | 6.3/1.1/1.9 | 12.1/3.0/4.8 | 26.2/10.7/15.2 | 9.3/1.2/2.2 | 19.4/4.1/6.7 | 35.8/13.0/19.0 | 8.8/0.5/1.0 | 18.6/2.1/3.7 | 36.5/8.0/13.1 |
| UGW | 0.0/0.0/0.0 | 0.0/0.0/0.0 | 0.0/0.0/0.0 | 0.0/0.0/0.0 | 0.0/0.0/0.0 | 0.0/0.0/0.0 | 2.5/0.0/0.0 | 2.5/0.0/0.0 | 2.5/0.0/0.0 |
| AO | 0.3/0.3/0.3 | 0.2/0.2/0.2 | 0.4/0.4/0.4 | 0.4/0.4/0.4 | 0.2/0.2/0.2 | 0.4/0.4/0.4 | 0.1/0.1/0.1 | 0.2/0.2/0.2 | 0.3/0.3/0.3 |
| GEH | **51.9/67.6/58.7** | **53.1/68.3/59.8** | **53.7/67.1/59.7** | **61.0/65.7/63.2** | **61.8/66.7/64.1** | **62.4/66.6/64.4** | **75.4/68.6/71.8** | **75.9/68.1/71.8** | **75.4/67.5/71.2** |

*Table 18.* **Vector linking on NFCorpus (`Mistral↔OpenAI`):** each cell reports **precision/recall/F1** (%). Best values per metric are **bolded**.

| Method | Overlap .15 | | | Overlap .20 | | | Overlap .30 | | |
|---|---|---|---|---|---|---|---|---|---|
| | Seeds 15 | Seeds 20 | Seeds 30 | Seeds 15 | Seeds 20 | Seeds 30 | Seeds 15 | Seeds 20 | Seeds 30 |
| Linear | 4.3/6.6/5.2 | 5.6/10.7/7.3 | 10.2/16.0/12.5 | 2.9/5.6/3.8 | 4.6/10.7/6.5 | 10.1/21.0/13.6 | 4.1/9.1/5.7 | 5.7/15.0/8.2 | 11.4/25.9/15.9 |
| CCA | 18.6/7.5/10.7 | 28.3/15.9/20.3 | 39.4/34.9/37.0 | 22.7/6.3/9.9 | 37.1/14.8/21.2 | 46.4/31.3/37.4 | 34.1/6.7/11.2 | 50.3/15.8/24.1 | 58.8/30.4/40.1 |
| MLP | 14.5/2.1/3.6 | 21.1/5.7/9.0 | 28.0/10.9/15.7 | 18.3/1.5/2.8 | 26.6/4.1/7.1 | 40.4/15.3/22.3 | 29.8/2.0/3.7 | 33.4/4.7/8.3 | 48.1/10.3/17.0 |
| RCSLS | 53.8/3.7/6.9 | 42.5/6.1/10.7 | 43.4/17.0/24.4 | 45.5/5.7/10.1 | 52.6/16.4/25.0 | 46.9/2.0/3.8 | 47.6/4.3/7.8 | 55.4/13.0/21.0 |
| Proc | 29.6/10.4/15.4 | 35.1/18.5/24.2 | 46.3/40.2/43.0 | 31.1/9.0/13.9 | 41.0/15.7/22.7 | 51.5/33.9/40.9 | 41.7/7.9/13.3 | 58.5/18.4/28.0 | 71.4/37.3/49.0 |
| UGW | 3.5/1.7/2.3 | 3.5/1.7/2.3 | 3.5/1.7/2.3 | 0.5/0.1/0.2 | 0.5/0.1/0.2 | 0.5/0.1/0.2 | 9.9/2.1/3.5 | 9.6/2.1/3.4 | 9.6/2.1/3.4 |
| AO | 4.8/4.8/4.8 | 17.1/17.1/17.1 | 26.2/26.2/26.2 | 4.4/4.4/4.4 | 13.1/13.1/13.1 | 32.5/32.5/32.5 | 2.5/2.5/2.5 | 41.0/41.0/41.0 | 41.9/41.9/41.9 |
| GEH | **57.6/74.8/65.1** | **57.5/75.7/65.4** | **55.4/76.1/64.1** | **67.3/77.4/72.0** | **66.8/76.2/71.2** | **66.2/77.5/71.4** | **79.1/77.0/78.0** | **79.5/78.0/78.7** | **78.3/78.2/78.3** |

*Table 19.* **Vector linking on SciDocs (`Mistral↔OpenAI`):** each cell reports **precision/recall/F1** (%). Best values per metric are **bolded**.

| Method | Overlap .15 | | | Overlap .20 | | | Overlap .30 | | |
|---|---|---|---|---|---|---|---|---|---|
| | Seeds 15 | Seeds 20 | Seeds 30 | Seeds 15 | Seeds 20 | Seeds 30 | Seeds 15 | Seeds 20 | Seeds 30 |
| Linear | 3.1/1.6/2.1 | 4.4/3.6/3.9 | 7.3/7.2/7.2 | 2.8/1.6/2.0 | 4.2/3.1/3.6 | 7.7/9.0/8.3 | 2.2/1.7/1.9 | 4.1/4.9/4.4 | 7.5/11.9/9.2 |
| CCA | 7.3/1.8/2.8 | 13.1/4.9/7.2 | 21.0/12.5/15.7 | 8.5/1.3/2.3 | 13.8/3.7/5.9 | 27.7/12.2/16.9 | 11.0/1.4/2.4 | 19.8/3.9/6.5 | 39.1/13.6/20.1 |
| MLP | 4.2/0.3/0.6 | 6.5/0.8/1.5 | 16.6/3.3/5.5 | 12.2/0.4/0.8 | 14.3/1.1/2.0 | 20.2/3.8/6.5 | 10.4/0.3/0.5 | 18.5/0.7/1.4 | 29.3/2.8/5.1 |
| RCSLS | 34.6/0.5/0.9 | 23.6/1.0/2.0 | 28.1/3.8/6.7 | 38.2/0.3/0.7 | 25.4/0.6/1.2 | 30.4/2.9/5.2 | 32.6/0.3/0.6 | 31.1/0.7/1.4 | 39.2/2.9/5.3 |
| Proc | 11.8/2.0/3.5 | 16.3/4.9/7.5 | 29.8/14.3/19.3 | 11.6/1.6/2.7 | 20.7/4.7/7.7 | 37.5/14.9/21.3 | 15.6/1.6/2.9 | 28.4/4.8/8.2 | 51.6/15.6/24.0 |
| UGW | 0.9/0.0/0.1 | 0.9/0.0/0.1 | 0.9/0.0/0.1 | 0.8/0.0/0.1 | 0.8/0.0/0.1 | 0.8/0.0/0.0 | 3.8/0.1/0.1 | 3.8/0.1/0.1 | 3.8/0.1/0.1 |
| AO | 0.4/0.4/0.4 | 0.6/0.6/0.6 | 17.7/17.7/17.7 | 5.1/5.1/5.1 | 0.6/0.6/0.6 | 1.3/1.3/1.3 | 0.2/0.2/0.2 | 0.3/0.3/0.3 | 0.5/0.5/0.5 |
| GEH | **62.1/81.7/70.5** | **63.2/77.6/69.6** | **64.9/72.6/68.5** | **71.8/82.2/76.6** | **72.2/79.0/75.4** | **73.4/75.4/74.4** | **82.8/81.6/82.2** | **83.6/81.2/82.4** | **83.6/79.4/81.5** |

*Table 20.* **Vector linking on ArguAna (`Mistral↔OpenAI`):** each cell reports **precision/recall/F1** (%). Best values per metric are **bolded**.

| Method | Overlap .15 | | | Overlap .20 | | | Overlap .30 | | |
|---|---|---|---|---|---|---|---|---|---|
| | Seeds 15 | Seeds 20 | Seeds 30 | Seeds 15 | Seeds 20 | Seeds 30 | Seeds 15 | Seeds 20 | Seeds 30 |
| Linear | 2.9/3.0/3.0 | 4.4/6.7/5.3 | 9.5/13.0/11.0 | 3.6/4.4/4.0 | 4.8/8.4/6.1 | 9.6/17.4/12.3 | 3.2/6.6/4.3 | 4.4/11.0/6.3 | 8.9/22.6/12.8 |
| CCA | 21.8/7.6/11.3 | 31.4/15.1/20.4 | 43.0/34.2/38.1 | 28.0/8.4/13.0 | 40.0/17.1/23.9 | 51.9/36.1/42.6 | 37.5/7.0/11.7 | 49.8/15.6/23.7 | 65.2/36.9/47.2 |
| MLP | 10.5/0.8/1.5 | 23.9/2.6/4.6 | 16.3/1.7/3.0 | 19.0/1.0/1.9 | 22.2/1.5/2.8 | 31.5/4.5/7.9 | 23.7/0.7/1.3 | 33.4/1.8/3.5 | 50.9/8.1/13.9 |
| RCSLS | 36.5/1.7/3.3 | 33.6/3.2/5.8 | 40.7/9.8/15.8 | 38.0/1.4/2.7 | 37.7/3.0/5.6 | 47.5/9.7/16.1 | 37.8/1.2/2.3 | 44.0/2.6/5.0 | 54.2/7.8/13.7 |
| Proc | 25.7/8.7/13.0 | 36.2/17.5/23.6 | 47.7/36.2/41.1 | 33.1/10.0/15.3 | 46.6/19.8/27.8 | 59.1/40.4/48.0 | 42.7/8.3/13.9 | 59.7/19.6/29.5 | 71.8/42.4/53.3 |
| UGW | 9.4/0.6/1.2 | 9.2/0.6/1.2 | 10.2/0.7/1.3 | 9.1/0.5/0.9 | 9.2/0.5/0.9 | 9.1/0.5/0.9 | 14.0/0.5/0.9 | 14.3/0.5/0.9 | 14.1/0.5/0.9 |
| AO | 1.3/1.3/1.3 | 27.8/27.8/27.8 | 28.1/28.1/28.1 | 1.8/1.8/1.8 | 34.2/34.2/34.2 | 34.3/34.3/34.3 | 2.2/2.2/2.2 | 45.8/45.8/45.8 | 46.2/46.2/46.2 |
| GEH | **58.1/96.0/72.4** | **58.0/95.5/72.2** | **57.4/95.3/71.7** | **68.0/96.3/79.7** | **67.7/95.9/79.4** | **67.5/95.7/79.2** | **80.4/96.1/87.6** | **80.8/95.9/87.7** | **80.2/95.9/87.4** |

*Table 21.* **Vector linking on FiQA (`Mistral↔OpenAI`):** each cell reports **precision/recall/F1** (%). Best values per metric are **bolded**.

| Method | Overlap .15 | | | Overlap .20 | | | Overlap .30 | | |
|---|---|---|---|---|---|---|---|---|---|
| | Seeds 15 | Seeds 20 | Seeds 30 | Seeds 15 | Seeds 20 | Seeds 30 | Seeds 15 | Seeds 20 | Seeds 30 |
| Linear | 1.7/0.9/1.2 | 2.8/2.6/2.7 | 5.7/7.6/6.5 | 1.7/1.0/1.3 | 3.0/3.3/3.1 | 5.3/9.0/6.7 | 1.8/2.0/1.9 | 3.1/4.6/3.7 | 7.0/12.6/9.0 |
| CCA | 9.8/2.1/3.4 | 18.0/6.1/9.1 | 31.9/19.6/24.3 | 15.4/2.3/4.0 | 26.0/7.3/11.4 | 42.5/22.7/29.6 | 16.0/1.3/2.5 | 29.2/5.0/8.6 | 46.7/16.4/24.3 |
| MLP | 4.1/0.1/0.2 | 9.1/0.4/0.7 | 9.2/0.4/0.8 | 6.6/0.1/0.3 | 9.6/0.2/0.4 | 18.7/1.1/2.0 | 9.0/0.1/0.2 | 9.6/0.1/0.3 | 16.3/0.5/1.0 |
| RCSLS | 19.2/0.2/0.5 | 18.1/0.6/1.2 | 29.7/4.0/7.1 | 20.9/0.2/0.4 | 22.3/0.6/1.1 | 35.7/3.5/6.3 | 24.7/0.2/0.4 | 27.4/0.6/1.2 | 46.5/4.0/7.3 |
| Proc | 15.4/2.8/4.8 | 23.6/6.8/10.6 | 42.1/21.3/28.3 | 21.0/2.9/5.2 | 34.5/8.2/13.2 | 54.0/24.9/34.0 | 22.1/1.8/3.2 | 36.5/5.5/9.6 | 59.9/19.0/28.9 |
| UGW | 1.3/0.0/0.0 | 1.3/0.0/0.0 | 1.3/0.0/0.0 | 2.1/0.0/0.1 | 2.1/0.0/0.1 | 2.1/0.0/0.1 | 4.1/0.0/0.1 | 4.1/0.0/0.1 | 4.1/0.0/0.1 |
| AO | 0.2/0.2/0.2 | 0.3/0.3/0.3 | 0.4/0.4/0.4 | 0.1/0.1/0.1 | 0.2/0.2/0.2 | 0.3/0.3/0.3 | 0.1/0.1/0.1 | 44.7/44.7/44.7 | 0.4/0.4/0.4 |
| GEH | **60.2/92.8/73.1** | **60.7/92.8/73.4** | **60.6/93.2/73.5** | **70.1/93.8/80.3** | **70.4/93.7/80.4** | **70.0/93.7/80.2** | **82.9/94.2/88.2** | **82.9/94.1/88.1** | **82.7/94.0/88.0** |

*Table 22.* **Vector linking on NFCorpus (`Qwen↔KaLM`):** each cell reports **precision/recall/F1** (%). Best values per metric are **bolded**.

| Method | Overlap .15 | | | Overlap .20 | | | Overlap .30 | | |
|---|---|---|---|---|---|---|---|---|---|
| | Seeds 15 | Seeds 20 | Seeds 30 | Seeds 15 | Seeds 20 | Seeds 30 | Seeds 15 | Seeds 20 | Seeds 30 |
| Linear | 2.1/4.2/2.8 | 3.4/7.9/4.7 | 7.8/13.9/9.8 | 2.4/5.1/3.3 | 3.3/8.8/4.8 | 8.3/17.7/11.3 | 3.0/7.4/4.3 | 4.1/12.2/6.1 | 9.7/22.6/13.6 |
| CCA | 26.4/7.9/12.2 | 37.4/18.9/25.1 | 44.2/37.3/40.5 | 29.4/7.4/11.9 | 44.0/15.6/23.0 | 57.2/35.4/43.8 | 39.0/6.8/11.6 | 49.3/13.9/21.7 | 70.9/34.7/46.5 |
| MLP | 18.2/2.6/4.6 | 25.0/5.3/8.8 | 32.9/9.5/14.8 | 24.2/3.2/5.7 | 31.7/5.5/9.4 | 33.6/6.3/10.6 | 32.0/2.2/4.2 | 36.8/5.0/8.7 | 51.8/9.6/16.2 |
| RCSLS | 30.8/4.2/7.4 | 34.1/8.4/13.5 | 39.6/22.9/29.0 | 36.2/4.0/7.2 | 43.1/8.7/14.4 | 48.6/21.9/30.1 | 36.4/2.8/5.2 | 44.3/6.1/10.7 | 57.0/19.4/29.0 |
| Proc | 29.6/9.6/14.5 | 43.8/21.3/28.7 | 48.6/41.4/44.7 | 34.1/8.6/13.7 | 49.5/19.4/27.8 | 60.2/37.7/46.4 | 48.9/8.6/14.6 | 55.5/18.1/27.2 | 72.8/39.2/51.0 |
| UGW | 8.0/1.5/2.5 | 8.0/1.5/2.6 | 7.4/1.4/2.3 | 6.9/1.0/1.7 | 6.9/1.0/1.7 | 7.0/1.0/1.8 | 13.3/1.1/2.1 | 13.0/1.1/2.1 | 10.2/0.8/1.6 |
| AO | 6.2/6.2/6.2 | 24.6/24.6/24.6 | 27.2/27.2/27.2 | 6.1/6.1/6.1 | 21.0/21.0/21.0 | 32.9/32.9/32.9 | 42.3/42.3/42.3 | 42.1/42.1/42.1 | 43.0/43.0/43.0 |
| GEH | **59.1/89.4/71.2** | **59.1/90.3/71.4** | **57.6/89.7/70.2** | **67.7/87.9/76.5** | **65.7/87.6/75.1** | **65.1/86.4/74.2** | **79.9/89.4/84.4** | **79.4/88.4/83.7** | **79.7/88.6/83.9** |

*Table 23.* **Vector linking on SciDocs (Qwen↔KaLM):** each cell reports **precision/recall/F1** (%). Best values per metric are **bolded**.

| Method | Overlap .15 | | | Overlap .20 | | | Overlap .30 | | |
|---|---|---|---|---|---|---|---|---|---|
| | Seeds 15 | Seeds 20 | Seeds 30 | Seeds 15 | Seeds 20 | Seeds 30 | Seeds 15 | Seeds 20 | Seeds 30 |
| Linear | 1.2/0.6/0.8 | 1.5/1.4/1.4 | 2.9/3.8/3.3 | 1.0/0.6/0.7 | 1.3/1.2/1.2 | 3.3/4.3/3.7 | 0.9/0.7/0.8 | 1.6/2.0/1.8 | 3.8/6.7/4.9 |
| CCA | 11.6/2.0/3.5 | 20.9/6.4/9.8 | 34.7/19.0/24.5 | 12.9/1.8/3.2 | 23.4/5.2/8.6 | 38.9/16.0/22.6 | 15.8/2.0/3.5 | 30.2/4.9/8.5 | 52.3/17.2/25.9 |
| MLP | 8.9/0.4/0.7 | 12.8/0.7/1.3 | 21.4/1.9/3.6 | 13.9/0.3/0.6 | 20.6/0.7/1.4 | 25.9/1.2/2.2 | 14.8/0.4/0.7 | 22.5/0.8/1.6 | 23.2/0.7/1.4 |
| RCSLS | 31.0/0.5/0.9 | 15.4/1.6/2.8 | 25.9/6.0/9.8 | 30.5/0.4/0.8 | 19.7/1.0/1.8 | 29.9/5.2/8.8 | 20.6/0.4/0.8 | 22.8/1.3/2.5 | 39.9/5.8/10.1 |
| Proc | 16.8/2.4/4.2 | 27.8/7.5/11.8 | 44.8/22.1/29.6 | 17.6/2.5/4.4 | 30.9/6.3/10.5 | 51.5/20.9/29.7 | 24.9/2.7/4.8 | 41.3/6.4/11.1 | 63.5/20.3/30.8 |
| UGW | 0.0/0.0/0.0 | 0.0/0.0/0.0 | 0.0/0.0/0.0 | 0.0/0.0/0.0 | 0.0/0.0/0.0 | 0.0/0.0/0.0 | 0.0/0.0/0.0 | 0.0/0.0/0.0 | 0.0/0.0/0.0 |
| AO | 0.6/0.6/0.6 | 0.6/0.6/0.6 | 23.6/23.6/23.6 | 0.3/0.3/0.3 | 0.5/0.5/0.5 | 30.7/30.7/30.7 | 0.2/0.2/0.2 | 0.3/0.3/0.3 | 0.4/0.4/0.4 |
| GEH | **61.4/87.0/72.0** | **62.0/87.6/72.6** | **63.2/87.0/73.2** | **70.1/89.1/78.4** | **71.6/88.9/79.3** | **71.8/87.7/78.9** | **82.6/89.4/85.9** | **82.7/88.8/85.7** | **83.2/89.0/86.0** |

*Table 24.* **Vector linking on ArguAna (Qwen↔KaLM):** each cell reports **precision/recall/F1** (%). Best values per metric are **bolded**.

| Method | Overlap .15 | | | Overlap .20 | | | Overlap .30 | | |
|---|---|---|---|---|---|---|---|---|---|
| | Seeds 15 | Seeds 20 | Seeds 30 | Seeds 15 | Seeds 20 | Seeds 30 | Seeds 15 | Seeds 20 | Seeds 30 |
| Linear | 1.4/1.4/1.4 | 2.7/2.6/2.7 | 5.3/7.1/6.1 | 1.4/1.6/1.5 | 2.6/2.9/2.7 | 5.6/7.6/6.4 | 1.7/2.2/1.9 | 2.8/4.9/3.5 | 6.2/11.3/8.0 |
| CCA | 14.1/3.8/6.0 | 21.1/8.1/11.7 | 34.9/21.8/26.8 | 21.5/5.7/9.0 | 33.4/12.1/17.8 | 49.2/29.7/37.0 | 22.6/3.4/5.9 | 32.6/7.6/12.3 | 55.4/24.5/34.0 |
| MLP | 15.0/1.2/2.2 | 14.2/1.4/2.5 | 13.6/1.5/2.7 | 11.8/0.9/1.7 | 18.9/2.2/3.9 | 22.3/2.9/5.1 | 17.1/1.0/1.9 | 24.2/1.7/3.2 | 36.6/4.8/8.5 |
| RCSLS | 21.6/2.1/3.7 | 20.6/4.3/7.1 | 32.1/16.5/21.8 | 24.0/1.6/2.9 | 22.5/4.1/6.9 | 36.7/15.3/21.6 | 23.6/1.1/2.1 | 29.0/3.1/5.7 | 45.4/11.4/18.2 |
| Proc | 17.1/4.8/7.5 | 28.8/11.2/16.2 | 42.0/27.3/33.1 | 27.9/7.4/11.8 | 42.2/16.3/23.5 | 56.0/33.9/42.2 | 28.8/4.9/8.4 | 40.9/9.6/15.6 | 64.1/31.5/42.3 |
| UGW | 2.1/0.1/0.1 | 2.1/0.1/0.2 | 2.1/0.1/0.2 | 1.9/0.1/0.1 | 1.9/0.1/0.1 | 1.9/0.1/0.1 | 4.4/0.1/0.2 | 4.4/0.1/0.2 | 4.4/0.1/0.2 |
| AO | 8.9/8.9/8.9 | 27.4/27.4/27.4 | 26.6/26.6/26.6 | 32.5/32.5/32.5 | 32.9/32.9/32.9 | 33.4/33.4/33.4 | 10.1/10.1/10.1 | 44.8/44.8/44.8 | 44.7/44.7/44.7 |
| GEH | **53.8/81.1/64.7** | **54.0/79.9/64.4** | **53.4/79.6/63.9** | **64.3/81.0/71.7** | **64.7/79.8/71.4** | **63.4/79.9/70.7** | **75.1/80.8/77.9** | **76.0/79.6/77.7** | **75.9/80.0/77.9** |

*Table 25.* **Vector linking on SciFact (Qwen↔KaLM):** each cell reports **precision/recall/F1** (%). Best values per metric are **bolded**.

| Method | Overlap .15 | | | Overlap .20 | | | Overlap .30 | | |
|---|---|---|---|---|---|---|---|---|---|
| | Seeds 15 | Seeds 20 | Seeds 30 | Seeds 15 | Seeds 20 | Seeds 30 | Seeds 15 | Seeds 20 | Seeds 30 |
| Linear | 3.4/2.4/2.8 | 5.0/5.4/5.2 | 7.8/9.9/8.7 | 2.4/3.1/2.7 | 3.6/5.7/4.4 | 6.1/13.2/8.3 | 2.1/4.1/2.8 | 4.0/9.1/5.5 | 7.9/18.5/11.1 |
| CCA | 19.8/5.9/9.1 | 29.2/14.1/19.0 | 46.8/37.8/41.8 | 25.6/6.3/10.1 | 38.7/13.5/20.0 | 56.5/36.0/44.0 | 32.3/5.6/9.5 | 53.6/14.8/23.3 | 70.0/37.4/48.7 |
| MLP | 13.4/1.4/2.6 | 19.8/2.9/5.1 | 27.6/4.7/8.0 | 18.2/1.6/2.9 | 24.6/2.6/4.8 | 40.4/6.5/11.1 | 29.8/1.6/3.1 | 41.4/3.1/5.8 | 43.2/3.7/6.9 |
| RCSLS | 27.4/3.1/5.7 | 34.3/7.2/11.9 | 38.0/17.4/23.8 | 33.0/2.3/4.3 | 36.4/4.9/8.6 | 48.0/15.3/23.2 | 32.7/1.9/3.6 | 40.8/4.3/7.9 | 54.3/14.5/22.9 |
| Proc | 26.4/7.2/11.3 | 40.3/17.0/24.0 | 52.7/40.4/45.8 | 32.6/8.1/13.0 | 47.9/16.5/24.6 | 63.3/36.4/46.2 | 38.9/6.8/11.6 | 60.6/15.8/25.1 | 74.8/38.9/51.2 |
| UGW | 3.6/0.1/0.3 | 3.6/0.1/0.3 | 3.6/0.1/0.3 | 6.2/0.2/0.4 | 6.2/0.2/0.4 | 6.5/0.2/0.4 | 6.9/0.1/0.3 | 6.9/0.1/0.3 | 6.5/0.1/0.3 |
| AO | 22.0/22.0/22.0 | 24.3/24.3/24.3 | 27.4/27.4/27.4 | 7.5/7.5/7.5 | 9.6/9.6/9.6 | 32.6/32.6/32.6 | 12.9/12.9/12.9 | 5.3/5.3/5.3 | 45.3/45.3/45.3 |
| GEH | **62.0/96.5/75.5** | **61.7/96.0/75.1** | **59.7/96.1/73.7** | **71.4/95.7/81.8** | **71.4/95.7/81.8** | **70.2/95.8/81.1** | **83.2/96.0/89.2** | **83.2/95.8/89.1** | **82.1/95.9/88.5** |

*Table 26.* **Vector linking on FiQA (Qwen↔KaLM):** each cell reports **precision/recall/F1** (%). Best values per metric are **bolded**.

| Method | Overlap .15 | | | Overlap .20 | | | Overlap .30 | | |
|---|---|---|---|---|---|---|---|---|---|
| | Seeds 15 | Seeds 20 | Seeds 30 | Seeds 15 | Seeds 20 | Seeds 30 | Seeds 15 | Seeds 20 | Seeds 30 |
| Linear | 0.5/0.4/0.4 | 0.9/1.1/1.0 | 2.2/3.8/2.8 | 0.4/0.6/0.5 | 0.8/1.4/1.0 | 2.3/4.4/3.0 | 0.7/0.8/0.8 | 1.2/2.2/1.6 | 3.2/6.8/4.3 |
| CCA | 8.6/1.4/2.5 | 16.1/4.0/6.4 | 33.2/15.7/21.3 | 13.9/1.7/2.9 | 26.1/5.5/9.1 | 44.7/19.1/26.7 | 12.5/0.7/1.4 | 23.8/3.1/5.5 | 49.0/14.5/22.4 |
| MLP | 7.0/0.3/0.6 | 11.0/0.7/1.4 | 10.7/0.6/1.1 | 10.4/0.4/0.7 | 15.5/0.9/1.6 | 18.4/1.1/2.1 | 9.2/0.1/0.3 | 12.0/0.3/0.5 | 21.5/0.8/1.6 |
| RCSLS | 11.9/0.3/0.6 | 13.5/1.0/1.9 | 25.4/5.8/9.4 | 14.1/0.2/0.4 | 14.5/0.8/1.5 | 28.2/5.0/8.6 | 15.8/0.2/0.4 | 18.9/0.8/1.6 | 40.4/6.3/10.9 |
| Proc | 13.7/2.1/3.6 | 24.3/5.5/9.0 | 42.3/18.6/25.8 | 20.3/2.5/4.5 | 36.1/7.9/13.0 | 54.6/23.1/32.4 | 20.6/1.3/2.4 | 35.0/4.7/8.3 | 62.7/19.1/29.3 |
| UGW | 2.6/0.0/0.0 | 2.6/0.0/0.0 | 2.6/0.0/0.0 | 1.3/0.0/0.0 | 1.3/0.0/0.0 | 1.3/0.0/0.0 | 2.4/0.0/0.0 | 2.4/0.0/0.0 | 2.4/0.0/0.0 |
| AO | 0.2/0.2/0.2 | 0.2/0.2/0.2 | 23.4/23.4/23.4 | 0.3/0.3/0.3 | 0.2/0.2/0.2 | 30.1/30.1/30.1 | 0.1/0.1/0.1 | 0.1/0.1/0.1 | 0.3/0.3/0.3 |
| GEH | **55.7/75.9/64.2** | **57.8/74.6/65.1** | **57.8/73.6/64.8** | **66.5/76.9/71.3** | **67.0/76.2/71.3** | **66.7/72.1/69.3** | **79.8/79.9/79.8** | **79.3/76.6/77.9** | **79.2/75.7/77.4** |

*Table 27.* **Vector linking on NFCorpus (Qwen↔OpenAI):** each cell reports **precision/recall/F1** (%). Best values per metric are **bolded**.

| Method | Overlap .15 | | | Overlap .20 | | | Overlap .30 | | |
|---|---|---|---|---|---|---|---|---|---|
| | Seeds 15 | Seeds 20 | Seeds 30 | Seeds 15 | Seeds 20 | Seeds 30 | Seeds 15 | Seeds 20 | Seeds 30 |
| Linear | 2.3/4.4/3.0 | 3.7/8.8/5.2 | 9.4/15.7/11.8 | 2.1/4.6/2.9 | 3.5/9.8/5.2 | 9.4/18.8/12.6 | 2.8/7.2/4.0 | 5.4/13.6/7.7 | 10.8/25.4/15.2 |
| CCA | 31.5/11.7/17.1 | 40.2/21.9/28.4 | 46.7/42.9/44.7 | 35.8/10.2/15.9 | 48.7/21.1/29.4 | 59.2/43.8/50.3 | 46.7/10.6/17.3 | 59.9/19.6/29.5 | 73.7/43.8/54.9 |
| MLP | 16.4/3.2/5.4 | 24.8/7.6/11.7 | 30.1/13.4/18.6 | 15.7/2.2/3.9 | 26.7/6.1/9.9 | 41.4/15.6/22.7 | 36.1/3.3/6.0 | 31.9/5.0/8.6 | 51.0/9.9/16.6 |
| RCSLS | 32.0/5.5/9.3 | 39.9/10.5/16.7 | 41.8/27.6/33.2 | 35.8/4.3/7.7 | 40.3/8.8/14.5 | 49.9/24.7/33.1 | 38.9/3.0/5.6 | 43.6/7.2/12.4 | 59.7/23.5/33.7 |
| Proc | 36.8/15.5/21.8 | 43.8/24.0/31.0 | 51.4/49.1/50.2 | 44.5/13.8/21.0 | 52.3/23.6/32.6 | 60.4/47.5/53.2 | 52.5/11.8/19.3 | 62.2/22.8/33.4 | 75.0/45.5/56.6 |
| UGW | 11.7/3.6/5.5 | 11.7/3.6/5.5 | 12.3/3.7/5.7 | 14.5/3.5/5.6 | 14.5/3.5/5.7 | 14.5/3.6/5.8 | 14.8/2.4/4.2 | 13.7/2.2/3.9 | 15.3/2.5/4.2 |
| AO | 12.8/12.8/12.8 | 23.1/23.1/23.1 | 27.9/27.9/27.9 | 13.4/13.4/13.4 | 31.5/31.5/31.5 | 33.3/33.3/33.3 | 22.5/22.5/22.5 | 43.9/43.9/43.9 | 44.5/44.5/44.5 |
| GEH | **61.9/93.2/74.4** | **59.8/93.5/73.0** | **58.3/95.5/72.4** | **68.7/94.1/79.4** | **68.9/94.3/79.6** | **68.4/95.1/79.5** | **82.1/95.6/88.3** | **82.4/95.2/88.3** | **81.0/94.8/87.4** |

*Table 28.* **Vector linking on SciDocs (Qwen↔OpenAI):** each cell reports **precision/recall/F1** (%). Best values per metric are **bolded**.

| Method | Overlap .15 | | | Overlap .20 | | | Overlap .30 | | |
|---|---|---|---|---|---|---|---|---|---|
| | Seeds 15 | Seeds 20 | Seeds 30 | Seeds 15 | Seeds 20 | Seeds 30 | Seeds 15 | Seeds 20 | Seeds 30 |
| Linear | 1.2/0.9/1.0 | 2.0/2.0/2.0 | 4.3/5.1/4.7 | 1.2/0.7/0.9 | 1.7/1.5/1.6 | 4.3/6.0/5.0 | 1.0/1.0/1.0 | 2.0/2.8/2.3 | 4.8/8.8/6.2 |
| CCA | 8.9/1.8/3.0 | 14.8/5.3/7.9 | 29.4/16.6/21.2 | 11.8/1.6/2.8 | 21.1/5.1/8.1 | 36.8/16.2/22.5 | 14.4/1.9/3.4 | 28.6/5.2/8.9 | 52.6/18.3/27.1 |
| MLP | 3.4/0.3/0.6 | 6.1/0.8/1.4 | 15.7/2.4/4.1 | 10.7/0.5/0.9 | 15.0/1.2/2.2 | 24.9/3.2/5.7 | 15.2/0.5/0.9 | 24.1/1.4/2.7 | 28.8/1.6/3.0 |
| RCSLS | 14.6/0.8/1.5 | 16.0/2.0/3.5 | 27.2/7.9/12.2 | 15.1/0.5/1.0 | 16.7/1.2/2.2 | 32.7/6.9/11.4 | 18.2/0.6/1.1 | 24.7/1.7/3.1 | 43.4/7.8/13.2 |
| Proc | 12.1/2.0/3.4 | 21.9/6.4/9.9 | 41.8/20.6/27.6 | 15.0/2.0/3.5 | 29.6/6.6/10.8 | 51.1/21.0/29.8 | 24.5/2.9/5.2 | 39.8/7.3/12.4 | 65.5/22.5/33.5 |
| UGW | 6.2/0.1/0.1 | 6.5/0.1/0.1 | 6.5/0.1/0.1 | 0.0/0.0/0.0 | 0.0/0.0/0.0 | 0.0/0.0/0.0 | 0.0/0.0/0.0 | 0.0/0.0/0.0 | 0.0/0.0/0.0 |
| AO | 0.5/0.5/0.5 | 8.1/8.1/8.1 | 24.4/24.4/24.4 | 10.6/10.6/10.6 | 0.7/0.7/0.7 | 0.8/0.8/0.8 | 0.3/0.3/0.3 | 0.3/0.3/0.3 | 0.6/0.6/0.6 |
| GEH | **63.2/90.9/74.6** | **64.2/91.3/75.4** | **63.7/90.4/74.7** | **71.9/90.9/80.3** | **72.0/90.6/80.3** | **72.6/90.6/80.6** | **83.0/91.9/87.2** | **83.3/91.0/87.0** | **83.7/91.1/87.2** |

*Table 29.* **Vector linking on ArguAna (Qwen↔OpenAI):** each cell reports **precision/recall/F1** (%). Best values per metric are **bolded**.

| Method | Overlap .15 | | | Overlap .20 | | | Overlap .30 | | |
|---|---|---|---|---|---|---|---|---|---|
| | Seeds 15 | Seeds 20 | Seeds 30 | Seeds 15 | Seeds 20 | Seeds 30 | Seeds 15 | Seeds 20 | Seeds 30 |
| Linear | 1.4/1.8/1.6 | 3.2/4.4/3.7 | 7.1/9.3/8.0 | 1.9/2.7/2.2 | 3.0/5.3/3.8 | 7.7/12.8/9.6 | 1.9/3.1/2.4 | 3.1/7.1/4.3 | 7.1/15.3/9.7 |
| CCA | 18.2/5.6/8.6 | 30.8/12.7/18.0 | 41.8/30.2/35.1 | 25.5/6.8/10.7 | 36.6/14.1/20.4 | 54.1/34.1/41.8 | 31.7/5.9/10.0 | 40.8/12.3/18.9 | 63.9/34.3/44.6 |
| MLP | 11.8/1.5/2.6 | 19.7/2.9/5.0 | 22.7/4.4/7.4 | 13.7/1.5/2.6 | 21.7/3.2/5.6 | 36.9/9.2/14.7 | 14.0/1.0/1.8 | 27.3/2.2/4.1 | 43.5/6.3/11.1 |
| RCSLS | 21.7/2.4/4.3 | 25.6/5.2/8.6 | 36.1/18.2/24.2 | 22.2/2.2/3.9 | 27.6/5.1/8.7 | 46.4/18.0/26.0 | 27.1/1.4/2.7 | 36.7/4.9/8.6 | 55.1/15.6/24.3 |
| Proc | 23.1/7.5/11.4 | 38.7/17.7/24.3 | 47.4/37.6/41.9 | 34.6/9.8/15.2 | 47.3/19.4/27.5 | 59.8/40.3/48.2 | 39.4/7.2/12.2 | 54.6/17.3/26.3 | 70.4/41.7/52.4 |
| UGW | 2.3/0.1/0.2 | 2.3/0.1/0.2 | 2.3/0.1/0.2 | 0.0/0.0/0.0 | 0.0/0.0/0.0 | 0.0/0.0/0.0 | 2.2/0.0/0.1 | 2.2/0.0/0.1 | 2.2/0.0/0.1 |
| AO | 3.3/3.3/3.3 | 14.9/14.9/14.9 | 28.6/28.6/28.6 | 1.7/1.7/1.7 | 22.8/22.8/22.8 | 34.3/34.3/34.3 | 20.5/20.5/20.5 | 22.5/22.5/22.5 | 46.1/46.1/46.1 |
| GEH | **55.2/87.3/67.6** | **56.2/86.0/68.0** | **55.3/85.8/67.3** | **65.6/87.3/74.9** | **66.2/87.3/75.3** | **65.0/87.6/74.6** | **77.1/87.7/82.1** | **77.7/86.9/82.0** | **78.0/86.4/82.0** |

*Table 30.* **Vector linking on SciFact (Qwen↔OpenAI):** each cell reports **precision/recall/F1** (%). Best values per metric are **bolded**.

| Method | Overlap .15 | | | Overlap .20 | | | Overlap .30 | | |
|---|---|---|---|---|---|---|---|---|---|
| | Seeds 15 | Seeds 20 | Seeds 30 | Seeds 15 | Seeds 20 | Seeds 30 | Seeds 15 | Seeds 20 | Seeds 30 |
| Linear | 2.6/3.0/2.7 | 4.5/6.0/5.2 | 8.2/11.7/9.6 | 2.3/3.8/2.9 | 4.1/7.4/5.2 | 8.4/15.8/11.0 | 2.0/4.5/2.8 | 4.2/9.2/5.8 | 10.8/21.1/14.3 |
| CCA | 27.4/8.4/12.9 | 30.8/14.3/19.5 | 46.7/38.0/41.9 | 28.6/8.0/12.5 | 42.8/16.3/23.6 | 56.1/40.0/46.7 | 32.5/6.2/10.5 | 52.1/16.4/25.0 | 67.6/38.0/48.7 |
| MLP | 18.7/2.2/4.0 | 23.7/6.2/9.8 | 28.7/11.4/16.3 | 23.3/4.9/8.1 | 28.3/7.7/12.1 | 41.3/16.0/23.1 | 24.1/2.1/3.9 | 37.5/6.1/10.5 | 48.7/13.4/21.1 |
| RCSLS | 30.9/3.6/6.4 | 32.7/7.9/12.7 | 41.2/22.5/29.1 | 26.9/2.7/4.9 | 35.5/6.4/10.9 | 48.5/21.3/29.6 | 30.7/2.4/4.5 | 40.5/5.9/10.3 | 57.7/20.3/30.0 |
| Proc | 26.5/7.6/11.8 | 36.1/15.8/22.0 | 50.8/41.2/45.5 | 34.5/9.8/15.2 | 53.8/20.3/29.4 | 65.2/43.3/52.0 | 42.3/7.9/13.3 | 61.4/18.1/27.9 | 74.5/42.4/54.1 |
| UGW | 3.1/0.3/0.5 | 1.6/0.1/0.2 | 1.6/0.1/0.2 | 5.1/0.3/0.6 | 5.1/0.3/0.6 | 5.1/0.3/0.6 | 8.5/0.3/0.6 | 8.6/0.3/0.6 | 8.6/0.3/0.6 |
| AO | 4.5/4.5/4.5 | 15.1/15.1/15.1 | 27.7/27.7/27.7 | 5.2/5.2/5.2 | 13.3/13.3/13.3 | 34.3/34.3/34.3 | 29.1/29.1/29.1 | 46.4/46.4/46.4 | 47.5/47.5/47.5 |
| GEH | **63.5/96.2/76.5** | **64.9/95.9/77.4** | **62.3/95.6/75.4** | **72.8/96.7/83.1** | **73.2/96.3/83.1** | **70.9/96.6/81.8** | **83.8/96.9/89.9** | **84.0/96.9/90.0** | **83.9/96.4/89.7** |

*Table 31.* **Vector linking on FiQA (Qwen↔OpenAI):** each cell reports **precision/recall/F1** (%). Best values per metric are **bolded**.

| Method | Overlap .15 | | | Overlap .20 | | | Overlap .30 | | |
|---|---|---|---|---|---|---|---|---|---|
| | Seeds 15 | Seeds 20 | Seeds 30 | Seeds 15 | Seeds 20 | Seeds 30 | Seeds 15 | Seeds 20 | Seeds 30 |
| Linear | 1.1/0.6/0.8 | 1.7/1.5/1.6 | 3.8/4.9/4.3 | 1.3/0.6/0.8 | 1.9/1.7/1.8 | 3.9/5.5/4.5 | 1.2/1.3/1.3 | 2.1/3.1/2.5 | 5.1/8.9/6.5 |
| CCA | 5.9/1.1/1.9 | 10.3/2.9/4.5 | 24.5/12.7/16.8 | 9.9/1.4/2.5 | 18.1/4.3/6.9 | 34.9/15.8/21.8 | 9.3/0.7/1.3 | 20.7/3.1/5.4 | 38.1/11.0/17.1 |
| MLP | 2.3/0.1/0.3 | 5.1/0.4/0.8 | 10.0/1.0/1.8 | 4.1/0.2/0.5 | 8.8/0.7/1.3 | 19.3/2.6/4.6 | 3.8/0.1/0.2 | 9.7/0.4/0.7 | 17.2/1.0/1.9 |
| RCSLS | 6.5/0.3/0.6 | 10.2/1.1/2.0 | 21.5/5.7/9.1 | 7.9/0.3/0.6 | 10.9/0.8/1.5 | 24.4/4.7/7.8 | 10.7/0.3/0.6 | 16.1/1.0/1.9 | 32.6/5.2/9.0 |
| Proc | 10.4/1.8/3.1 | 17.5/4.7/7.4 | 37.0/17.2/23.5 | 15.8/2.1/3.7 | 26.3/5.9/9.6 | 48.6/20.4/28.7 | 17.0/1.3/2.4 | 32.0/4.5/7.8 | 53.0/14.8/23.2 |
| UGW | 0.0/0.0/0.0 | 2.2/0.0/0.0 | 2.1/0.0/0.0 | 0.0/0.0/0.0 | 0.0/0.0/0.0 | 0.0/0.0/0.0 | 0.0/0.0/0.0 | 0.0/0.0/0.0 | 0.0/0.0/0.0 |
| AO | 0.2/0.2/0.2 | 0.3/0.3/0.3 | 0.4/0.4/0.4 | 0.2/0.2/0.2 | 0.2/0.2/0.2 | 0.4/0.4/0.4 | 0.1/0.1/0.1 | 43.0/43.0/43.0 | 43.6/43.6/43.6 |
| GEH | **54.5/70.3/61.4** | **55.1/68.9/61.3** | **56.8/68.8/62.2** | **65.3/69.9/67.5** | **66.1/70.2/68.1** | **66.0/69.7/67.8** | **77.9/70.8/74.2** | **79.0/71.5/75.1** | **78.0/70.3/74.0** |

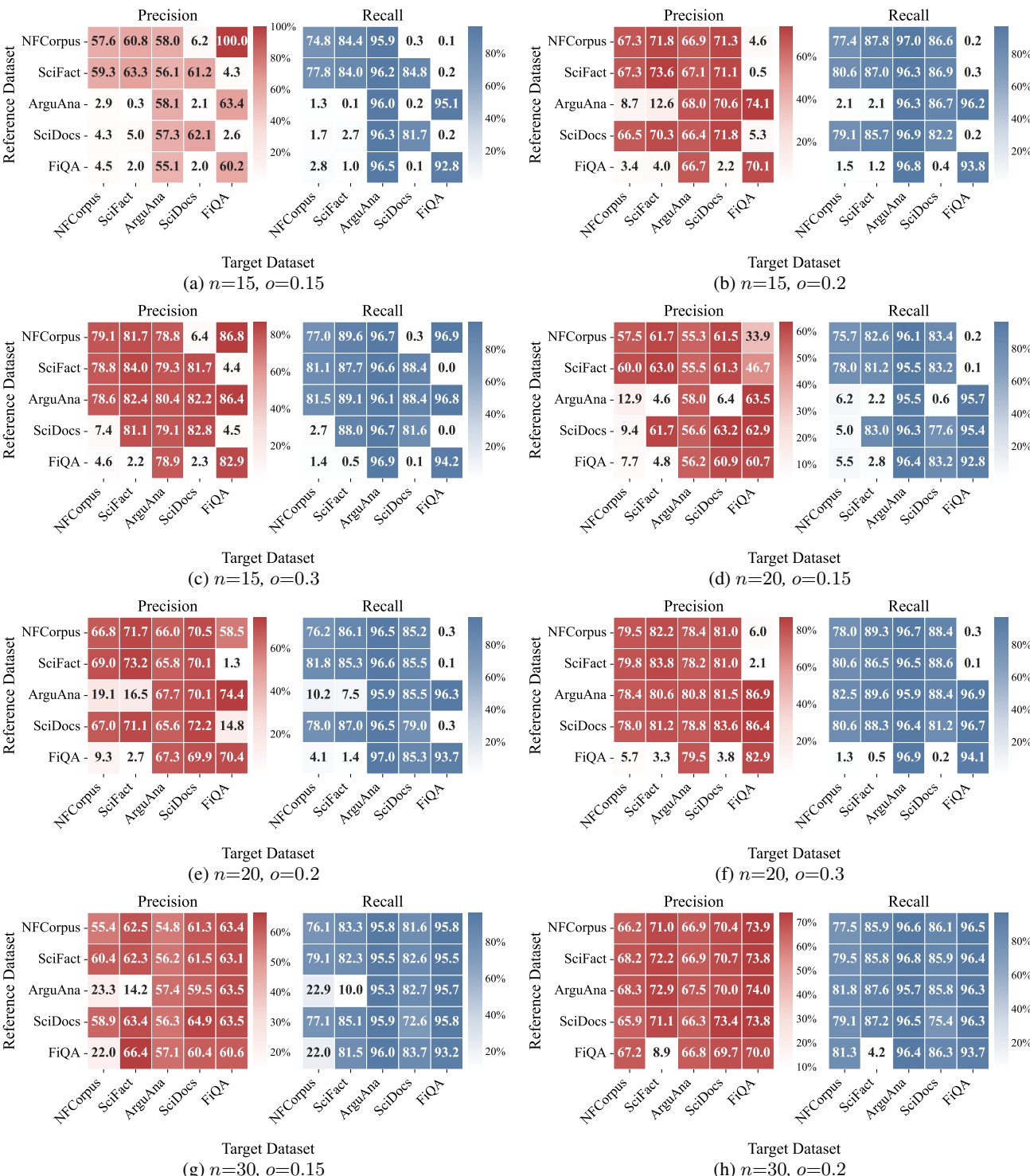

*Figure 11.* **Out-of-domain reference transfer (additional settings):**Accuracy (left) and recall (right) on five target datasets (columns) when seeds are drawn from an out-of-domain reference dataset (rows). Each panel varies the number of seeds $n$ and target overlap $o$. The main text reports the case $n$=30, $o$=0.3 (Fig. 5).

*Table 32.* **Cross-model clustering datasets from MTEB (test split metadata).**Lengths are measured in characters per title.

| Dataset | #Titles | #Clusters | Avg. len. | Min/Max len. |
|---|---|---|---|---|
| RedditClustering.v2 (Geigle et al., 2021) | 2048 | 50 | 65.49 | 18 / 299 |
| StackExchangeClustering.v2 (Geigle et al., 2021) | 2048 | 121 | 57.51 | 19 / 148 |

return the integrated database $T(D_1) \cup D_2$. Let $Q$ be the benchmark query set, and for each query $q \in Q$ let $\text{ans}_q$ denote its ground-truth relevant set. Let $\text{top-}k(q)$ be the top-$k$ results returned by searching the integrated database with $\text{emb}_2(q)$. We report:

$$\text{Recall@}k \; = \; \frac{1}{|Q|} \sum_{q \in Q} \frac{|\text{ans}_q \cap \text{top-}k(q)|}{|\text{ans}_q|}.$$

For rank-sensitive evaluation, we also report NDCG:

$$\text{NDCG@}k \; = \; \frac{\text{DCG@}k}{\text{IDCG@}k}, \qquad \text{DCG@}k \; = \; \sum_{i=1}^{k} \frac{\text{rel}_i}{\log_2(i+1)},$$

where $\text{rel}_i$ is the graded relevance of the item at rank $i$ and $\text{IDCG@}k$ is the DCG of the ideal ranking.

We use a FAISS GPU index with inner-product search over $\ell_2$-normalized embeddings (equivalently cosine similarity).

### D.1.2. GLOBAL CROSS-MODEL CLUSTERING

We evaluate cross-model clustering using two clustering benchmarks from MTEB (Enevoldsen et al., 2025). Both datasets consist of short *titles* and provide gold cluster labels (e.g., subreddit or StackExchange community). Dataset statistics are summarized in Table 32.

For a dataset with texts $\{t_\ell\}_{\ell=1}^N$ and gold labels $\{y_\ell\}$, we generate two embedding sets $\mathbf{E}_1 = \{e_\ell^{(1)}\}$ and $\mathbf{E}_2 = \{e_\ell^{(2)}\}$ using two embedding models (e.g., Qwen and KaLM). We then create a partial-overlap partition by selecting index sets $\mathcal{I}_1, \mathcal{I}_2 \subseteq [N]$ such that $\mathcal{I}_\cap = \mathcal{I}_1 \cap \mathcal{I}_2$ contains the shared items and $\mathcal{I}_1 \setminus \mathcal{I}_2$, $\mathcal{I}_2 \setminus \mathcal{I}_1$ are model-specific items. The ground-truth correspondence set is $\mathcal{P}^\star = \{(\ell, \ell) : \ell \in \mathcal{I}_\cap\}$; predicted correspondences $\hat{\mathcal{P}}$ are produced by a linking method.

**Graph construction.** For each embedding space, we build a $k$-NN graph $G_1$ on $\{e_\ell^{(1)} : \ell \in \mathcal{I}_1\}$ and $G_2$ on $\{e_\ell^{(2)} : \ell \in \mathcal{I}_2\}$ (cosine similarity). We choose $k$ adaptively to ensure connectivity, by increasing $k$ until the largest connected component covers at least 95% of nodes. We then form a unified graph $G$ by merging each correspondence pair $(\ell_1, \ell_2) \in \hat{\mathcal{P}}$ into a single super-node that inherits the incident edges from both $G_1$ and $G_2$. When no correspondences are provided, $G$ is simply the disjoint union of $G_1$ and $G_2$.

**Clustering.** We apply Leiden community detection on $G$. To make comparisons fair across methods, we use the same graph-based clustering pipeline throughout and tune the Leiden resolution by binary search to match the known number of gold clusters in the evaluated set.

We report : (i) **Full-space** single-model clustering on each complete embedding space independently (Qwen and KaLM), serving as optimal references; (ii) **Concat**, which zero-pads embeddings to a common dimension and concatenates them without using any cross-space correspondences; (iii) **Seed**, which stitches the two $k$-NN graphs by node-merging using only ground-truth seed correspondences; and (iv) **Ours**, which performs the same node-merging procedure using predicted correspondences $\hat{\mathcal{P}}$ from GEH.

### D.2. Experimental Results on Cross Model Clustering

We report cross-model clustering results for Qwen $\leftrightarrow$ KaLM at overlap ratios $\alpha \in \{0.2, 0.3\}$ and seed budgets $n \in \{20, 30\}$. We evaluate clustering quality using V-measure, NMI, and ARI, and additionally report *Overlap Agreement Rate* (OAR), defined as the fraction of overlapped items whose two embeddings (one from each space) are assigned to the same community in the unified clustering. OAR is not reported for naive concatenation since it produces a disjoint union of the two graphs and does not induce cross-space communities. As shown in Table 33, using only seed correspondences yields limited cross-space connectivity and suboptimal global coherence, whereas using GEH to stitch the graphs achieves high cross-space coupling (OAR = 75–98%) and recovers cluster quality within $\approx 1\%$ of single-space performance.

*Table 33.* **Cross-model clustering performance for Qwen↔KaLM embeddings.** Each cell reports V-measure / NMI / ARI (%). OAR = Overlap Agreement Rate (%). **Bold** indicates best per metric among Concat/Seed/OURS for each configuration.

| Dataset | Overlap | Seed | Concat | V / NMI / ARI Seed | OURS | Seed | OAR OURS | V / NMI / ARI Qwen | KaLM |
|---------|---------|------|--------|------|------|------|------|------|------|
| Reddit | 0.2 | 20 | 52.8/52.8/20.7 | 53.4/53.4/20.8 | **62.8/62.8/37.6** | 6.6 | **83.2** | | |
| | | 30 | 52.8/52.8/20.7 | 54.0/54.0/22.3 | **63.0/63.0/38.0** | 10.5 | **84.6** | 63.5/63.5/39.8 | 65.6/65.6/42.7 |
| | 0.3 | 20 | 51.7/51.7/19.2 | 53.6/53.6/21.6 | **66.5/66.5/44.6** | 4.6 | **96.9** | | |
| | | 30 | 51.7/51.7/19.2 | 53.9/53.9/22.1 | **67.1/67.1/45.6** | 7.7 | **97.6** | | |
| StackEx | 0.2 | 20 | 62.2/62.2/19.2 | 62.5/62.5/20.2 | **67.0/67.0/28.9** | 7.1 | **75.9** | | |
| | | 30 | 62.2/62.2/19.2 | 62.4/62.4/20.2 | **67.4/67.4/30.2** | 9.8 | **88.8** | 68.7/68.7/30.7 | 68.8/68.8/31.1 |
| | 0.3 | 20 | 61.8/61.8/18.9 | 61.8/61.8/19.6 | **67.5/67.5/31.2** | 5.9 | **78.8** | | |
| | | 30 | 61.8/61.8/18.9 | 62.1/62.1/20.1 | **68.3/68.3/32.7** | 8.3 | **91.9** | | |

