# OpenReview forum: "Vector Linking via Cross-Model Local Isometric Consistency"
_ICML.cc/2026/Conference — ICML 2026 regular_

### Official Review · Reviewer_2hBy · 2026-03-02

**Soundness:** 3
**Presentation:** 3
**Significance:** 3
**Originality:** 3
**Overall Recommendation:** 4
**Confidence:** 3

**Summary:**

This paper studies the problem of vector linking, which aims to recover object correspondences between embedding spaces produced by different black-box encoders under partial and unknown overlap. Unlike classical embedding alignment methods that assume a global mapping between spaces, the authors argue that independently trained contrastive encoders preserve local geometric consistency while exhibiting significant global distortion.Based on this observation, the paper proposes a geometric embedding hashing (GEH) framework that represents vectors using distance-to-anchor signatures and iteratively expands correspondence sets through multi-view hashing and posterior-guided bootstrapping. Experiments across multiple datasets and embedding model pairs demonstrate strong performance improvements over adapted alignment baselines, using only a small number of seed correspondences.

**Compliance With Llm Reviewing Policy:**

Affirmed.

**Final Justification:**

I think the assumptions upon which this paper relies are too idealistic and may be difficult to accept in real-world scenarios.  However, for other aspects of the paper, I keep the score unchanged.

**Key Questions For Authors:**

Strengths And Weaknesses

**Limitations:**

No, the authors should add that.

**Strengths And Weaknesses:**

# Strengths
1. Novel and well-motivated problem formulationThe paper introduces vector linking as a distinct problem from traditional embedding alignment. The formulation is clearly defined and practically relevant, especially for scenarios involving evolving embedding models or heterogeneous vector databases.
2. Insightful geometric observationA central contribution is the empirical and theoretical observation that contrastive encoders exhibit local isometric consistency across models. This insight provides a compelling conceptual explanation for why correspondence recovery may be feasible without accessing raw data or model parameters.
3. Strong empirical performanceExperiments demonstrate substantial improvements over several alignment-based baselines across multiple datasets and encoder pairs. The method appears robust to small seed sizes and shows promising results under out-of-domain anchor settings.
# Weaknesses
1. Theoretical analysis relies on some assumptionsThe theoretical justification assumes locally optimal contrastive encoders under manifold smoothness and isotropic augmentation conditions. While the analysis provides useful intuition, it does not fully establish guarantees under realistic encoder behavior. As
presented, the theory primarily explains plausibility rather than rigorously characterizing when vector linking should succeed or fail.
2. Limited diversity of experimental settingsAll experiments focus on text embedding models trained with contrastive objectives. Since the main claim concerns cross-model geometric consistency, evaluation on additional modalities (e.g., vision or multimodal embeddings)or non-contrastive representations would strengthen the generality of the conclusions.
3. Computational efficiency and resource usage are insufficiently evaluated.While the paper briefly discusses asymptotic complexity, it does not provide concrete runtime measurements or computational resource comparisons with competing methods. Given that the proposed approach involves multi-view hashing and iterative bootstrapping over entire embedding collections, empirical analysis of time cost and scalability would be important to support the practicality claims.

---

> ### Author Rebuttal · Authors · 2026-03-30
>
> We thank the reviewer for the positive assessment and for recognizing the novelty and practical relevance of vector linking. We also appreciate the concrete concerns regarding theory, experimental scope, and efficiency. We address them below.
>
> **W1: Theory**
>
> We agree that Theorem 1 is not intended as a full worst-case guarantee for arbitrary production encoders. Its role is narrower and more targeted: to explain why *local scaled isometry* is plausible for independently trained *contrastive encoders*, and therefore why a *scale-free local geometric signature* is the right ingredient for vector linking across partially overlapping spaces. This is also why our empirical study focuses on contrastive encoders.
>
> GEH is then designed precisely for the *non-ideal regime* where these assumptions hold only approximately. In particular, GEH "implements" the theory in practice by: (a) normalization and kernelization to suppress long-range distortions, where consistency is weakest; and (b) multi-view voting with iterative anchor expansion to concentrate evidence on short-range distances, where cross-encoder consistency is strongest despite noise. Further, Appendix A.3 already relaxes constant isotropy to point-dependent $c(x)$, yielding a point-dependent local scale $\kappa(x)$.
>
> We will clarify more explicitly in the revision that Theorem 1 is intended as a *geometric explanation and motivation* for local geometric hashing, not as an end-to-end guarantee for the full vector-linking problem.
>
> **W2: Evaluation generality**
>
> We agree that testing additional modalities or non-contrastive representations would broaden the scope. However, our current empirical scope is intentionally aligned with the paper’s main claim and target application: a geometric property of *contrastive encoders* with motivating application to vector database interoperability. Modern text embeddings are currently the most relevant and widely deployed family in that setting.
>
> Within that intended scope, the current diversity is already substantial: five model pairs, five domains, proprietary and open-weight encoders, heterogeneous embedding dimensions (1024, 1536, 3584, 3840, 4096), out-of-domain anchors, and downstream tasks beyond retrieval (database integration and cross-model clustering).
>
> To further stress-test generality within the target family, we added a new low-dimensional experiment: OpenAI (256d) vs. Mistral (1024d) on SciFact. Local consistency remains clearly present:
>
> | k | 1 | 10 | 50 | 100 |
> |---|---|----|----|-----|
> | Mean Jaccard | 0.68 | 0.320 | 0.283 | 0.280 |
>
> And GEH remains much stronger than alignment baselines for full linking:
>
> | Method | Precision | Recall |
> |--------|-----------|--------|
> | Proc   | 4.5 | 3.9 |
> | TPS    | 4.6 | 4.1 |
> | RCSLS  | 33.0 | 5.9 |
> | GEH    | 71.5 | 66.0 |
>
> This suggests that the phenomenon is not tied to large ambient dimension. That said, we agree that extensions to vision/multimodal or non-contrastive representations are important next steps, and we will state this explicitly as a limitation/future direction.
>
> **W3: Efficiency/scalability**
>
> As suggested, we have now added explicit runtime/scalability tests.
>
> On FEVER (5.42 million corpus), Mistral↔OpenAI, overlap $\alpha=0.3$, $|S|=30$, using a single A100 80GB GPU, we obtain:
>
> | Method | Precision (%) | Recall (%) | Wall-clock runtime (s) |
> |---------|---------------|------------|------------------------|
> | Procrustes | 9.03 | 0.73 | 4494 |
> | TPS        | 9.06 | 0.73 | 4517 |
> | CCA        | 5.22 | 0.61 | 1613 |
> | MLP        | 1.93 | 0.01 | 4414 |
> | Linear     | 4.13 | 0.00 | 4420 |
> | RCSLS      | 6.40    | 0.11    | 3348    |
> | **GEH**      | **93.8** | **68.9** |  **3328** |
>
> These results show that (a) GEH is practical at million scale on a single GPU, remains within the same order of magnitude as the fastest baseline (CCA), and is faster than Procrustes/TPS/MLP/Linear/RCSLS; (b) the accuracy gap is *very large*: compared with CCA, GEH improves precision from 5.22% to 93.8% and recall from 0.61% to 68.9%, while remaining only about 2.1× slower in wall-clock time. The quality gaps relative to the slower baselines are similarly large.
>
> The dominant cost in vector linking is CSLS/MNN-based link extraction, which all alignment baselines also require after fitting a mapping. GEH's extra cost comes from evaluating multiple views, but each view operates in low-dimensional distance-to-anchor hash space induced by a small anchor set, rather than repeatedly matching in the original embedding space. Thus the iterative procedure adds many moderate-cost hash-space retrieval steps, making GEH practical at million-scale while significantly improving accuracy.
>
>
> $\newline$
>
> Finally, as suggested, we will add an explicit *Limitations* discussion, covering the current focus on contrastive text encoders, the explanatory rather than worst-case role of Theorem 1, and the need to validate beyond text and beyond contrastive training.

---

> > ### Author Rebuttal · Reviewer_2hBy · 2026-04-01
> >
> > This resolved most of my questions. I'm maintaining my rating. My final rating will take into account the reviewer's suggestions who gave a negative rating.

---

### Official Review · Reviewer_5oZ9 · 2026-03-07

**Soundness:** 2
**Presentation:** 2
**Significance:** 2
**Originality:** 3
**Overall Recommendation:** 3
**Confidence:** 4

**Summary:**

This paper introduces a novel problem: vector linking, i.e., recovering correspondences between two black-box embedding clouds produced by different encoders, under partial and unknown overlap. The authors empirically observe that independently trained contrastive encoders exhibit local isometric consistency: short-range distances are preserved across models, while long-range distances are not. They provide a theoretical justification for this phenomenon based on a localized analysis of contrastive learning objectives. Leveraging this insight, they propose a multi-view geometric hashing algorithm (GEH) that uses distance-to-anchor signatures and a posterior-guided bootstrapping mechanism to iteratively expand a small set of seed correspondences. Experiments on five BEIR datasets and multiple encoder pairs show that GEH significantly outperforms adapted alignment baselines.

**Compliance With Llm Reviewing Policy:**

Affirmed.

**Key Questions For Authors:**

see the weaknesses.

**Limitations:**

--

**Strengths And Weaknesses:**

+The paper identifies a practical and under-explored problem—linking embeddings from different black-box encoders without access to raw data or model internals. This has clear relevance to real-world scenarios like integrating vector databases from different providers.

+The observation of local distance consistency across models (Figure 1 and Appendix A.5) is both surprising and practically useful.

+The paper provides a theoretical justification (Theorem 1) linking local optimality in contrastive learning to local isometry, grounding the empirical observation in first principles.

-Complexity and Practicality: The GEH algorithm is quite complex, with multiple hyperparameters (view scheduling, FPS vs. random, kernel bandwidth, CSLS neighborhood size, Beta prior, Otsu thresholding). While the paper includes some sensitivity analysis (Appendix C.2.4), the method may be difficult to tune in practice without extensive experimentation.

-Limited Comparison to Simple Baselines: The baselines are adapted from alignment methods, but a simpler approach—e.g., using the seed anchors to train a local distance-preserving mapping (like a thin-plate spline) and then performing MNN—is not explored. It is unclear whether the complexity of GEH is necessary to achieve the reported gains.

-Theoretical Assumptions: Theorem 1 relies on several assumptions (smooth manifold, local optimality, isotropic augmentation) that may not hold in practice. While the authors relax the isotropy assumption in Appendix A.3, the overall theoretical framework is still quite idealized.

-Scalability Concerns: The algorithm involves repeated hashing and nearest-neighbor search across multiple views. While the authors claim moderate cost (Appendix C.2), no runtime comparisons are provided. For large-scale databases (millions of vectors), the iterative multi-view approach could become prohibitively expensive.

-Lack of Ablation on Key Components: The ablation study (Section 5.3) compares FPS vs. random and kernelized vs. raw distances, but does not ablate other critical components like the Beta-Bernoulli posterior, Otsu thresholding, or the view scheduling rule. It is unclear which parts of the algorithm are most essential.

---

> ### Author Rebuttal · Authors · 2026-03-30
>
> We appreciate reviewer's concerns regarding complexity, baseline, scalability, and ablation. We provide evidences and targeted tests that directly address these points.
>
> **W1/W5. Complexity/practicality/ablation**
>
> GEH is less tunable than it may appear: Otsu is parameter-free, the Beta prior is fixed to Beta(1,1), bandwidth uses the median heuristic, and FPS is fixed rather than tuned. The main remaining knobs are the schedule and $k_{\text{CSLS}}$, for which our existing sensitivity already shows broad stable ranges (Appendix C.2.4). Thus GEH does not rely on delicate tuning.
>
> We also added the requested staged ablation:
>
> A0 = seeds-only (1 iter, raw votes, fixed threshold);
>
> A1 = A0 + multi-view voting with iterative anchor expansion
>
> A2 = A1 + Beta-Bernoulli posterior;
>
> A3 = A2 + adaptive schedule with Otsu  = full GEH.
>
> On FiQA / SciDocs / SciFact / ArguAna with $|S|=15$, $\alpha=0.3$, the average F1 evolves as:
> A0 = 1.6 → A1 = 68.3 → A2 = 73.7 →  A3 = 85.9.
>
> Thus, even a stripped-down variant (A1) is already strong, with all settings fixed;  A2 adds posterior calibration, and A3 captures the combined gain from adaptive scheduling and Otsu thresholding, improving the precision-recall tradeoff rather than being carefully tuned requirements for the method to work.
>
> **W2. Baselines**
>
> As suggested, we implemented TPS (the requested seed-trained nonlinear mapping baseline) and Gromov-Wasserstein (GW), followed by the same CSLS+MNN extraction. At $\alpha=0.3$, $|S|=30$ (Mistral↔OpenAI), TPS remains far below GEH on every dataset for precision/recall:
> - FiQA 15.4/18.8 → 82.7/94.0,
> - SciDocs 12.0/15.6 → 83.6/79.4,
> - SciFact 20.1/30.0 → 83.6/87.0, and
> - ArguAna 27.1/40.6 → 80.2/95.9
>
> GW is near-zero in most cases. This is consistent with the paper's main claim: under partial and unknown overlap, the issue is not insufficient mapping flexibility; the issue is that a reliable global cross-space geometry is often absent, which is precisely why GEH uses local, multi-view matching rather than a single learned transformation.
>
>
> **W3. Theory**
>
> Theorem 1 is intended as a local geometric explanation, not as a full worst-case guarantee for every production encoder. Its role is to justify why *local scaled isometry* is plausible for independently trained *contrastive encoders*, and therefore why a *scale-free local geometric signature* is key for vector linking. This is also why we focus empirically on contrastive encoders.
> GEH is then built for the non-ideal regime: normalization removes unknown local scale, kernelization downweights nonlocal distortions, multi-view voting denoises, and App. A.3 already relaxes constant isotropy to point-dependent $\kappa(x)$. In other words, the theorem motivates *why local hashing should work*, while GEH design accounts for the non-ideal regime where the assumptions hold only approximately.
>
>
>
> **W4. Scalability**
>
> As suggested, we have now added explicit million-scale runtime comparisons. On FEVER (5.42M total vectors), using Mistral ↔ OpenAI, overlap $\alpha=0.3$, $|S|=30$, and a single A100 80GB GPU, we obtain:
>
> | Method | Precision (%) | Recall (%) | Wall-clock runtime (s) |
> |--|-|-|-|
> | Procrustes | 9.03 | 0.73 | 4494 |
> | TPS | 9.06 | 0.73 | 4517 |
> | CCA | 5.22 | 0.61 | 1613 |
> | MLP | 1.93 | 0.01 | 4414 |
> | Linear | 4.13 | 0.00 | 4420 |
> | RCSLS  | 6.40    | 0.11 | 3348 |
> | **GEH** | **93.8** | **68.9** | **3328** |
>
> These results directly address the concern that the iterative multi-view procedure may become prohibitively expensive at million scale. GEH is practical on a single GPU, remains within the same order of magnitude as the fastest baseline in this table (CCA), and is faster than Procrustes/TPS/MLP/Linear/RCSLS despite its iterative design. At the same time, the accuracy gap is very large: compared with CCA, GEH improves precision from 5.22% to 93.8% and recall from 0.61% to 68.9% (roughly 18× and 113×, respectively), while being only about 2.1× slower in wall-clock time. The quality gaps relative to the slower baselines are similarly large.
>
> CSLS+MNN link extraction is also not unique to GEH: all alignment baselines incur the same stage after fitting a mapping. GEH's extra cost comes from evaluating multiple views, but each view operates in low-dimensional distance-to-anchor hash space induced by a small anchor set, rather than repeatedly matching in the original embedding space. Thus the iterative procedure adds many moderate-cost hash-space retrieval steps, rather than repeated dense searches over the original full embedding clouds. While this does not change the conservative worst-case asymptotic complexity, it allows GEH to remain practical at million scale while substantially improving accuracy.
>
> Overall, these additions address the main concerns: stronger nonlinear baselines still fall well short of GEH, the newly added staged ablation clarifies which components matter, and GEH remains practical at million scale while substantially improving accuracy.

---

> > ### Author Rebuttal · Reviewer_5oZ9 · 2026-04-08
> >
> > The authors did not answer my question about the theoretical assumption. I still think the theoretical results are still weak for an ICML submission.

---

> > > ### Author Response · Authors · 2026-04-08
> > >
> > > The **smooth manifold** hypothesis is a common approximation in machine learning, it simply posits that high-dimensional data concentrates near a lower-dimensional continuous structure, a premise underlying manifold learning [A], dimensionality reduction [B], and many theoretical analysis of deep representations [C].
> > >
> > > The **local optimality** assumes that the encoder's induced metric tensor $G_f(x)$ minimizes the per-point localized alignment-uniformity objective (Section 2.2). This is reasonable given that modern contrastive encoders are trained on massive data with extensive optimization. And the assumption’s predicted consequence of local cross-model distance consistency is supported extensively by our experiments. (Figure 1, 7, and 8). Moreover, GEH does not require exact local optimality to hold every where, it remains robust to deviation from exact local optimality through its kernelization, multi-view hashing, and posterior filtering mechanisms.
> > >
> > > The **isotropic assumption** has already been relaxed in Appendix A.3 to a point-dependent variant (A3'), yielding a region-dependent scale factor rather than a single global one. This is precisely why GEH uses cosine similarity over per-view local hashes, making the algorithm invariant to varying $\kappa(x)$.
> > >
> > > [A] Manifold Learning: What, How, and Why. Annual Review Statistics and Its Application, 2024
> > >
> > > [B] A survey of dimension reduction techniques. Technical report, Lawrence Livermore National Lab., CA (US), 2002.
> > >
> > > [C] Intrinsic dimension of data representations in deep
> > > neural networks, Neural Information Processing Systems, 2019

---

### Official Review · Reviewer_w6Ho · 2026-03-12

**Soundness:** 3
**Presentation:** 3
**Significance:** 2
**Originality:** 2
**Overall Recommendation:** 4
**Confidence:** 3

**Summary:**

The paper addresses the issue of vector linking, namely connecting two different vector representations of the same object produced by two different encoders in two separate vector spaces. The paper attempts to address the issue without learning a transformation between the spaces, but rather by embedding a small set of objects to be used as anchor points. The authors empirically show that short distances between objects are preserved between various embedding spaces, and use this as their main assumption for their methodology. In their approach, GEH, they use bagging to form the optimal set of anchors and calculate the distance to these anchors to form hashes for each unknown object in the vector spaces. In their experiments, the authors show the effectiveness of their method compared to various baselines and with different combinations of encoders.

**Compliance With Llm Reviewing Policy:**

Affirmed.

**Key Questions For Authors:**

1. Why is more recent methodology not included in the related work section?
2. Please elaborate on why the newer "Embedding alignment methods" that are brought up in the related work section is not included in the experiment baseline.
3. The embedding dimensions of the used encoders are not accounted for. Are there any impact of this? Does Local Distance Consistency still hold for models with smaller embedding dimensions.
4. The empirical experiments that support Local Distance Consistency are only conducted on retrieval-based benchmarks. Please elaborate on why not a more comprehensive benchmark is used that addresses other applications of embeddings?

**Strengths And Weaknesses:**

The paper is well written, and the underlying assumptions are elaborated on. The authors also present both theoretical and empirical support for the claimed underlying assumptions, thus providing a strong methodology section. The paper’s experimental baseline is questionable as it consists of simplistic and outdated methods. A comparison with "Embedding alignment methods" as outlined in the related work section, but trained on more than just the seeds, would be insightful as a baseline (although these methods have the advantage of training). Further, computational assessments and comparisons with other methods would also be useful. The paper also highlights the potential applications of the GEH method and shows its potential on various embedding-related tasks like retrieval and clustering. In their related work section, the authors do not include recent work like "Integrating Vector Databases across Embedding Models" (2025, Yang et. al.).
https://dl.acm.org/doi/10.1145/3769803

---

> ### Author Rebuttal · Authors · 2026-03-30
>
> We thank the reviewer for the positive feedback and for the specific questions that help improve the paper: related work, baselines, embedding dimensionality, and evaluation scope. We address each point below.
>
>
> **1. Related work**
>
> We agree Yang et al. (2025) should be cited explicitly in related work, not only later in the applications section.
> Its relation to our paper is complementary rather than overlapping:
> - Yang et al. (2025) assume paired anchors are already available and study how to *integrate* vector databases once such links are known.
> - Our paper studies how to *find* those links from two black-box embedding clouds with partial, unknown overlap.
>
> Hence, GEH is not an alternative to that method; GEH *bootstraps it* by producing the links its integration stage assumes. This is also why Section 6 already uses Yang et al. (2025) as a downstream application pipeline. We will make this distinction explicit.
>
>
> **2. Newer alignment methods as baselines**
>
> As suggested, we added two new baselines:
>
> - GW (Gromov-Wasserstein)
> - TPS (thin-plate spline)-based alignment
>
> These are stronger baselines as they allow transport-based or smooth *nonlinear* warping from seeds.
> On Mistral↔OpenAI with overlap $\alpha=0.3$ and $|S|=30$:
>
> | Dataset | TPS (Precision/Recall) | GW (P/R)   | GEH (P/R)   |
> |:---------|------------------------|------------|-------------|
> | FiQA    | 15.4 / 18.8            | 0.0 / 0.0  | 82.7 / 94.0 |
> | SciDocs | 12.0 / 15.6            | 0.0 / 0.0  | 83.6 / 79.4 |
> | SciFact | 20.1 / 30.0            | 3.5 / 0.2  | 83.6 / 87.0 |
> | ArguAna | 27.1 / 40.6            | 13.3 / 0.4 | 80.2 / 95.9 |
>
> For methods that can additionally exploit unlabeled full clouds (e.g., GW), doing so did not help under partial overlap and often collapsed to near-zero matching.
>
> These suggest that simply increasing mapping flexibility is insufficient under partial, unknown overlap. The bottleneck is the absence of *global* cross-space geometry for partial overlap + cross-model distortion.
>
> From a broader perspective, under our targeted scenario and corresponding protocol, the directly comparable baseline class is post hoc alignment without modifying encoders. Our original evaluation already covered this family with Linear, CCA, MLP, RCSLS, Procrustes, and also AO (Cannistraci et al., 2023).  By contrast, the more recent representation-compatibility/alignment methods discussed in Related Work require training-time model modification, raw-data/model access, or additional supervision. Hence, they are not directly comparable in the black-box vector-linking setting.
>
> **3. Smaller embedding dimensions**
>
> Thanks for the suggestion. The current experiments already span substantial dimensional variation across models (Appendix Table 4): 1024, 1536, 3584, 3840, 4096. LIC is observed across these heterogeneous dimensions (main Fig. 1; Appendix Fig. 7), suggesting that it is not tied to matched ambient dimensionality.
>
> To further test the reviewer's question, we ran an additional low-dimensional experiment: OpenAI (256d) vs Mistral (1024d) on SciFact. Local retrieval consistency still remains clearly positive:
>
> | k            |    1 |    10 |    50 |   100 |
> |--------------|------|-------|-------|-------|
> | Mean Jaccard | 0.68 | 0.320 | 0.283 | 0.280 |
>
> And GEH remains effective for linking the two:
>
> | Method | Precision | Recall |
> |--------|-----------|--------|
> | Proc   |       4.5 |    3.9 |
> | TPS    |       4.6 |    4.1 |
> | RCSLS  |      33.0 |    5.9 |
> | GEH    |      71.5 |   66.0 |
>
> So the answer is *yes*: lower dimensions slightly weaken neighborhood stability due to their weaker expressiveness, but LIC remains strong for effective linking.
>
>
> **4. Benchmark**
>
> We used BEIR since our Section 2 analysis and evaluation pipeline requires large corpora with clean object identity across encoders so the same items can be compared as points in two embedding clouds. This choice is also aligned with the scope of the paper: both our theory and method target contrastive encoders, and text retrieval embeddings are a natural and widely used family of them. However, we agree LIC needs broader verification, so we evaluated LIC on clustering datasets and confirmed the property firmly holds (https://anonymous.4open.science/r/fig-8625/c.pdf).  Additionally, our evaluation extends beyond retrieval to vector database integration and cross-model clustering. We will clarify this broader scope in the revision and state more explicitly that extending beyond contrastive text encoders is important future work.
>
> $\newline$
>
> Regarding computational comparisons, we also ran a million-scale scalability test. On FEVER (5.42M vectors), GEH achieved 93.8 / 68.9 precision/recall in 3328s on one A100 GPU; CCA took 1613s but only reached 5.22 / 0.61, while Procrustes/TPS/MLP/Linear/RCSLS were slower than GEH and far less accurate, like CCA (see response to Reviewer 5oZ9 for a full report).

---

> > ### Author Rebuttal · Reviewer_w6Ho · 2026-03-31
> >
> > We believe that the All comments are fully resolved. We suggest changing soundness til 4

---

### Decision · Program_Chairs · 2026-04-30

**Decision:**

Accept (regular)

**Comment:**

This submission received diverging scores: two weak accepts and one weak reject. While all the reviewers appreciated the novel problem of vector linking, an insightful geometric observation, and good empirical results, there were main concerns regarding strong assumptions, limited comparisons, and missing analyses. There was a rebuttal and discussion, and they addressed many of the reviewers' concerns. Reviewers w6Ho and 2hBy recommended weak accept, with their main concerns resolved, but Reviewer 5oZ9 maintained a weak reject, criticizing the theoretical assumptions as unrealistic, which is partly shared with Reviewer 2hBy. Carefully reading the reviews, rebuttal, and discussions, AC finds that the authors’ argument for the assumptions, such as smooth manifold, local optimality, and isotropicity, is reasonable enough (they can be plausible or even relaxed in real-world applications), and most of the other concerns are addressed in the rebuttal. AC thus recommends acceptance and encourages the authors to carefully incorporate the rebuttal in their camera-ready version.